# The influence of HLA genetic variation on plasma protein expression

Chirag Krishna[1] ✉, Joshua Chiou [1], Saori Sakaue [2,3,4,5], Joyce B. Kang[2,3,4,5], Stephen M. Christensen [1], Isac Lee[1], Melis Atalar Aksit[1], Hye In Kim[1], David von Schack[1], Soumya Raychaudhuri [2,3,4,5], Daniel Ziemek[1] & Xinli Hu [1] ✉

Genetic variation in the human leukocyte antigen (HLA) loci is associated with risk of immune-mediated diseases, but the molecular effects of HLA polymorphism are unclear. Here we examined the effects of HLA genetic variation on the expression of 2940 plasma proteins across 45,330 Europeans in the UK Biobank, with replication analyses across multiple ancestry groups. We detected 504 proteins affected by HLA variants (HLA-pQTL), including widespread *trans* effects by autoimmune disease risk alleles. More than 80% of the HLA-pQTL fine-mapped to amino acid positions in the peptide binding groove. HLA-I and II affected proteins expressed in similar cell types but in different pathways of both adaptive and innate immunity. Finally, we investigated potential HLA-pQTL effects on disease by integrating HLA-pQTL with fine-mapped HLA-disease signals in the UK Biobank. Our data reveal the diverse effects of HLA genetic variation and aid the interpretation of associations between HLA alleles and immune-mediated diseases.

Major histocompatibility complex (MHC) molecules, encoded by the highly polymorphic human leukocyte antigens (HLA) genes in humans, present self and foreign antigens for recognition by T cells and thus comprise the foundation of the adaptive immune response for a wide range of diseases. MHC presentation of viral peptides and neoantigens facilitates clearance of infectious pathogens and cancerous cells, respectively[1–3]; in contrast, aberrant presentation of self-peptides by HLA risk alleles is thought to underly the pathogenesis of autoimmune disease[4,5]. Polymorphism in the HLA-I (*HLA-A*, *B*, and *C*) and HLA-II (*HLA-DRB1*, *DQB1*, *DQA1*, *DPB1*, and *DPA1*) loci is a hallmark of genetic risk for autoimmune diseases; for example, genetic variation in the HLA-II loci explains up to 30% of the genetic heritability in rheumatoid arthritis and type 1 diabetes[4–8].

Genetic studies linking HLA genes to disease risk have historically been complicated by extreme polymorphism and linkage disequilibrium (LD) in the HLA region. Advances in HLA imputation and statistical association methods have largely overcome these challenges[5,9], allowing HLA fine-mapping studies which have pinpointed amino acid positions within the MHC peptide binding groove[5,7]. These studies lend support to the idea that variation in antigen presentation is likely the pathogenic mechanism by which HLA polymorphism affects disease risk.

Despite much evidence linking HLA to disease risk, the extent to which HLA genetic variation—in particular, the two-field alleles that encode the amino acid sequence of the MHC (e.g., HLA-DRB1*04:01)— affects quantitative traits such as protein expression levels is incompletely understood[10–13]. Plasma proteins can serve as biomarkers for disease risk and clinical outcomes, and protein quantitative trait loci (pQTL) mapping has improved our understanding of the mechanisms underlying genetic associations with disease[14]. From an immunological perspective, it is conceivable that T cell receptor (TCR) recognition of peptides presented by MHC molecules may lead to changes in protein expression in the corresponding T cell or antigen-presenting cell (APC). Moreover, it is possible that certain HLA alleles modulate the strength of the immune response to disease, which in turn may drive differences in protein expression. Thus, a systematic investigation of

[1]Pfizer Research and Development, Pfizer Inc., Cambridge, MA, USA. [2]Center for Data Sciences, Brigham and Women's Hospital, Harvard Medical School, Boston, MA, USA. [3]Divisions of Genetics and Rheumatology, Department of Medicine, Brigham and Women's Hospital, Harvard Medical School, Boston, MA, USA. [4]Program in Medical and Population Genetics, Broad Institute of MIT and Harvard, Cambridge, MA, USA. [5]Department of Biomedical Informatics, Harvard Medical School, Boston, MA, USA. ✉e-mail: Chirag.Krishna@pfizer.com; xinli.hu@pfizer.com

how HLA variants (e.g., two-field alleles and amino acids) affect protein expression would reveal the intermediate molecular effects of HLA polymorphism, and aid the interpretation of associations between HLA polymorphism and disease.

To address this question, we used data from the UK Biobank, a massive prospective cohort that has facilitated critical insights into the genetic determinants of disease through genome-wide genotyping and health phenotype collection on all participants[15]. A recent flagship pQTL effort led by the UK Biobank Pharma Proteomics Project (UKB-PPP) quantified and performed rigorous quality control of 2940 plasma protein levels measured with the Olink platform across 54,306 UK Biobank participants[14,16]. We used these data as input to our study, to comprehensively investigate associations between HLA genetic variation—here defined as one-field alleles, two-field alleles, or amino acids—and plasma protein expression. We term HLA genetic variants that associate with plasma protein levels "HLA-pQTL", and affected proteins "pGenes". Following systematic identification of lead HLA-pQTL, we performed conditional analyses and fine mapping to broaden the scope of our discovered HLA-pQTL effects and identify associations specific to HLA class I (HLA-I) or HLA class II (HLA-II) alleles. We examined the expression of proteins affected by HLA genetic variants on two large single-cell RNA-seq (scRNA-seq) atlases—one of the peripheral immune cells from 909 donors from Yazar et al.[17], and a lung atlas with three donors containing lung-resident immune and non-immune cells from Travaglini et al.[18]. Finally, we integrated our fine-mapped HLA-pQTL with fine-mapped HLA-disease analyses in the UK Biobank from Sakaue et al.[11] to create an atlas and of potential HLA-pQTL effects on disease, and hypotheses about the mechanisms underlying HLA genetic associations with disease.

## Results

### Discovery and fine mapping of HLA-pQTL in the UK Biobank

To probe the effects of HLA genetic variation on plasma protein expression, we first split the UKB-PPP participants into discovery ($N = 34,490$ individuals) and replication ($N = 10,840$) cohorts, as also performed by the flagship UKB-PPP study[14] (section "Methods") (Fig. 1a). In brief, the UKB-PPP discovery cohort is a "randomized baseline" European cohort highly representative of the overall UK Biobank cohort and is neither enriched nor depleted for any particular disease.

After excluding individuals for whom genotypes did not pass stringent quality control filters recently established for HLA association studies[9] ($N = 674$) (section "Methods"), we used the multi-ancestry reference panel[11,12] included in the Michigan Imputation Server[19] to impute HLA one- and two-field alleles and amino acids at MAF ≥ 1% and imputation dosage $R^2 ≥ 0.7$ across 45,330 Europeans in the UK Biobank ("UKB-PPP cohort"). The final set of HLA variants included 2284 one-field alleles (e.g., DRB1*04), two-field alleles (e.g., DRB1*04:01), and amino acids (e.g., AA_DRB1_position13_exon2_FLS) in the classical HLA-I (HLA-A, B, and C) and II (HLA-DRB1, DQB1, DQA1, DPB1, DPA1) genes. We used these data to fine-map HLA-pQTL. Importantly, HLA fine mapping is distinct from standard fine-mapping procedures—in HLA studies, it is assumed that coding variation is the primary driver of the heritability of the trait, which is not assumed by standard fine mapping. Moreover, in contrast to standard fine mapping, HLA fine mapping is squarely related to the biology of the MHC—namely, the presentation of peptide antigens. Thus, HLA fine mapping seeks to discover amino acid polymorphisms—either through single-marker conditional tests or conditional haplotype analysis jointly testing all amino acid variants at a particular position—that are associated with a trait, as these are the functional components of the MHC that are responsible for antigen presentation. These core assumptions of HLA fine mapping are consistent with many prior studies which have investigated the effects of HLA genetic variation on disease risk[5,7,9–12,20,21].

To fine-map HLA-pQTL, we first tested the association between each imputed HLA variant and scaled, covariate-residualized inverse rank-normalized protein expression with REGENIE LOCO (leave one chromosome out) values subtracted (section "Methods") using linear regression models. The LOCO values are known to more precisely account for population structure, sample relatedness, and LD—indeed, according to the original REGENIE and flagship pQTL studies, this approach performs these corrections while avoiding proximal contamination, which can result in reduced power of genetic association analyses[22,23]. We termed proteins whose expression was affected by an HLA genetic variant as "pGenes" or "HLA-pGenes". We included demographic and technical covariates (Methods); additionally, to account for latent sources of variation in the data and the presence of diseases in each cohort, we included the top 20 protein expression PCs as covariates (Supplementary Fig. 1). We set a stringent Bonferroni significance threshold of $P ≤ 1.70 × 10^{-11}$ ($5 × 10^{-8}$/2940 proteins tested). To ensure the robustness of lead HLA-pQTL, we performed 1000 permutations in which we randomized protein expression values across individuals in the discovery cohort (section "Methods"). This analysis revealed that our statistical tests had little chance of inflation, as none of the pGenes achieved a $p$ value in the random data more significant than those observed from the real data (Supplementary Fig. 2).

We identified 504 unique plasma proteins ("HLA-pGenes"; 17.1% of the 2940 proteins tested, Supplementary Data 1) in the discovery cohort whose expression was affected by at least one HLA variant. To identify independent HLA variants affecting protein expression, we performed conditional analysis by adjusting for the lead HLA variant for each protein, which identified 835 additional independent HLA variants (Supplementary Data 2). Of the 504 pGenes, 305 had two or more independent associations (Supplementary Fig. 3a). Notably, despite high LD within the MHC region, the lead and conditionally independent variants demonstrated low pairwise $R^2$ (Supplementary Fig. 3b), suggesting that the conditionally independent HLA-pQTL reported here are not simply an artifact attributable to LD. Strikingly, the majority (478/504; 94.8%) of the HLA-pGenes were in *trans* (i.e., proteins outside the MHC region on chromosome 6) (Fig. 1b), suggesting that HLA genetic variation has widespread effects on protein expression. Importantly, here we defined *trans* HLA-pQTL as HLA-pQTL involving pGenes outside of the MHC region (hg19 chr6: 28,000,000–34,000,000) due to the strong LD within the MHC region. Notably, HLA-DRB1 variants were associated with the largest number of proteins (Fig. 1b). Prior studies have shown that alleles and fine-mapped amino acid positions in HLA-DRB1 drive the risk of autoimmune diseases and affect the TCR repertoire[24]. Thus, we performed an exploratory analysis in which we conditioned each lead HLA-pQTL on all two-field HLA-DRB1 alleles. We found that after adjusting for all two-field HLA-DRB1 alleles, the majority of HLA-I lead pQTL remained significant (191 remained significant at $P ≤ 5 × 10^{-8}$, out of 225 lead HLA-I pQTL total), while the majority of HLA-II lead pQTL lost significance (18 remained significant at $P ≤ 5 × 10^{-8}$, out of 279 lead HLA-II pQTL total tested) (Supplementary Fig. 3c), suggesting that HLA-DRB1 alleles are a major genetic determinant of plasma protein expression. Our results corroborate the important role of genetic variation in HLA-DRB1 in influencing both disease risk and intermediate molecular traits.

Moreover, we asked whether plasma protein expression levels were affected by genetic variants in the HLA class III region, which contains the complement genes and others associated with the immune response[25]. Of our 504 pGenes, 29 of them had a significant class III SNP pQTL from the flagship pQTL study by Sun et al. [23]. Thus, we performed a sensitivity analysis, in which for each of these 29 pGenes we fit a combined model assessing the effect of both our lead HLA-pQTL and the class III SNP pQTL (Supplementary Fig. 3d; Supplementary Data 3). This analysis revealed that the expression of only 8

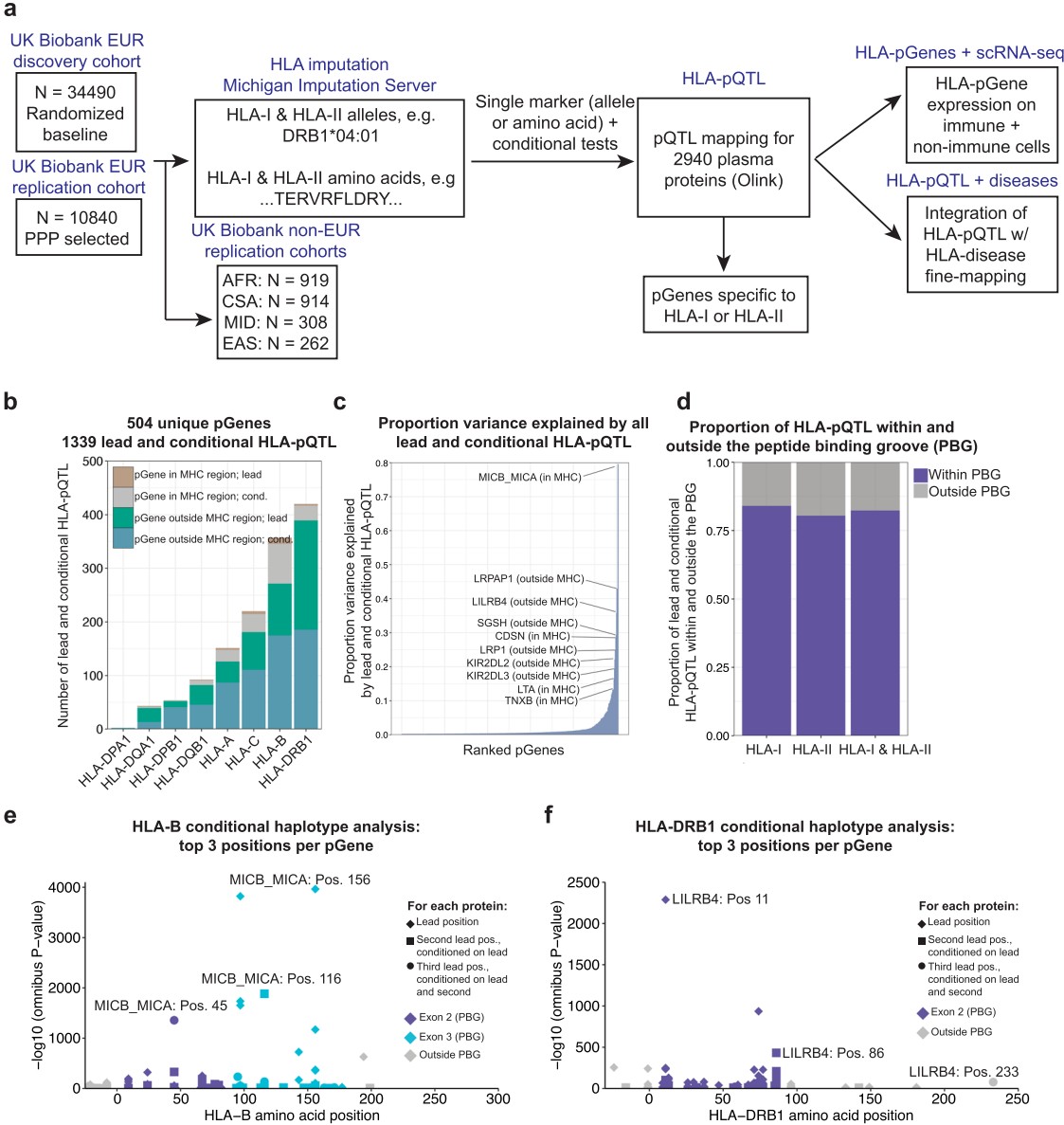

**Fig. 1 | HLA-pQTL in the UK Biobank. a** Schematic of the study. We used imputation of one and two-field HLA alleles and amino acid polymorphisms across 45,330 Europeans from the UKB Pharma Proteomics Project (UKB-PPP) together with plasma protein levels quantified using the OLINK platform as input to pQTL mapping for 2940 proteins. For each protein, we tested each HLA variant (allele or amino acid) independently in a multivariable linear model incorporating covariates as specified by the UKB-PPP. We performed replication studies in subgroups of non-European ancestry African (AFR); Central/South Asian (CSA), Middle Eastern (MID), East Asian (EAS), and admixed American (AMR). Following systematic pQTL mapping, conditional, and fine-mapping analyses, we examined the expression of HLA-pGenes on two independent single-cell RNA-sequencing datasets, and integrated our HLA-pQTL fine mapping with summary statistics of HLA-disease fine mapping from Sakaue et al.[11]. **b** Number and location of proteins (pGenes) in the discovery cohort affected by lead HLA variants corresponding to each HLA-I and HLA-II locus. **c** For each ranked pGene in the discovery cohort, the proportion of protein

expression variance explained by all lead and conditional HLA-pQTL. **d** Proportion of all fine-mapped lead and conditional HLA-pQTL in the discovery cohort within and outside the peptide binding groove (exons 2 and 3 for HLA-I; exon 2 for HLA-II). **e** Top three significant ($P < 5.0 \times 10^{-8}$) amino acid positions identified via conditional haplotype analysis for HLA-B. The protein with the lowest $P$ value across all proteins tested is labeled. $P$ value determined via two-sided ANOVA and omnibus test; the threshold of $P < 5.0 \times 10^{-8}$ is the commonly used conservative GWAS threshold after Bonferroni correction for the number of genome-wide SNPs. **f** Top three significant ($P < 5.0 \times 10^{-8}$) amino acid positions identified via conditional haplotype analysis for HLA-DRB1. $P$ value determined via two-sided ANOVA and omnibus test; the threshold of $P < 5.0 \times 10^{-8}$ is the commonly used conservative GWAS threshold after Bonferroni correction for the number of genome-wide SNPs. The protein with the lowest $P$ value across all proteins tested is labeled together with its top three significant conditional positions.

of the 29 pGenes was driven by class III variants, whereas 16 of the 29 were driven either primarily by our lead HLA-pQTL or both the lead HLA-pQTL and the class III SNP pQTL independently. For the remaining five pGenes, neither our original lead HLA-pQTL nor the class III SNP pQTL remained significant in the combined model, suggesting that the two signals may be in tight LD. Altogether, these data indicate that

both genetic variation in the classical HLA loci and class III region can independently drive protein expression.

Given the strong associations of HLA genetic variation with immune-mediated diseases, we explored whether removing individuals with HLA-associated diseases—either major autoimmune diseases identified from prior bespoke GWAS, or HLA-associated diseases identified in an analysis of the UK Biobank by Sakaue et al.[11]—impacted

the discovery of the 504 lead HLA-pQTL in the discovery cohort. To this end, we removed 6833 individuals with either diagnoses or deaths due to immune diseases (identified via ICD-10 codes) in the UKB discovery cohort (Supplementary Data 4) and repeated the analysis. We found that of the 504 pGenes discovered initially, 500/504 (99.2%) remained significant at $P < 5 \times 10^{-8}$ (Supplementary Data 4). Effect sizes were also strongly correlated between the full discovery cohort and discovery cohort without HLA-associated diseases (Supplementary Fig. 3e). These data underscore the robustness of our HLA-pQTL model, and suggest that the HLA-pQTL reported in our study are not confounded by the presence of HLA-associated diseases in the UK Biobank.

For each of the 504 pGenes, we quantified the proportion of protein expression variance explained by all independent HLA-pQTL (Fig. 1c). Notably, for some proteins, HLA-pQTL collectively explained more than 20% of the protein expression variance (e.g., 42.89% attributed to LRPAP1; SGSH 29.24%; LRP1 24.90%; KIR2DL2 22.07%). Comparing the variance explained by HLA-pQTL to variance explained by all SNPs from the flagship UKB-pQTL study, we found that for several pGenes (e.g., MICB_MICA; LRP1), the majority of the total SNP variance explained is accounted for by HLA genetic variation (Supplementary Fig. 3f). Next, we quantified the proportion of HLA amino acid variants from our single-marker tests that are located within the peptide binding groove (exons 2 & 3 for the HLA-I loci; exon 2 for the HLA-II loci)[26]. We found that 84.0% of HLA-I pQTL and 80.4% of HLA-II pQTL were located within the peptide binding groove, suggesting that for most proteins, variation in antigen presentation is the likely mechanism by which HLA variants affect protein expression (Fig. 1d).

To pinpoint HLA amino acid positions that affect protein expression, we performed omnibus and conditional haplotype tests for all 2940 proteins (section "Methods"; Fig. 1e, f; Supplementary Fig. 4a–e and Supplementary Data 5). These analyses revealed enrichment of the fine-mapped amino acid positions in peptide binding groove, consistent with the enrichment observed for single-marker lead and conditional HLA-pQTL (Supplementary Fig. 4f). These analyses also revealed highly statistically significant associations between amino acid positions previously associated with the risk of autoimmune diseases[5,7] with disease-relevant proteins. For example, the protein most significantly associated with the HLA-B locus was MICB_MICA (MICB and MICA were measured together as a single analyte) (Fig. 1e). MICB_MICA serves as a critical ligand for the NKG2D receptor expressed by NK and gamma-delta T cells, with prior reports suggesting a critical role for this protein in inflammation[27] and anti-tumor immunity[28]. The most significant signal at the DRB1 locus was the association with the LILRB4 protein, at HLA-DRB1 position 11 (omnibus $P = 1.0 \times 10^{-2288}$), which was previously shown to be associated with the risk of rheumatoid arthritis[5]. LILRB4 is a myeloid cell receptor broadly associated with tolerogenic APCs[29]. Importantly, the LILR family of proteins (multiple of which were affected by HLA genetic variants in our analysis) are thought to bind MHC molecules, suggesting that HLA-pQTL may reflect cell–cell interactions between APCs and myeloid cells expressing LILR proteins. To identify additional independent amino acid positions in the DRB1–DQB1–DQA1 superlocus (section "Methods") associated with LILRB4 expression, we conditioned on DRB1 position 11, and observed a second independent position at DRB1 position 86 (omnibus $P = 1.0 \times 10^{-432}$), and a third independent position still in DRB1 at position 233 (omnibus $P = 1.0 \times 10^{-78}$). Moreover, HLA-DRB1 positions 67 and 13 were associated with the expression of the LILRA2 protein, demonstrating that different amino acid positions in the HLA can affect the expression of multiple proteins within the same gene family. Similar analyses for other proteins identified conditionally independent amino acid positions previously associated with disease. For example, the expression of CPVL, an enzyme with loosely defined roles in antigen processing[30], was affected by HLA-DRB1 position 74

($P = 1.0 \times 10^{-936}$), position 11 ($P = 6.10 \times 10^{-37}$), and position 71 ($P = 7.25 \times 10^{-13}$). These amino acid positions in DRB1 have previously been shown to be associated with the risk of type 1 diabetes[7] and rheumatoid arthritis[5], suggesting that HLA-DRB1 effects on CPVL and antigen presentation may influence type 1 diabetes and rheumatoid arthritis risk. The highly statistically significant associations between HLA polymorphism and expression of proteins such as LILRB4/A2, CPVL, CD74, B2M, CD8, CD4, and genes of the KIR family suggest that that the strongest effects of HLA genetic variation on protein expression are related to the immune response, including components of the antigen processing and presentation pathway.

## Replication of lead HLA-pQTL

To evaluate the replicability of lead HLA-pQTL from the discovery cohort, we performed HLA-pQTL mapping in the UKB-PPP replication cohort. The replication cohort consists of individuals from the UK Biobank enriched for diseases manually selected by the consortium partners as well as COVID-19 imaging participants; thus, it is more heterogeneous in terms of demographics and disease representation than the discovery cohort. We note that the discovery cohort in the flagship UKB-PPP analysis consisted entirely of Europeans while the replication cohort was comprised of individuals with mixed ancestries; thus, we first limited the replication cohort to Europeans to ensure consistency with the discovery cohort. We observed high replicability of lead HLA-pQTL, both by significance and concordance of effect sizes (96% at $P < 0.05$ in the replication cohort; Spearman $\rho = 0.94$ between discovery and replication; Supplementary Fig. 3g and Supplementary Data 6). Combined analysis of both the discovery and replication EUR cohorts together yielded an additional 119 pGenes (623 total, Supplementary Data 11). Furthermore, we explored ancestry-specific HLA-pQTL in four additional genetic ancestry groups including African (AFR; $N = 919$), Central/South Asian (CSA; $N = 914$), Middle Eastern (MID; $N = 308$), East Asian (EAS; $N = 262$) (Supplementary Fig. 5; Supplementary Data 7–10). As in our EUR replication analyses, we detected strong correlations of effect sizes between the discovery and non-EUR replication cohorts. Furthermore, at the stringent significance cutoff of $P < 1.7 \times 10^{-11}$ used in the original EUR discovery cohort, we detected one pGene (DPP7) that was significant in the AFR group but not the EUR group. While these results could suggest that there may exist ancestry-specific HLA-pQTL, we caution against overinterpretation given the very low sample size relative to the EUR group. Critically, more samples in non-EUR ancestry groups are urgently needed to validate the observed HLA-pQTL effects in our analyses.

## Distinct *trans* regulatory effects of HLA-I and HLA-II polymorphism

We next sought to more closely examine the biology of proteins affected by HLA polymorphism, including two-field alleles associated with the risk of immune-mediated diseases. We hypothesized that our analyses could help understand the unique regulatory effects of the HLA-I vs HLA-II loci. Indeed, it is generally well-established that HLA-I and HLA-II are recognized by CD8+ and CD4+ T cells respectively, and that HLA-II-restricted antigens are qualitatively different compared to HLA-I antigens[31]. However, the extent to which the *trans* regulatory effects of HLA-I and HLA-II genetic variation differ is unclear.

We first assembled two sets of pGenes from the discovery cohort that were exclusively affected by HLA-I ($N = 224$ pGenes) or HLA-II ($N = 280$ pGenes) lead variants. To rule out any potential confounding of HLA-II genetic variation influencing HLA-I-specific pQTL or vice versa, we tested the effects of lead HLA-I pQTL (e.g., the HLA-I pGene set) while also adjusting for all two-field alleles corresponding to all HLA-II loci, and vice versa. We found that at $P < 5 \times 10^{-8}$, 72.7% (163/224) of lead HLA-I pQTL remained significant after adjusting for all HLA-II alleles (Supplementary Data 12), and 73.5% (205/280) of lead HLA-II pQTL remained significant after adjusting for all HLA-I alleles

(Supplementary Data 13). Next, we performed gene ontology enrichment analysis in each pGene set using the full set of Olink panel proteins as the background, and constructed *trans* HLA-pQTL networks separately for HLA-I and HLA-II based on the enriched gene families (Fig. 2a, b; Supplementary Figs. 6, 7; Supplementary Data 14, 15). These analyses showed that inhibitory receptors such as proteins of the KIR family and some members of the LILR family (e.g., LILRB1 and LILRB2) were HLA-I-specific, as well as proteins involved in TNF binding and NK activity (MICB). In contrast, many chemokines (CCL and CXCL proteins) were HLA-II specific, in addition to general signaling and pattern recognition receptors such as TLR1, IL1, and IGF1R, among others. The effect of HLA polymorphism on KIR and pattern recognition receptors suggests that HLA polymorphism also affects components of innate immunity, in addition to its traditional role in the adaptive immune response. Notably, both HLA-I-specific and HLA-II-specific pGenes included known drug targets for autoimmune diseases, infectious diseases, and cancer such as LAG3, PDCD1, TNF, and ACE2 affected by HLA-I; and IL18, TREM2, VSIG4, and TLR1 by HLA-II. The presence of well-known inhibitor receptors, molecules related to T and NK function, cytokines, and chemokines further suggests that the strongest effects of HLA genetic variation on protein expression are related to the canonical immune response (Fig. 2).

### Tissue and cell-type expression of HLA-pGenes

To explore the expression patterns of pGenes identified in our analysis, we first asked whether our pGenes were enriched in tissues-specific gene sets developed from gene expression data from Uhlén et al.[32] (Supplementary Fig. 8). This analysis revealed enrichment of HLA-pGenes in spleen, lymph node, tonsil, appendix, bone marrow, and lung, which are thought to be extensively immune infiltrated.

Motivated by this result, we next asked to what extent our HLA-pGenes are transcriptionally expressed in peripheral blood immune cells. To explore this question, we obtained a doublet-cleaned version of a peripheral blood immune cell scRNA-seq atlas[17]—comprised of 909 donors and 765,075 PBMC immune cells—to explore the expression patterns of HLA-I-specific and HLA-II-specific pGenes (Fig. 3a, b). We then plotted all of the HLA-I-specific and HLA-II-specific pGenes on the re-processed Yazar et al. atlas (Fig. 3c; Supplementary Data 16). This analysis demonstrated strong cell-type-specificity of HLA-pGenes, with subsets of the pGenes preferentially expressed on myeloid cells, T cells, and B cells. Curiously, this analysis also showed that despite the pGenes' specificities to HLA-I or HLA-II, the cell-type expression patterns are largely overlapping between the sets of pGenes. This result is especially intriguing—despite the well-known, specific expression and activity of HLA-II in APCs[33], HLA-II genetic variants affect the expression of genes in non-APCs (i.e., T cells) as well.

We next sought to explore the cell-type expression patterns of HLA-pGenes in non-immune cells compared to immune cells. Given the enrichment of HLA-pGenes in lung-specific genes, we obtained a lung cell scRNA-seq atlas from Travaglini et al.[18] and plotted HLA-I-specific and HLA-II-specific pGenes. Importantly, we chose this dataset because it is comprised of healthy lung unaffected by disease, and has an excellent representation of both immune and non-immune cells. We used cluster definitions exactly as specified by the Travaglini et al. study.

As observed in the immune cell atlas, we saw no clear differences in cell-type-specific expression between HLA-I-specific and HLA-II-specific pGenes in the lung; but we did detect pronounced expression of HLA-pGenes in non-canonical immune cells, e.g., in fibroblasts and lung epithelial cells (Supplementary Fig. 9; Supplementary Data 17). Importantly, the pGenes we originally identified in the UK Biobank were found using peripheral blood data, and not lung tissue; as such, we caution against overinterpretation of these data. However, the data do suggest that pGenes affected by HLA genetic variants may be expressed comparably in non-immune cells compared to immune

cells. More generally, our data motivate future single-cell *trans* HLA-pQTL studies to more formally investigate the myriad effects of HLA genetic variation on gene expression in disease-relevant tissues and cell types.

To extend our scRNA-seq analysis which suggested both immune and non-immune functions of the pGenes, we manually examined the individual pGenes identified in our analysis, particularly among the top 50 strongest associations by *p* value (Fig. 3d). For example, SFTPD is generally thought of as a pattern recognition receptor, and thus might be considered a part of the innate immune response. The identification of this pGene is surprising and novel in the context of HLA biology; given the primary role of the HLA genes in antigen presentation to T cells, a plausible expectation may have been that HLA genetic variation would affect only T cell-related pathways. LRPAP1 is the LDL receptor-related protein 1; aside from a recent report suggesting that LRPAP1 may facilitate immune evasion from viruses[34], it is both unclear and intriguing to speculate as to how LRPAP1 expression may be affected by HLA genetics. ENPP6 is an enzyme involved in choline metabolism; NPTX1 is a member of the pentraxin gene family with reported roles in endoplasmic reticulum stress[35], and SGSH encodes the enzyme *N*-sulfoglucosamine sulfohydrolase, with no known roles in immunity. Altogether, these data highlight the potential of using HLA-pQTL associations to discover novel relationships between HLA genetic variation and proteins with unclear connections to the immune response.

In addition, we sought to explore whether both our HLA-pQTL and *cis*-eQTLs of the HLA-pGenes identified in our study independently contribute to protein expression. For 266 of our 504 HLA-pGenes, we detected significant *cis*-eQTLs from the GTEx whole blood cohort that were also detected in the UK Biobank. For each of these 266 pGenes, we fit a combined model of plasma protein expression including both the lead HLA-pQTL and the *cis*-eQTLs for the corresponding gene from GTEx. Notably, this analysis revealed that of the 266 pGenes, 265 HLA-pQTL remained significant in this analysis at $P < 5 \times 10^{-8}$ (Supplementary Data 18). The only pGene that lost significance for HLA-pQTL in this analysis was BTN3A2, which is within the MHC region. For 157 of the pGenes, both the HLA-pQTL and *cis*-eQTLs remained significant in the combined models (Supplementary Data 18). Thus, these data suggest that there are no long-range confounding effects between HLA-pQTL and *cis*-eQTLs of our pGenes; furthermore, both HLA-pQTL and *cis*-eQTL can independently contribute to plasma protein expression.

### Atlas of proteins affected by disease-associated HLA alleles

To explore the relevance of HLA-pQTL to human disease, we created an atlas of all proteins affected by HLA-I or HLA-II alleles previously associated with immune-mediated diseases (Table 1; Supplementary Data 19). To create the atlas, we manually searched the literature for major autoimmune diseases that have prior reported genetic associations with HLA. Furthermore, since the one-field and two-field alleles are often described as being associated with disease (e.g., without amino acid-level fine-mapping information), we performed an additional conditional analysis for each pGene using only one and two-field alleles, and asked which HLA-pQTL are associated with immune-mediated diseases. Notably, the DRB1*04:01 allele, strongly associated with type 1 diabetes[7] and rheumatoid arthritis[5], affected the expression of multiple proteins and in particular drove higher expression of NFATC3 ($\beta = 0.07$; $P = 5.49 \times 10^{-45}$), which is a key transcription factor regulating helper T cell differentiation[36]. The C*06 allele, which is a strong genetic determinant of psoriasis[37], drove higher expression of CD8A ($\beta = 0.22$; $P = 1.0 \times 10^{-363}$), providing further evidence for the role of CD8 T cells in psoriasis[38].

We sought to more systematically explore the relationships between HLA-pQTL and HLA associations with disease. Standard colocalization methods are limited by the extreme polymorphism and

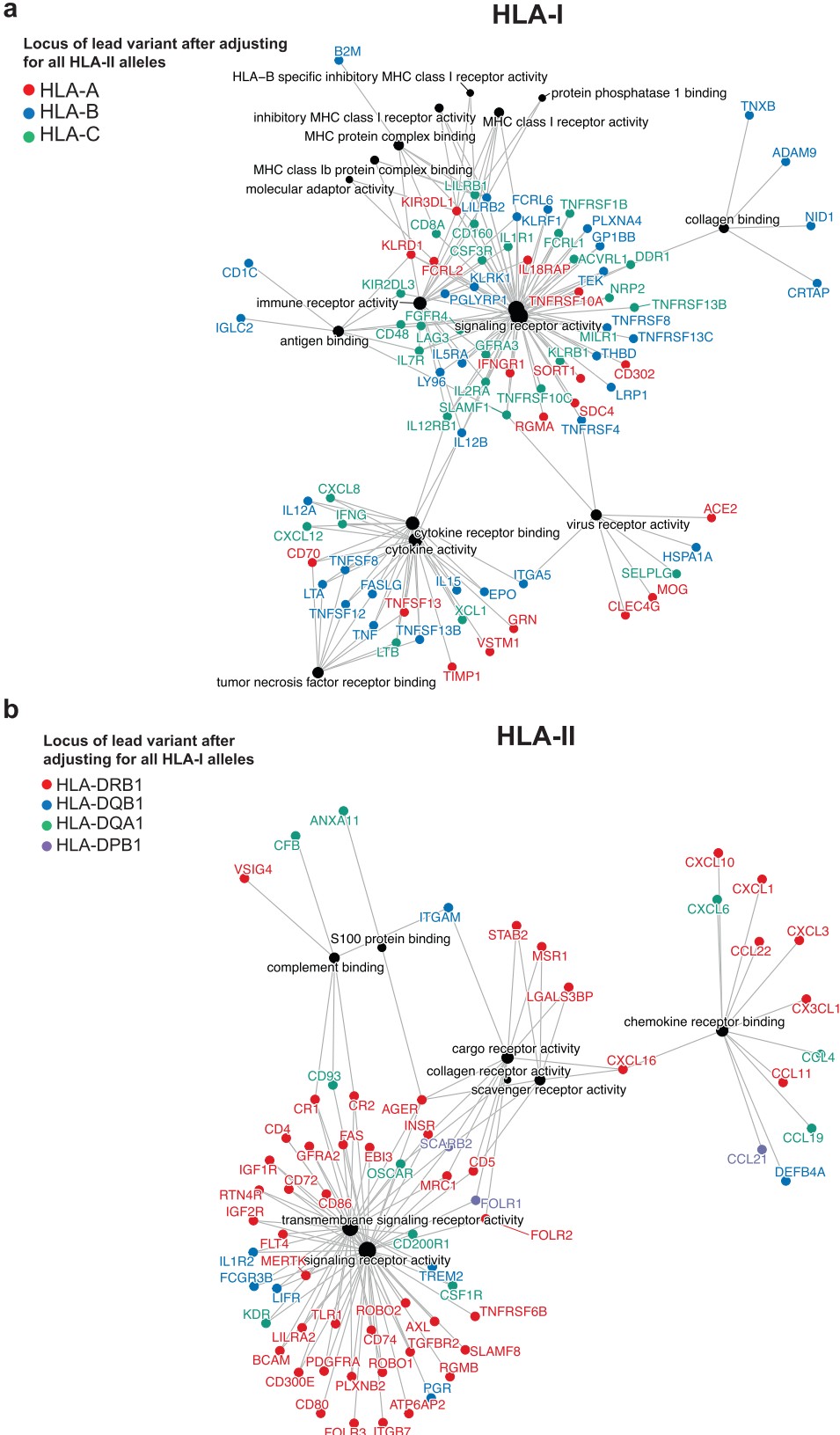

**Fig. 2 | *Trans* HLA-pQTL networks and gene families affected by HLA-I and HLA-II genetic variation. a** Network depicting proteins and gene ontology terms affected (at FDR *P* < 0.0001) by only HLA-I lead variants after adjusting for all HLA-II two-field alleles. Edges connect proteins to gene ontology terms. **b** Network depicting proteins affected (at FDR *P* < 0.0001) by only HLA-II lead variants from after adjusting for all HLA-I two-field alleles. Edges connect proteins to gene ontology terms.

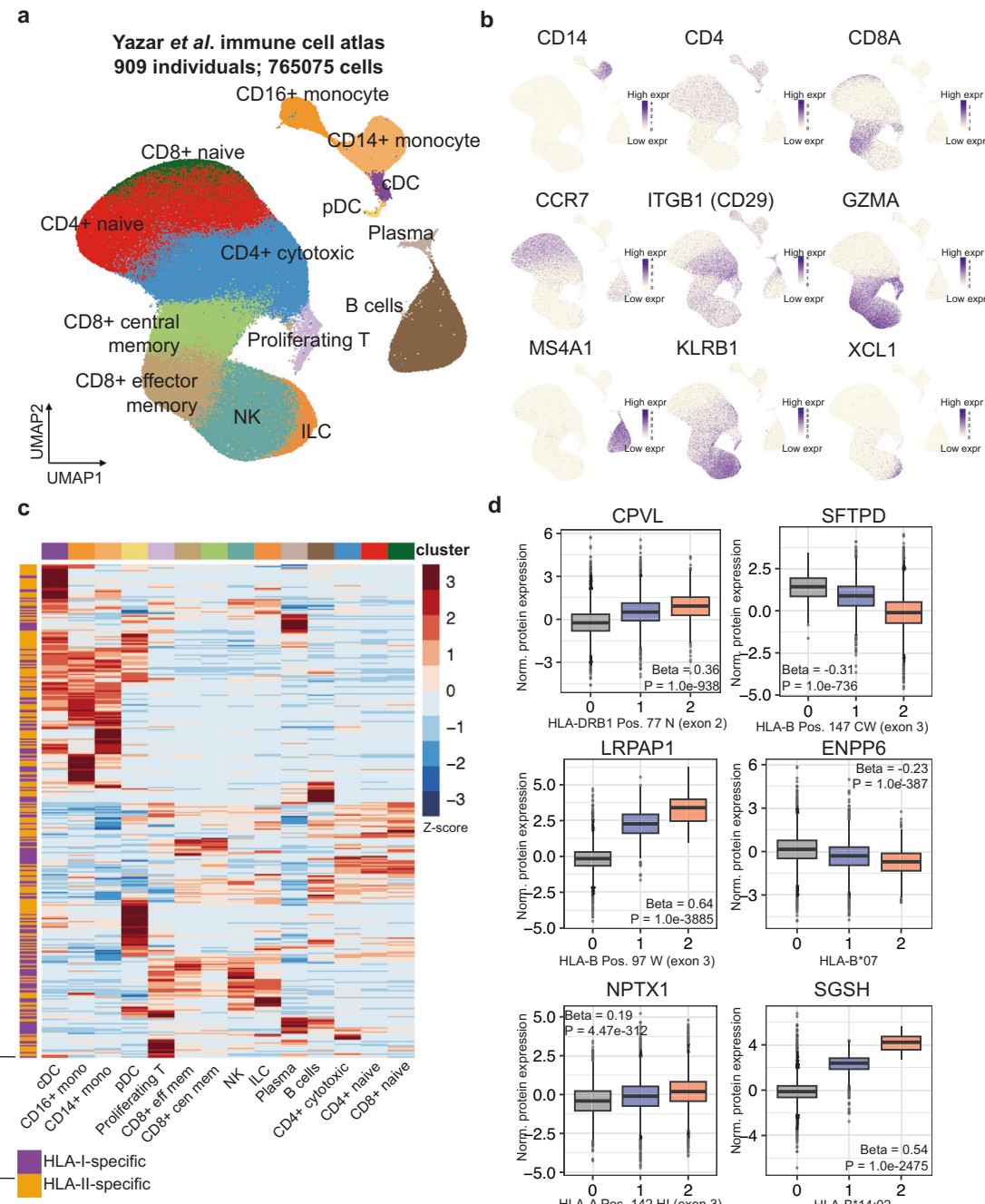

**Fig. 3 | Expression of HLA-pGenes on immune cells from the Yazar et al. atlas.**
**a** UMAP depicting cell types from the Yazar et al. immune cell atlas **b** Selected marker genes for the clusters named in (**a**). **c** (**a**). **d** Selected pGenes from among the top 50 lead HLA-pQTL. These proteins were selected due to their strong associations with HLA genetic variation and unclear or undefined roles in the immune response. Boxplots show minimum, maximum, interquartile range, and

outliers. *P* values are unadjusted, two-sided, and determined via *t* test and linear regression analysis. The *p* value threshold is after Bonferroni correction for the total number of proteins test ($5 \times 10^{-8}/2940$ proteins tested $= P \leq 1.70 \times 10^{-11}$). For all boxplots, the sample size used for analysis was $N = 34{,}490$ individuals, e.g., the number of individuals in the UKB-PPP discovery cohort.

LD of the HLA region. In addition, a major assumption of HLA-trait association studies is that coding variation drives the majority of the genetic heritability. Moreover, most colocalization methods assume a single causal variant; while some methods do allow for more than one causal variant, such methods have not been tested or rigorously benchmarked on the MHC.

Despite these limitations, we obtained summary statistics for fine-mapped HLA-disease summary statistics from the UK Biobank from Sakaue et al.[11], and overlapped them with our fine-mapped HLA-pQTL (Fig. 4; Supplementary Data 20). Specifically, Sakaue et al.[11] performed

single-marker tests with imputed HLA variants; thus, we overlapped all of our single-marker lead and conditional HLA-pQTL with the single-marker summary statistics from Sakaue et al.[11]. We reasoned that since this analysis represents an overlap between two sets of fine-mapped summary statistics, it effectively amounts to a colocalization between HLA-pQTL and HLA-disease signals. Notably, the analysis by Sakaue et al.[11] analyzed HLA associations with both diseases and quantitative traits (e.g., HDL cholesterol) in the UK Biobank.

The integration of HLA-pQTL and HLA-disease summary statistics allows us to visualize novel links between fine-mapped HLA genetic

**Table 1 | HLA-pQTL involving alleles associated with risk of immune-mediated diseases**

| HLA risk allele | Disease | Protein | β | P value |
|---|---|---|---|---|
| B*08:01 | Seronegative RA | MICB_MICA | 0.34 | 1.0e − 903 |
| B*08:01 | Seronegative RA | B2M | 0.13 | 1.24e − 119 |
| B*08:01 | Seronegative RA | BTN2A1 | 0.12 | 1.54e − 73 |
| B*08:01 | Seronegative RA | SIGLEC6 | 0.08 | 1.26e − 43 |
| B*08:01 | Seronegative RA | HBEGF | 0.07 | 7.70e − 42 |
| B*27:05 | Ankylosing spondylitis | GP1BB | −0.15 | 1.06e − 171 |
| B*27:05 | Ankylosing spondylitis | MICB_MICA | −0.05 | 2.04e − 70 |
| B*27:05 | Ankylosing spondylitis | KLRD1 | 0.05 | 2.12e − 22 |
| B*27:05 | Ankylosing spondylitis | AIF1 | −0.04 | 8.11e − 15 |
| B*27:05 | Ankylosing spondylitis | TNF | 0.03 | 3.34e − 09 |
| C*06 | Psoriasis | MICB_MICA | 0.28 | 1.0e − 466 |
| C*06 | Psoriasis | CD8A | 0.22 | 1.0e − 364 |
| C*06 | Psoriasis | DDR1 | 0.14 | 4.28e − 142 |
| C*06:02 | Psoriasis | IL7R | 0.10 | 8.21e − 79 |
| C*06:02 | Psoriasis | TXDNC15 | −0.09 | 7.10e − 63 |
| DQB1*02:01 | Celiac disease | AIF1 | 0.05 | 5.51e − 21 |
| DQB1*02:01 | Celiac disease | TNFRSF13C | 0.05 | 8.59e − 17 |
| DQB1*02:01 | Celiac disease | IKZF2 | −0.04 | 1.57e − 14 |
| DQB1*02:01 | Celiac disease | PGR | −0.04 | 4.77e − 14 |
| DQB1*02:01 | Celiac disease | FAS | 0.03 | 3.39e − 10 |
| DRB1*01:03 | IBD | LTA | −0.09 | 2.12e − 70 |
| DRB1*01:03 | IBD | IL10 | 0.09 | 2.50e − 62 |
| DRB1*01:03 | IBD | CCL19 | 0.09 | 2.07e − 58 |
| DRB1*01:03 | IBD | KLRD1 | −0.06 | 3.95e − 25 |
| DRB1*01:03 | IBD | GZMH | −0.05 | 2.19e − 21 |
| DRB1*03:01 | Lupus | CPVL | 0.35 | 1.0e − 934 |
| DRB1*03:01 | Lupus | HAVCR2 | −0.09 | 6.02e − 66 |
| DRB1*03:01 | Lupus | PLA2G15 | 0.10 | 1.0e − 70 |
| DRB1*03:01 | Lupus | AMBP | −0.08 | 1.01e − 51 |
| DRB1*03:01 | Lupus | APOM | 0.08 | 2.08e − 44 |
| DRB1*04:01 | T1D/RA | EFCAB14 | −0.09 | 1.47e − 62 |
| DRB1*04:01 | T1D/RA | NFATC3 | 0.08 | 5.49e − 45 |
| DRB1*04:01 | T1D/RA | MANSC1 | −0.06 | 8.41e − 26 |
| DRB1*04:01 | T1D/RA | IFNL1 | −0.06 | 4.68e − 25 |
| DRB1*04:01 | T1D/RA | CXCL3 | −0.05 | 6.01e − 20 |
| DRB1*04:04 | RA | LILRB4 | 0.27 | 1.0e − 725 |
| DRB1*04:04 | RA | CD74 | 0.18 | 3.31e − 238 |
| DRB1*04:04 | RA | SDK2 | 0.07 | 1.57e − 40 |
| DRB1*04:04 | RA | FABP6 | 0.06 | 4.17e − 29 |
| DRB1*04:04 | RA | PDZK1 | 0.07 | 3.87e − 28 |
| DRB1*15:01 | MS | LILRB4 | 0.49 | 1.0e − 1957 |
| DRB1*15:01 | MS | CD22 | 0.18 | 1.41e − 253 |
| DRB1*15:01 | MS | C2 | −0.15 | 1.33e − 183 |
| DRB1*15:01 | MS | CD72 | 0.12 | 4.52e − 111 |
| DRB1*15:01 | MS | TIMD4 | 0.10 | 5.81e − 69 |

Top associations for each allele associated with the indicated disease are shown. The full results with all significant associations are shown in Supplementary Data 1 and 19.

P values are unadjusted, two-sided and determined via t-test and linear regression analysis. The p value threshold is after Bonferroni correction for the total number of proteins test ($5 \times 10^{-8}/2940$ proteins tested = $P \leq 1.70 \times 10^{-11}$).

RA rheumatoid arthritis, IBD inflammatory bowel disease, T1D type 1 diabetes, MS multiple sclerosis.

variants, proteins, diseases, and traits. For example, this analysis reveals that HLA-B position 180_Q affects both CLEC4A, a C-type lectin receptor important for the innate immune response, and HDL cholesterol. The DRB1*01:03 allele—which is thought to be a risk allele for ulcerative colitis (UC) based on prior GWAS[39]—was associated with both UC and the expression of several inflammatory proteins such GZMA, GZMH, and IL10. Mechanistically, as the GZM genes are thought to be expressed on gut T and NK cells, our data may suggest that the HLA risk allele DRB1*01:03 modulates T and NK function to affect the risk of UC. Moreover, the IL10 gene, which is also affected by DRB1*01:03, has long been considered a potential drug target for immune-mediated diseases[40]. Thus, our data suggest that targeting IL10 may be particularly viable in individuals with the DRB1*01:03 UC risk allele.

To further explore the potential therapeutic implications of our HLA-pQTL, we obtained data from the druggable genome[41], which combines multiple lines of evidence to separate genes into tiers based on their potential druggability. In particular, Tier 1 genes are the targets of approved drugs or clinical-phase drug candidates obtained from the ChEMBL database. Using the Olink panel genes as background as we did previously for pathway enrichments, we asked whether our HLA-pGenes are enriched in druggable genes[41]. Indeed this was the case, regardless of whether we considered all tiers (e.g., including lower tiers defined by various molecular evidence for the target) or only Tier 1 approved targets (Supplementary Fig. 8b).

Among the overlap between our HLA-pGenes and Tier 1 targets were IL10, CD80/CD86 (targets of abatacept, used for the treatment of RA), and the well-known cancer tyrosine kinase MERTK (Supplementary Data 21). Thus, our analysis may motivate considering the use of drugs targeting these genes, especially in individuals with predisposing HLA alleles. Further mechanistic hypotheses enabled by the integration of our HLA-pQTL with HLA-disease data are the effect of HLA-B position 114 DK on the CD1C gene and rheumatoid arthritis. These data may implicate dendritic cells as the primary APC in the pathogenesis of RA, as CD1C is a canonical lineage marker for conventional dendritic cells.

Finally, we examined cis-pQTLs of the HLA-pGenes identified from the flagship pQTL study, and found that the cis-pQTLs of five HLA-pGenes—CD5, CD6, IL7R, SLAMF8, and TNFSF11—are themselves associated with autoimmune diseases with HLA risk (Supplementary Fig. 10a). In particular, CD5, CD6, and IL7R were abundantly expressed on T and myeloid cells in ref. 17 dataset (Supplementary Fig. 10b); IL7R in particular is a critical marker for both CD4+ and CD8+ T cells. Prior studies have implicated CD5 in T cell activation and Th17 cell differentiation[42,43]; similarly, CD6 is thought to have important roles in T cell development in the thymus and the generation of regulatory T cells[44].

To further annotate the disease relevance of proteins affected by HLA-pQTL, we asked which pGenes found in our analyses harbor coding variation associated with disease-relevant traits in the UK Biobank, including well-known biomarkers such as BMI, albumin, and urate. We found links between 57 HLA-pGenes and 38 traits (Supplementary Fig. 12). Intriguingly, we detected multiple corneal traits associated with HLA-pGenes, which may support the previously described roles of HLA in corneal immune and epithelial cells[45,46].

## Discussion

Altogether, our data suggest that HLA risk alleles exert disease-relevant trans effects on plasma protein expression. We hypothesize that there may be two primary mechanisms by which HLA polymorphism affects protein expression. First, as our HLA-pQTL analyses—both single-marker tests and fine-mapping analyses—strongly implicate amino acid positions within the peptide binding groove, we suspect that variation in antigen presentation affects T and NK cell recognition of peptide-MHC, which in turn may modulate the expression levels of proteins related to T and NK effector function.

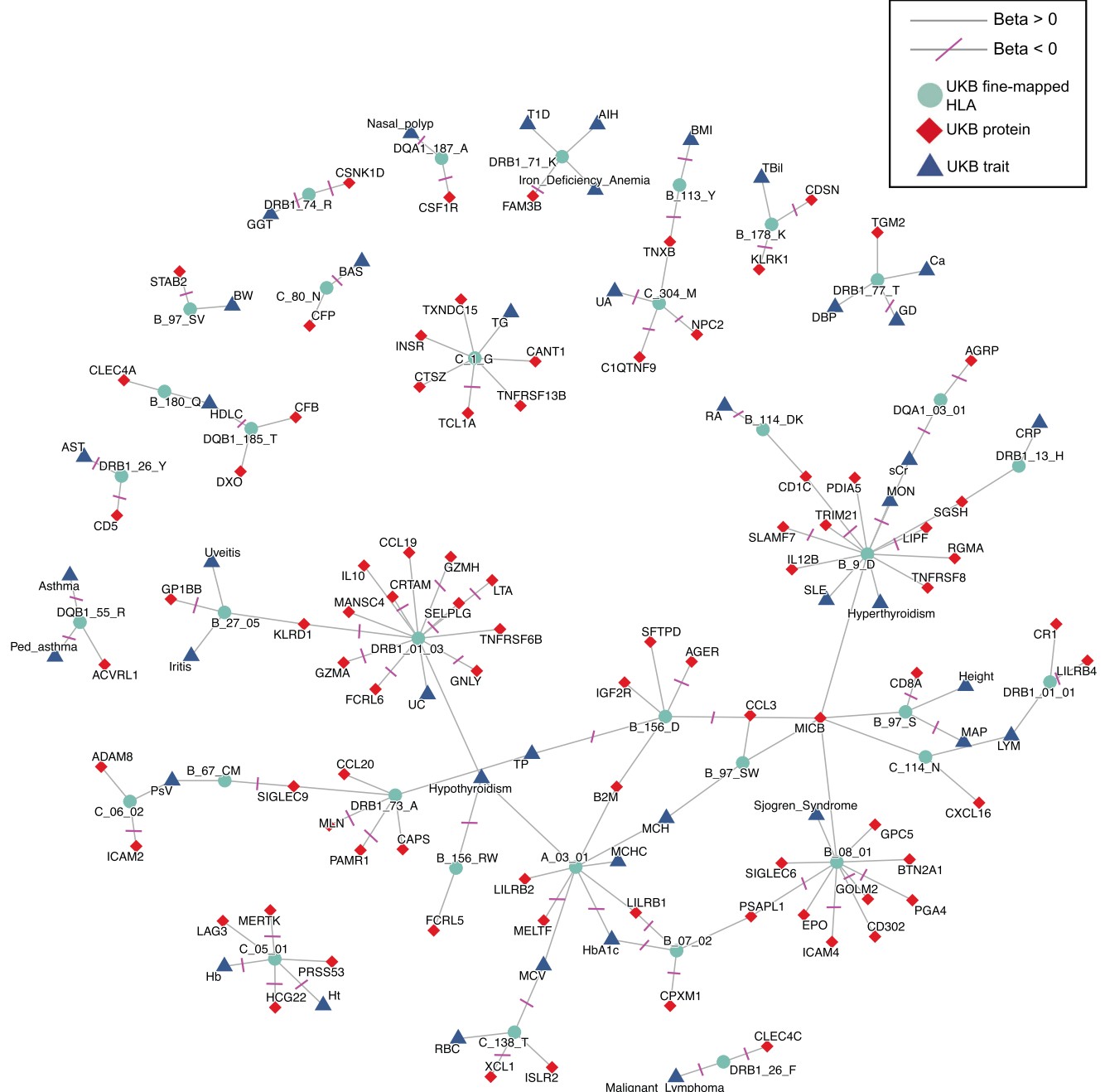

**Fig. 4 | Integration of fine-mapped HLA-pQTL with fine-mapped HLA-trait data from Sakaue et al.[11].** Abbreviations for diseases and biomarkers are provided in Supplementary Data 20.

Indeed, the expression of a wide array and chemokines and cytokines may be related to T and NK cell recognition of peptides presented by APCs[47]. The second potential mechanism by which such changes in protein expression may arise is through cell-intrinsic effects in macrophages and other APCs with high HLA expression. This could explain the discovery of proteins involved in antigen processing and presentation, macrophage activating, and inhibitory receptors.

We examined the transcriptional expression of HLA-pGenes in two independent scRNA-seq datasets, and observed enrichment of expression of HLA-pGenes in immune cells, and non-immune cells in tissues. Furthermore, the data suggest that while HLA-I and HLA-II genetic variations do affect distinct genes, the HLA-I-specific and HLA-II-specific pGenes largely converge on shared immune cell types. This finding may be surprising given the restricted expression of HLA-II to APCs[33], as

compared to HLA-I, which is expressed on all nucleated cells. These analyses strongly motivate future studies formally investigating the extent to which HLA genetic variation exerts *trans* effects on gene and/ or protein expression in particular cell types (e.g., single-cell HLA-pQTL), especially since several of the proteins affected by HLA genetic variation do not have an immediate connection to T cell or APC biology.

As discussed, standard fine-mapping and colocalization methods optimized for bi-allelic SNPs in other parts of the genome are not suitable for analysis in the MHC region. Taking advantage of previously fine-mapped HLA-disease associations, we integrated HLA-pQTL with HLA-trait associations to create a map of the potential effects of HLA variation on human disease through protein expression regulation. Such integration enables exploratory hypothesis generation regarding the mechanisms underlying HLA associations with disease and

potential drug targets. Indeed, the proteins shown in Fig. 4 may serve as candidates for future functional studies. Importantly, given the key differences in assumptions and methodology between genome-wide fine-mapping methods and HLA fine mapping, further benchmarking and developments in statistical methodology are warranted to perform formal fine mapping, colocalization, and Mendelian randomization analyses with HLA genetic variants.

Our study reports a wide range of variance in protein expression explained by HLA-pQTL. Given the *trans* SNP pQTL reported by the flagship Sun et al. study, an intriguing question is whether HLA effects on protein expression act in similar pathways to *trans* pQTL outside the HLA locus. We performed an exploratory analysis investigating the pathways enriched in the genes proximal (i.e., in the *cis*-window) to these non-HLA *trans* pQTL (Supplementary Fig. 11). Curiously, no pathways reached statistical significance after adjusting for multiple tests, and only one—natural killer cell lectin-like receptor binding—was related to the immune response. This analysis may suggest that both HLA genetic variation and non-immune-related genetic variation act independently to influence protein expression. However, we urge caution in overinterpretation of the results, as we cannot rule out the possibility that genes containing non-HLA *trans* pQTL are indirectly related to the immune response or the HLA genes themselves through currently unknown mechanisms.

A question that should be comprehensively addressed in the future as protein panels for pQTL mapping increase in size and scope is whether HLA-pGenes are more correlated with each other than by chance, e.g., compared to correlations between all proteins on the panel. In general, we observed weak correlations between most proteins in this analysis (median Spearman $\rho$ across all proteins: 0.05 (I.Q.R. 0.02–0.13; Supplementary Fig. 13a)). Furthermore, when we compared the distributions of correlations pairwise between HLA-pGenes and all pGenes, we found that statistically, correlations between the background set of all pGenes were higher than those between HLA-pGenes (Supplementary Fig. 13b). However, the difference between these two distributions was modest at best, and the observed statistical significance may simply reflect the large sample size in each distribution (e.g., pairwise correlations between hundreds of pGenes). In the present study we perform gene set enrichment and cell-type-specificity analyses arguing in favor of gene families or cell types affected by HLA genetic variation; however, there does not appear to be strong evidence for HLA-pQTL tagging correlated proteins. Whether the choice of protein expression profiling platform itself affects correlations between proteins[23,48] should also be investigated thoroughly in the future.

An important caveat of the current study is that only soluble proteins are measured, which may not reflect the fraction of proteins in physiologically relevant compartments. Prior studies have indicated that the soluble forms of at least a few of the proteins included in the Olink panel (e.g., PD1, LAG3) are cleaved directly from the cell surface through membrane shedding[49]. However, where possible, future studies should investigate the effects of HLA genetic variation on membrane-bound proteins.

Since existing methods for genetic colocalization and Mendelian randomization may not be nuanced enough for the MHC[10] given the high degree of LD, mechanistic functional studies may be necessary to validate the extent to which HLA-pQTL are truly causal for disease risk. Moreover, while our study did conduct preliminary replication analysis in individuals of non-European ancestry, the sample sizes of the non-European ancestry cohorts were much smaller than that of the European cohort. Thorough investigation of individuals of diverse genetic ancestry is sorely needed, both for HLA-pQTL analysis and more broadly. In addition, we emphasize the value of expanding HLA-pQTL analyses to specific cell types, tissues, and diseases pending data availability, as current efforts are limited to blood and a limited panel of proteins. More generally, we anticipate that our current study will pave the way for the investigation of HLA regulatory effects on quantitative traits beyond protein expression.

## Methods

### Study population
The UK Biobank is a prospective, population-scale cohort containing roughly 500,000 participants between the ages of 40–69 at enrollment. All individuals are assessed for demographic, clinical, and phenotypic variables at least once during a baseline visit; genome-wide imputed genetic data are available for all participants. For all analyses presented here, we used individuals from the UK Biobank that were included in the flagship UKB-PPP paper, which contains extensive details on participant selection and characteristics[14]. In brief, the UKB-PPP participants were split into a "randomized baseline" cohort of predominantly healthy individuals; all individuals were of European ancestry. The replication cohort consists of individuals manually selected by the participating pharmaceutical partners enriched for diseases of interest; individuals were of multiple ancestries in the replication cohort. To enable the replicability of HLA-pQTL across the discovery and replication cohorts and to confidently impute HLA variants across all individuals, we limited the replication cohort to individuals of European ancestry. In total, we analyzed 45,330 individuals from the UKB-PPP EUR cohorts, comprised of 34,490 individuals and 10,840 individuals in the discovery and replication cohorts, respectively. To assess the replicability of discovery EUR HLA-pQTL in individuals of diverse genetic ancestry, we obtained additional individuals from the flagship study of four distinct genetic ancestries: AFR ($N = 919$), CSA ($N = 914$), MID ($N = 302$), EAS ($N = 262$), testing imputed HLA variants at MAF > 1% and imputation dosage $R^2 > 0.7$ as performed in the EUR cohorts. The genetic ancestry labels were obtained directly from the flagship Sun et al. pQTL study; in that study, the genetic ancestry labels were also obtained directly from the UK Biobank (pan-UKBB; https://biobank.ctsu.ox.ac.uk/crystal/dset.cgi?id=2442).

### Plasma protein measurements and quality control
Full details on the measurement, processing, and quality control of Olink proteomic data for all UKB-PPP participants can be found in the Supplementary Information of the flagship UKB-PPP analysis. We followed the quality control steps on the protein expression data as the UKB-PPP flagship paper. The Olink technology is an antibody-based assay, with 2940 proteins measured across 8 panels (Cardiometabolic, Cardiometabolic II, Inflammation, Inflammation II, Neurology, Neurology II, Oncology, Oncology II), each focused on specific proteins selected by the UKB-PPP consortium partners across broad disease areas. Within each of the eight panels, the flagship study used principal component analysis (PCA) to remove individuals with PC1, PC2, or NPX (protein expression value $\log_2$ normalized relative to Olink assay controls) >5 standard deviations from the mean. The flagship study confirmed that there were no abnormalities in protein coefficients of variation; the percentage of variability attributable to batch and plate effects was also negligible. We used the same covariates for pQTL mapping as in the flagship study, namely age, age$^2$, sex, age × sex, age$^2$ × sex, batch, UKB center, UKB genetic array, time between blood sampling and measurement, and the top 20 principal components (PCs) of genetic ancestry. For each protein, we performed inverse rank normal transformation, regressed all covariates, and scaled the protein expression values to mean 0 and standard deviation 1 prior to pQTL mapping.

We used the protein expression values subject to all processing and quality control measures used in the flagship study (described below) for pQTL mapping in our study. The batch and plate of each sample and protein were recorded and used as input to pQTL modeling (see subsection below, "Covariates and linear models for HLA-pQTL mapping"). Furthermore, as performed in the flagship study, we subtracted REGENIE LOCO values from the residualized protein

expression values to account for local polygenic effects and sample relatedness[22]. For each protein, the LOCO values represent the overall contribution of SNPs on all chromosomes except for chromosome 6 to the expression of the protein of interest; thus, they represent polygenic effects that may confound the main genetic effect being tested. These values are known to more precisely account for population structure, sample relatedness, and LD[22]—indeed, according to the REGENIE paper, this approach avoids proximal contamination, which can result in reduced power of genetic association analyses. Critically, the REGENIE LOCO values were also used in the flagship pQTL paper[23] (subtracted from the protein expression residuals regressing all covariates); thus, we chose to include them in our study to remain consistent with the earlier study.

### Quality control of SNP genotypes from the UK Biobank and HLA imputation

We followed recently published guidelines[9] (https://github.com/immunogenomics/HLA_analyses_tutorial/blob/main/tutorial_HLAQCImputation.ipynb) for genotype QC and HLA imputation using the genome-wide imputed SNP array data from the UK Biobank as input. We used PLINK[50] to remove duplicated SNPs and used snpflip to correct reverse/forward strand flips. We then performed a first round of SNP QC to remove poor quality variants, i.e., those with high genotype call failure rate and violation of Hardy–Weinberg equilibrium (PLINK parameters –geno 0.1 –hwe 1e − 10). Next, we filtered samples with >3% missingness (--mind 0.03), based on examining the distribution of genotype missingness across the full cohort. We then removed individuals with genome-wide heterozygosity three standard deviations above the mean; samples with too much or too little heterozygosity may reflect contaminated DNA samples or inbreeding, respectively. Using PLINK[50], we next removed individuals with high genetic relatedness (IBD PI_HAT > 0.9) and performed genome-wide identity-by-state computation after removal of the MHC region (hg19 chr6: 28,000,000–34,000,000). In the next round of SNP QC, we first examined the distribution of SNP missingness and removed variants with more than 3% missingness (--geno 0.03). We then confirmed the accuracy of our quality control measures by comparing allele frequencies between the UK Biobank and 1000 Genomes Phase 3, and confirmed that allele frequencies were highly correlated both genome-wide and within the MHC region ($P < 2.2e − 16$, Spearman $\rho$ 0.92 for both). We then removed any SNPs with more than 20% difference in allele frequency between the UK Biobank and 1000 Genomes. In total, 7391 variants in the MHC passed QC and were used as input to HLA imputation. We performed HLA genotype imputation with Minimac4 as implemented in the Michigan Imputation Server, using the QC'd MHC SNP data from the UK Biobank as input. We used the most recent reference panel (Four-digit Multi-ethnic HLA reference panel v2) for imputation[9,12]. After imputation, we retained 2284 HLA variants at MAF ≥ 1% and imputation dosage $R^2 ≥ 0.7$, comprised of 74 one-field alleles, 93 two-field alleles, and 2117 amino acid polymorphisms.

### Covariates and linear models for HLA-pQTL mapping

For each of the 2940 proteins, we fit a multivariable linear regression model testing the protein expression (with normalization and adjustments described below), against the imputed dosage (continuous value between 0 and 2) for each of the 2284 HLA variants independently together with all covariates included in the flagship UKB-PPP study. Significance in the discovery cohort for lead HLA-pQTL was defined as $P ≤ 1.70 × 10^{-11}$ ($5 × 10^{-8}$/2940 proteins tested). *Trans* HLA-pQTL was defined as HLA-pQTL involving pGenes outside of the MHC region (chr6: 28,000,000–34,000,000). We used the same significance threshold regardless of whether the pGene was inside or outside the MHC region. For analyses combining the discovery and replication cohorts, we used a pooled analysis, directly combining individual-level data from the discovery and EUR replication cohorts,

and adding a covariate to the pQTL linear model indicating whether the individual was from the discovery or replication cohort. The significance threshold in this combined analysis was the same as in the discovery cohort. To control for additional sources of variation— including the presence of diseases in the cohort—we performed PCA on the normalized protein expression values across all individuals in the discovery cohort, performing mean imputation for individuals without particular protein values. Across all proteins, the mean proportion of individuals with missing values was 0.032 (median 0.03, I.Q.R. 0.02–0.04). In addition to the covariates included from the flagship UKB-PPP study, we also included the top 20 PCs of protein expression in the linear regression models. We repeated this procedure in the replication cohort. Linear models in the replication cohort included an additional covariate detailing whether the participant was pre-selected by the UKB-PPP consortium members or COVID imaging study.

### Evaluation of diseases in the UK Biobank captured by principal components of protein expression

The flagship UKB-PPP study computed associations between each of the 2940 proteins and the most common diseases (defined by ICD-10 codes) in the UK Biobank. We converted these associations to Z-scores, and performed Spearman correlation of these Z-scores with the protein loadings for the top 20 protein expression PCs.

### Single-marker tests

To identify lead and conditional single-marker HLA-pQTL, we tested each of the 2284 HLA variants (one-field alleles, two-field alleles, and amino acid polymorphisms) independently for association with protein expression levels using the linear regression models and covariates described above. We adopted a conservative threshold for the significance of lead pQTL of $P ≤ 1.70 × 10^{-11}$ determined by dividing the genome-wide significance threshold ($P ≤ 5 × 10^{-8}$) by the total number of proteins tested ($N = 2940$). For conditional analyses, we conditioned on the lead HLA variant and repeated the linear regression until no variant was significant at $P ≤ 5 × 10^{-8}$. To assess the replicability of lead HLA-pQTL, we performed these same analyses in the replication cohort, and checked the significance ($P < 0.05$) and direction of effect for all lead HLA-pQTL.

### Permutation analysis

To assess the robustness of HLA-pQTL in the discovery cohort, we performed 1000 permutations per protein randomizing protein expression values across donors on each iteration, and testing all HLA variants in each permutation. We retained the lead variant from each iteration to construct an empirical null distribution for each protein.

### Proportion of protein expression variance explained by HLA-pQTL vs genome-wide *trans* SNP pQTL

We wished to assess the proportion of *trans* genetic heritability explained by HLA genetic variation vs all other SNPs. Thus, we first used the anova() function in R to compute the variance explained by all lead and conditional HLA-pQTL. To compare the variance explained by HLA-pQTL to variance explained by all SNPs, we obtained total SNP heritability estimates directly from the flagship study[23]. In brief, the flagship study computed total heritability as the sum of variance explained by all independent SNP pQTL identified by SuSiE[51] and the remaining SNPs across the genome (polygenic component) calculated using LDSC[52].

### Sensitivity analysis accounting for potential effects of SNP pQTLs in the HLA class III region

To assess the robustness of our HLA-pQTL to potential effects from SNP pQTLs in the HLA class III region identified in the flagship study[23], we first checked whether any of the 504 HLA-pGenes identified in our analysis also had significant SNP pQTL associations from the class III

region from the flagship study. We identified 29 such genes that had a significant class III pQTL. For these 29 genes that have significant HLA-pQTL in our study and significant class III pQTL from the flagship study, we then investigated whether both the HLA-pQTL and class III pQTL independently contribute to plasma protein expression. Thus, for each of the 29 pGenes in the discovery cohort, we fit a combined pQTL model incorporating our lead HLA-pQTL, the significant class III SNP pQTL from the flagship study, and all clinical and demographic covariates. We then examined the significance of our lead HLA-pQTL from our analysis and the class III pQTL from the flagship study when both are placed in the same combined model. Significance was defined at $P < 5 \times 10^{-8}$.

### Sensitivity analysis removing individuals with HLA-associated diseases

To examine whether the HLA-pQTL from the discovery cohort were robust to the removal of individuals in the discovery cohort with diseases associated with HLA genetic variation in prior studies, we identified patients with diagnoses or deaths (ICD-10) due to the major autoimmune diseases previously linked to HLA genetic variation as depicted in Table 1 of the manuscript. We also excluded patients with any of the diseases associated with HLA genetic variation from the UK Biobank HLA analysis in Sakaue et al.[11] We detected 6833 individuals in the discovery cohort with confirmed ICD-10 codes matching any HLA-associated diseases. We thus excluded these 5021 individuals from the discovery cohort and re-tested the lead HLA-pQTL for each of the 504 pGenes. We examined the significance ($P < 5 \times 10^{-8}$) of our lead HLA-pQTL in this reduced discovery cohort, and concordance of effect sizes between the reduced cohort and the full, original discovery cohort.

### Conditional haplotype analysis

To determine whether groups of amino acids inherited together jointly influence variation in protein expression, we used phased best guess amino acid genotype information to conduct conditional haplotype tests. First, for each protein, we evaluated the association between each amino acid position and protein expression using the linear regression models and covariates described above. This first linear regression model includes all amino acid polymorphisms at the indicated position, created by grouping all two-field alleles for the particular locus into $M$ groups based on the amino acid at the tested position. To obtain an omnibus $P$ value for a particular amino acid position, we compared this first linear regression model to a null model without any of the $M$ groups, using the anova() function in R. Following identification of the amino acid position with minimal $P$ value ($P < 5.0e - 08$) from the first round of conditional haplotype analysis, we repeated this analysis sequentially conditioning on additional amino acid positions within the locus of the lead position until no position reached significance (i.e., if the lead position is in HLA-A, sequentially condition on other positions within HLA-A). Given the complexity of the DRB1–DQB1–DQA1 superlocus, for all proteins for which the lead position was within DRB1, DQB1, or DQA1, we tested all amino acid positions in DRB1, DQB1, and DQA1 when adjusting for the lead and additional independent variants, as performed previously[7]. We limited our conditional haplotype analyses to the top three conditionally independent amino acid positions per protein.

### HLA-I and HLA-II-specific pGenes and network analyses

For the initial assessment of proteins uniquely affected by HLA-I or HLA-II genetic variation, we assembled two sets of mutually exclusive pGenes—those with a lead variant in HLA-I (HLA-A, B, or C), and those with a lead variant in HLA-II (HLA-DRB1, DQB1, DQA1, DPB1, or DPA1). To identify HLA-I-specific pGenes, we sought to rigorously rule out any

confounding of HLA-II genetic variation for pGenes with an HLA-I lead variant (and likewise for pGenes with an HLA-II lead variant, i.e., rule out confounding by HLA-I). Thus, for each pGene, we tested the lead HLA variant adjusting for all two-field alleles of the other class (i.e., for a pGene with HLA-I lead pQTL, we included all two-field HLA-II alleles as covariates in the linear regression model). pGenes that remained significant ($P < 5 \times 10^{-8}$) after this analysis were used as input to gene ontology enrichment analyses, using the full Olink panel ($N = 2940$ proteins) as the background. Significant (FDR $P < 0.1$) gene ontology terms were used as input to the cnetplot() function in the clusterProfiler package[53].

### Enrichment of HLA-pGenes in genes specifically expressed in particular tissues

To evaluate the enrichment of HLA-pGenes in tissue-specific gene sets, we obtained lists of genes specifically expressed in each of 32 tissues from the Uhlen et al. study. Consistent with the Uhlen et al. study, we defined tissue-specific genes as those in the "enriched" (mRNA levels in a particular tissue at least five times those in all other tissues) or "enhanced" (mRNA levels in a particular tissue at least five times average levels in all tissues) groups. We evaluated the enrichment of our HLA-pGenes in genes specifically expressed in each tissue using a Fisher's exact test using the Olink panel genes ($N = 2940$) as background, and defined significance as FDR $P < 0.05$ correcting for multiple tests across all 32 tissues.

### Investigating transcriptional expression of HLA-pGenes across single-cell RNA-seq datasets

We sought to explore the expression of genes corresponding to our HLA-pGenes in various cell contexts using scRNA-seq data. We first examined the HLA-pGene expression in the Yazar et al. immune cell atlas, comprised of 909 individuals and 735,000 PBMC immune cells. We obtained count matrices for the four major immune lineages (B cell, T cell, myeloid, NK) from the Kang et al. study, which performed a rigorous re-analysis of the data removing all putative doublets. We then integrated these count matrices into a shared embedding using pre-processing measures and Seurat parameters specified by Kang et al.; as specified in that study, we used Harmony to correct for experimental batch ($\theta_{batch} = 2$). Importantly, the study by Kang et al. mapped cells from the Yazar et al. dataset to a shared embedding with other datasets; in our analysis, we analyzed the Yazar dataset alone. We assigned cluster names based on differentially expressed genes (computed using a Wilcoxon test) and manual inspection of markers as shown in Fig. 3b. We next sought to explore the expression patterns of HLA-pGenes on non-immune cells, and compare them to expression on immune cells. Given the enrichment of HLA-pGenes on lung-specific genes, we obtained a lung cell scRNA-seq atlas from Travaglini et al., and used it to plot HLA-I-specific and HLA-II-specific pGenes as we did with the immune cell atlas from Yazar et al. Importantly, we chose this dataset because it is comprised of normal lung unaffected by disease, and has excellent representation of both immune cells and non-immune cells. We used cluster assignments exactly as specified in the Travaglini et al. original study. In each dataset, we plotted the mean log-normalized expression of HLA-I-specific and HLA-II-specific pGenes across all clusters.

### Atlas of proteins affected by disease-associated HLA alleles

We manually searched the literature for major autoimmune diseases associated with HLA genetic variants—in particular, one-field and two-field alleles. To link our HLA-pQTL to these diseases, we first performed conditional analyses for each pGene using only one- and two-field alleles (i.e., without fine mapping) to identify significant ($P < 5 \times 10^{-8}$) HLA alleles associated with each pGene, and used these associations to populate Table 1.

## Integration of fine-mapped HLA-pQTL with fine-mapped HLA-trait signals in the UK Biobank

To more systematically examine the relevance of HLA-pQTL to disease, we obtained fine-mapped HLA-trait summary statistics from Sakaue et al.[11]. Sakaue et al.[11] performed HLA imputation and fine mapping in the full UK Biobank cohort (e.g., not the limited pQTL cohort used in our study) to obtain lead and conditional HLA variants associated with a wide array of disease and biomarkers. To approximate a colocalization between HLA-pQTL and HLA-trait associations, we looked for overlaps between our significant fine-mapped (single-marker lead and conditional) HLA-pQTL data with significant fine-mapped HLA-trait data, and present the overlapping signals in Fig. 4.

## Enrichment of HLA-pGenes in targets of approved drugs

To evaluate the enrichment of HLA-pGenes in targets of approved drugs, we obtained data from the druggable genome study by Finan et al.[41]. The study by Finan et al. defined druggable genes according to multiple lines of evidence ("Tiers"); Tier 1 genes include targets of approved small molecules, biotherapeutics, and clinical-phase drug candidates. We used a Fisher's exact test to overlap our HLA-pGenes with either all druggable genes ($N = 4479$) or only Tier 1 genes ($N = 1427$ genes), with significance defined as $P < 0.05$.

## Associations between *cis*-pQTLs of HLA-pGenes with immune-mediated diseases

We sought to explore whether *cis*-pQTLs of HLA-pGenes discovered in our study are themselves associated with immune-mediated diseases commonly thought to be associated with HLA genetic variation. We queried the flagship pQTL study for all *cis*-pQTLs of HLA-pGenes identified in our study and queried the GWAS catalog to find significant associations ($P < 5 \times 10^{-8}$) between these *cis*-pQTLs and immune-mediated diseases. We present the significant associations in Supplementary Fig. 11.

## Coding variation in HLA-pGenes associated with traits

To explore the association of coding variation in HLA-pGenes with traits, we retrieved data from the genebass browser[54], and searched for all HLA-pGenes with significant (SKAT-O $P < 2.5e - 07$) associations.

## Pairwise correlations of protein expression

To evaluate whether HLA-pGenes might simply be proteins that are strongly correlated with one another, we computed pairwise correlations of protein expression across all individuals and all 2940 proteins. We detected weak correlations between most proteins (median Spearman $\rho$ 0.05; I.Q.R. 0.02–0.13; Supplementary Fig. 13a).

## Reporting summary

Further information on research design is available in the Nature Portfolio Reporting Summary linked to this article.

## Data availability

The raw UKB-PPP data are available via application to the UK Biobank. The raw UK Biobank data are available under restricted access because the data involve individual-level genotypes; access can be obtained by submitting an application at https://www.ukbiobank.ac.uk/enable-your-research/apply-for-access. The summary statistics for all analyses generated in this study are available in the Supplementary Data files. The single-cell RNA-sequencing data used in this study are available at https://www.ncbi.nlm.nih.gov/geo/query/acc.cgi?acc=GSE196830 and https://www.synapse.org/#!Synapse:syn21041850.

## Code availability

Detailed code describing protocols used in the manuscript for HLA imputation and fine mapping can be found at https://github.com/immunogenomics/HLA_analyses_tutorial.

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

## Acknowledgements

The authors acknowledge Melissa Miller, Anders Malarstig, and the UKB-PPP for their contributions to the design of the UKB Olink dataset. The authors also acknowledge all members of the Systems Immunology Team at Pfizer for their helpful comments. This research has been conducted using the UK Biobank Resource under Application Numbers 65851 and 26041.

## Author contributions

C.K. and X.H. conceived the study. J.C. assisted with processing of the UKB-PPP cohort and pQTL mapping. S.S. advised on HLA imputation and fine mapping. J.B.K. provided scRNA-seq data and advised on scRNA-seq analyses. S.M.C. provided suggestions on scRNA-seq analysis. I.L., M.A.A., H.I.K., D.V.S., and D.Z. provided critical feedback on the manuscript. S.R. provided critical analysis suggestions and feedback on all analyses. X.H. supervised the analyses. C.K. and X.H. wrote the manuscript with input from all authors.

## Competing interests

C.K., J.C., I.L., M.A.A., H.I.K., S.M.C., D.V.S., D.Z., and X.H. are employees and/or stockholders of Pfizer. S.R. is a scientific advisor to Pfizer, Janssen, and Sonoma Biotherapeutics, a founder of Mestag Therapeutics, and a consultant for Abbvie and Sanofi. The remaining authors declare no competing interests.
