## [Peer Review File · Nature Communications]

The influence of HLA genetic variation on plasma protein expressionREVIEWER COMMENTS

Reviewer #1 (Remarks to the Author):

In this manuscript, Krishna and colleagues carry out systematic pQTL mapping of Olink proteomics data from the UK Biobank, focusing on genetic variation in the HLA region. The manuscript itself is written to highlight the discovery of trans-regulation of protein expression by HLA genetic variation, as well as downstream disease associations of the identified pQTL variants. The manuscript is generally well written and will provide a nice resource for downstream studies. In its current form, the manuscript does have general weaknesses pertaining to: (a) large amounts of results but inadequate adequate interpretation and synthesis of those results, (b) inadequate consideration of linkage disequilibrium (LD) as it pertains to fine-mapping of HLA pQTL signals, and (c) lack of detail in defining key concepts and terms within the paper such as cis- vs. trans-pQTL and pGenes. The point about considering LD in fine-mapping is of particular importance as downstream analyses in the manuscript also consider the identity, location and functional significance of the identified pQTL associations. We provide our detailed comments below.

Major:

1. Abstract: Define the term pQTL at first use.
2. Introduction: suggest to present a brief description of the study analyses in the last paragraph.
3. Figure 1: suggest to extend Fig 1a by adding follow-up analyses after pQTL mapping to make it more clear about the study design.
4. Results "Discovery of HLA-pQTL in the UK Biobank":
 - a. Please define the term "pGene" at first use in the text, including what statistical significance cutoff was used to identify the pGenes.
 - b. In the statement "we found that pGenes tagged by HLA-pQTL were less correlated with each other compared to pairs of all other proteins", while the difference in correlation is statistically significant, it does not appear from Extended Data Fig 3a,b that the difference is substantial and I am not convinced that the point that "HLA-pQTL do not simply tag correlated proteins" is not well-supported. The authors should remove this speculative point from this section and could consider presenting it as a Discussion point with some more development and/or supporting literature.
5. Results "Discovery of HLA-pQTL in the UK Biobank"/ Methods: Clarify distinction between cis- and trans-pQTL mapped to the HLA region, including what genomic coordinates were considered for pQTL mapping, and what window size was used to define each of cis- and trans-pQTL corresponding to the HLA region.
6. Results "Fine-mapping...": How does the approach to fine mapping account for LD? Could the authors quantify uncertainty in the identified pQTL signals using credible sets or another related approach? In the absence of formal fine mapping, it is difficult to tell whether the findings related to enrichment of pQTL within the peptide binding groove represent true enrichment vs. an artifact attributable to LD.
7. Results "Fine-mapping...": Please include some description of the Results from Figure 1e "HLA-B conditional haplotype analysis" within the main text. Current text is restricted to HLA-DRB1 (corresponding to Figure 1f).
8. Results "Fine-mapping...": Which figure or table supports the results noted in the text for CPVL? Additionally, please provide literature reference(s) for the association with risk of type 1 diabetes and rheumatoid arthritis for CPVL.
9. Results-Replication of lead HLA-pQTL: In describing "combined analysis of both discovery and replication cohorts"
 - a. Please clarify in the Methods what approach was used combined analysis – was it meta-analysis or pooled analysis?
 - b. What significance cutoff was used to identify pGenes from combined analysis?
10. Results "Replication of lead HLA-pQTL": Current recommendations for consideration of race/ancestry in genetic analysis suggest prioritizing pooled approaches over exclusion of specific ancestry groups (e.g. <https://www.sciencedirect.com/science/article/pii/S2666979X22000921>). While we understand the preference for reduced heterogeneity afforded by restricting replication to European ancestry, the authors may consider (a) additionally presenting pooled analyses incorporating other race/ancestry groups, or (b) providing additional information regarding the number of individuals represented from other race/ancestry groups and some brief consideration of whether it was worth examining ancestry-stratified replication at these sample sizes.
11. Results "Distinct trans regulatory effects": Please comment more on the interpretation of Fig. 2c-g to support the statement that "we observed effects on proteins with less well-defined functions and relevance to the immune response"? Additionally, the language in the statement should be clarified, as the concept of "proteins with less well-defined functions" is vague.

12. Results "Atlas...": Please clarify in the Methods the approach used to create the atlas of proteins affected by diseases
13. Discussion: The first paragraph of Discussion states that "our fine-mapping analyses strongly implicate amino acid positions within the peptide binding groove." However, I only see the peptide binding groove mentioned in the Results section labeled "Discovery of HLA-pQTL in the UK Biobank" and not in the section on fine-mapping. Please clarify.
14. Discussion: Overall, this section is very brief. It would be good to have more interpretation of key results and limitations, possibly including: (a) value of fine mapping of HLA-pQTL, (b) further consideration of disease associations for the HLA-pQTL, (c) limitations related to LD within the HLA region, and (d) limitations related to race/ancestry composition of the study.
15. Methods "Plasma protein quality control" and "Covariates and linear models for HLA-pQTL mapping":
 - a. Please clarify the relevant covariates when saying "For each protein, we performed..., regressed all covariates..."
 - b. Please clarify what are REGENIE LOCO values and define the abbreviation LOCO at first use in the text. Be consistent with the usage of term, you used "Regenie LOCO scheme" later in the Methods.
16. Methods "Conditional haplotype analysis":
 - a. What window size did you apply to select variants for conditional analysis?
 - b. Did you consider LD for the selection of variants for conditional analysis?

Minor:

1. Figure 1b: the conditional HLA-pQTL should also be presented in the legend
2. Figure 1d: please define the abbreviation "PBG" in the legend.
3. Results "Fine-mapping...": In the following sentence "These analyses revealed strong...", please clarify what is meant by "strong" association – is it based on p-value, effect size, or otherwise. If p-value, we suggest the wording "These analyses revealed highly statistically significant associations..."
4. Results "Fine-mapping...": Please define the abbreviation "APC" at first use in the text.
5. Results "Fine-mapping...": In the sentence "The unexpectedly strong associations between HLA...", please clarify why the associations were considered unexpectedly strong?
6. Supplementary Tables: Please add footnotes to define abbreviations and column labels.
 - a. Need to improve the resolution of Figs. 2a and 2b, they are not readable. Additionally, these figures provide too much information, and it is hard to tell the main message from these two panels.
 - b. Please add figure legends for panels f and g.
7. Table 1: Please define the disease abbreviations in a footnote
8. Methods "Plasma protein quality control..." and "Covariates and linear models for HLA-pQTL mapping": It is confusing of what protein value was used as input for pQTL. In the Results, you noted that "covariate-residualized inverse rank-normalized protein expression", but in the Methods you stated that "we subtracted REGENIE LOCO values from the residualized protein expression values to account for..." which is not consistent with the description of Results.
9. Methods "Quality control of SNP...":
 - a. Add literature reference for PLINK
 - b. What is the software/tool used for IBD and IBS?
 - c. Add reference for Minimac4 or Michigan Imputation Server.
10. Methods "Covariates and linear models for HLA-pQTL mapping": In the first sentence of this section, please clarify what was the outcome variable used in pQTL mapping.
11. Methods: Would help to re-order this section so that the sub-sections are presented in the same order as the Results.
12. Methods "Network analyses":
 - a. Need to clarify the association that was tested in the sentence "we tested the lead HLA variant adjusting for all two-field..."
 - b. Add reference for cluster Profiler package.
13. Methods "Coding variation...": Add reference for the genebase browser.

Reviewer #2 (Remarks to the Author):

Krishna et al. aim to determine the influence of HLA polymorphisms on plasma proteomic profiles. The major histocompatibility complex (MHC) region is characterized by its high polymorphism and

strong linkage disequilibrium (LD). While various autoimmune and other diseases are associated with this region, pinpointing the precise alleles and variants responsible has proven challenging. Imputation within this region has improved the granularity of potential links to HLA alleles, variants and amino acid changes.

The authors leverage the recently released UK Biobank proteomics (3k platform from OLINK) dataset encompassing approximately 50,000 individuals with proteomics and genomic data. Their primary goal was to gain insights into how specific polymorphism within HLA genes, including previously identified disease risk loci, influence plasma protein levels. They employ a systematic workflow involving extensive conditional analyses to identify potentially independent variants.

The authors tackle a highly complex and intricate region of the genome and make interesting observations. I have questions and concerns that arise in the context of this densely packed and complex region:

1. It would be valuable to provide a more comprehensive overview of the analyses conducted in the recent flagship GWAS paper in Nature. Specifically, what were the associations from this region previously reported?
2. Given the complexity of the MHC region, it is crucial to address whether the analysis accounts for potential effects from HLA Class III genes.
3. The authors use of Bonferroni correction for conditional analyses to identify independent polymorphism associations. However given the limited region being tested, should there not be a less stringent p value. Could the threshold used, influence the number of truly "distinct" signals?
4. While it is not entirely surprising to find immune-related proteins associated with the region, the manuscript could delve deeper into how this information can be leveraged to inform autoimmune disease pathology. Are there insights into the mechanistic links between genetic variations, protein expression, and autoimmune diseases? Can the study provide potential therapeutic targets? Are cis pQTLs of genes identified linked in a more mechanistic manner to diseases associated with HLA alleles?
5. Is there expression data that support the observed associations between genetic variants and plasma protein levels?
6. The manuscript could explore the potential for colocalization analysis, acknowledging any limitations in the context of this challenging region. This approach may offer additional insights into shared causal variants between disease risk associations and protein levels.

Reviewer #3 (Remarks to the Author):

The manuscript titled "Exploring the Impact of HLA Genetic Variation on Plasma Protein Expression" investigates the influence of HLA genetic variation on the expression of 2940 plasma proteins by employing imputed HLA variants in a cohort of 45,330 Europeans from the UK Biobank. The study, which generates an atlas of HLA-pQTL, is intriguing; however, the presented level of novelty does not meet the publication standards of Nature Communications. I have several major comments for the authors to consider:

1. The manuscript reports that HLA-pQTL accounts for over 20% of the protein expression variance. It would be valuable to understand how much variance is attributed to non-HLA pQTL, and whether these non-HLA variants operate in the same pathways as their corresponding HLA genes.
2. Considering the substantial role of HLA in immune-mediated diseases, it is important to know the number of samples in the cohort with autoimmune or autoinflammatory diseases. If a significant portion of samples exhibit such conditions, the authors should address whether excluding these samples would impact the results.
3. While the authors adjust for HLA-DRB1 and observe that the majority of HLA-I lead pQTL remains significant, the rationale for selecting HLA-DRB1 is not explicitly justified. Although its variants are associated with the largest number of proteins, it is unclear how this choice is supported.

Response to Reviewers' Comments

Research article for Nature Communications ID: NCOMMS-23-38578-T

“The influence of HLA genetic variation on plasma protein expression”
Krishna *et al.*

We thank the editor and reviewers for their positive and constructive reading of our manuscript. The comments provided by the editor and reviewers have inspired us to refine and expand the manuscript with additional analyses that together speak to the robustness and potential mechanisms implicated by our findings. Below we provide a summary of the most important new data provided in response to the reviewers:

- 1) Clarification and explanation regarding the assumptions of MHC fine-mapping compared to traditional fine-mapping studies, and consistency of the approaches used in our study compared to prior HLA association studies. Specifically, as described in previous studies, we outline why traditional approaches to fine-mapping and colocalization are not appropriate for the MHC.
- 2) We integrate summary statistics from a recently published HLA-disease association study in the UK Biobank (Sakaue *et al.* Nature Genetics 2021) with our fine-mapped HLA-pQTL to extend and deepen our atlas of potential HLA-pQTL effects on disease. Since this analysis combines fine-mapped summary statistics from the prior HLA-disease and current HLA-pQTL analyses, it effectively amounts to a colocalization between the two analyses.
- 3) We use the integrated map of HLA-pQTL-disease fine-mapping to make hypotheses and add discussion about the potential drug targets and mechanistic effects of HLA-pQTL. Intriguingly, we show using data from ChEMBL that HLA-pGenes are statistically enriched in approved drug targets.
- 4) We use tissue-specific gene sets determined from RNA-seq analysis of 32 tissues from Uhlen *et al.* Science 2015 to show that at the transcriptional level, HLA-pGenes are preferentially expressed in immunogenic tissues such as spleen, lymph node, and lung.
- 5) Furthermore, we use scRNA-seq of peripheral blood immune cells from 909 individuals from Yazar *et al.* Science 2022 to show that HLA-pGenes exhibit strong cell type-specificity, and that HLA-I-specific and HLA-II-specific pGenes are expressed on shared cell types.
- 6) We wrote in our original manuscript that HLA-pGenes are mostly related to the immune response; upon closer scrutiny, indeed several seemed not related to immunity- to support this observation, we obtained an additional scRNA-seq lung cell atlas from Travaglini *et al.* Nature 2021 to show that a subset of HLA-pGenes are also expressed on non-immune cells.
- 7) As a further extension of the gene expression analyses, we condition our HLA-pQTL models on cis-eQTLs for the HLA-pGenes obtained from the GTEx

whole blood cohort. Intriguingly, this analysis showed that a) our HLA allele associations with the HLA-pGenes remained significant after conditioning on GTEx whole blood cis-eQTLs for the pGenes, indicating no confounding of HLA allele associations by cis-eQTLs for the pGenes, and b) that the cis-eQTLs also remained significant after conditioning on our HLA allele associations. These data suggest that cis-eQTLs of HLA-pGenes are also cis-pQTLs for those genes. Altogether, this analysis suggests that for the pGenes identified in our study, both HLA alleles and cis-eQTLs of those pGenes from GTEx are independently associated with protein expression levels in the UK Biobank.

- 8) We conducted two additional sensitivity analyses- in the first, we conditioned on pQTLs from the HLA class III region; in the second, we removed all individuals from the dataset with confirmed ICD9/10 diagnoses of autoimmune/inflammatory diseases and individuals with any other diseases associated with HLA genetic variation from Sakaue *et al.* Nature Genetics 2021, and found that the HLA-pQTL associations from our original manuscript remained significant in both analyses.
- 9) We conducted HLA-pQTL subgroup analyses in individuals of non-European ancestry. Notably, despite much smaller sample sizes in these subgroups, we still detected *trans* HLA-pQTL associations overlapping with the European subgroup, with high effect size concordance.

Our manuscript has two principal contributions: (1) the finding that HLA genetic variation—specifically, rigorously imputed HLA alleles and amino acid variants—are associated with the expression of many proteins outside the MHC region (i.e. *trans* HLA-pQTL) in a cohort of unprecedented size and scope. (2) In contrast to the flagship UK Biobank pQTL study (Sun *et al.* Nature 2023) and many prior pQTL studies, our manuscript represents a specific, comprehensive investigation of the effects of HLA allelic variation on plasma protein expression with multiple genomic follow-up analyses to link HLA-pQTL to pathways, cell types, and mechanisms underlying HLA associations with disease.

Altogether, we believe that the analyses suggested by the reviewers and the results described above have greatly expanded and enhanced our manuscript and provided additional confidence in the main results.

Below we describe these points in more detail, providing a point-by-point response to the reviewers in blue. New additions to the revised manuscript are highlighted in yellow throughout.

Overall, we are grateful that the editor and reviewers provided a positive assessment of our study, and we hope that the new additions described below address their questions comprehensively.

Point-by-point response to reviewers' comments:

Reviewer 1:

Reviewer #1 (Remarks to the Author):

In this manuscript, Krishna and colleagues carry out systematic pQTL mapping of Olink proteomics data from the UK Biobank, focusing on genetic variation in the HLA region. The manuscript itself is written to highlight the discovery of trans-regulation of protein expression by HLA genetic variation, as well as downstream disease associations of the identified pQTL variants. The manuscript is generally well written and will provide a nice resource for downstream studies. In its current form, the manuscript does have general weaknesses pertaining to: (a) large amounts of results but inadequate adequate interpretation and synthesis of those results, (b) inadequate consideration of linkage disequilibrium (LD) as it pertains to fine-mapping of HLA pQTL signals, and (c) lack of detail in defining key concepts and terms within the paper such as cis- vs. trans-pQTL and pGenes. The point about considering LD in fine-mapping is of particular importance as downstream analyses in the manuscript also consider the identity, location and functional significance of the identified pQTL associations. We provide our detailed comments below.

We thank the reviewer for their constructive assessment of our manuscript, and for the rigorous questions raised in the revision. In our revised manuscript and in responses to the reviewer, we clarify all terms used throughout the manuscript, and explain in more detail the HLA fine-mapping procedures and points regarding LD (including a new analysis demonstrating that the lead and conditional variants for each pGene are not in strong LD, suggesting that they are truly independent variants instead of “tagged” effects in high LD). To address the reviewer’s point regarding synthesis of the results, we have added multiple genomic follow-up analyses including analysis of two scRNA-seq datasets—one in peripheral blood immune cells, and one in both immune/non-immune cells from the lung—to show and discuss the transcriptional expression patterns of HLA-pGenes identified in our study. We further perform HLA-pQTL analyses in additional ancestry groups, and overlap our fine-mapped HLA-pQTL data with fine-mapped UK Biobank HLA-disease data from Sakaue *et al*, to create an extensive map of potential HLA-pQTL effects on disease.

We thank the reviewer again for their questions and present a point-by-point response to their questions below:

Major:

1. Abstract: Define the term pQTL at first use.

Response:

We thank the reviewer for suggesting this clarification in the abstract. In our original manuscript we wrote the following two sentences:

“We detected 504 proteins (17.1% of all proteins tested) affected by HLA genetic variation (HLA-pQTL), including widespread trans regulation of protein expression by autoimmune disease risk alleles. HLA-pQTL were enriched in gene families related to antigen presentation...”

Per the reviewer’s suggestion, we have amended the first sentence slightly and added a subsequent sentence to define ‘pQTL’, before the sentence about enrichment of HLA-pQTL in gene families:

“We detected 504 proteins (17.1% of all proteins tested) affected by HLA genetic variation, including widespread trans regulation of protein expression by autoimmune disease risk alleles. We term the HLA genetic variants that plasma protein levels ‘HLA-pQTL’, and affected proteins ‘pGenes’. Proteins affected by HLA genetic variants were enriched in gene families related to antigen presentation...”

2. Introduction: suggest to present a brief description of the study analyses in the last paragraph.

Response:

We thank the reviewer for this suggestion and agree that the last paragraph of the introduction would benefit from a description of the study analyses.

Per the reviewer’s suggestion, we have now added to this paragraph as follows:

“To address this question, we took advantage of data from the UK Biobank, a massive prospective cohort that has facilitated critical insights into the genetic determinants of disease through genome-wide genotyping and health phenotype collection on all participants¹⁷. A recent flagship pQTL effort led by the UK Biobank Pharma Proteomics Project (UKB-PPP) quantified and performed rigorous quality control of 2940 plasma proteins levels measured with the Olink platform across 54,306 UK Biobank participants^{16,18}. We used these data as input to our study, to comprehensively investigate associations between HLA genetic variation—here defined as one-field alleles, two-field alleles, or amino acids—and plasma protein expression. Following systematic identification of lead HLA-pQTL, we performed conditional analyses and fine-mapping to broaden the scope of our discovered HLA-pQTL effects and identify associations specific to HLA class I (HLA-I) or HLA class II (HLA-II) alleles. We examined the transcriptional expression of genes corresponding to proteins affected by HLA genetic variants on two large single cell RNA-seq atlases—one of peripheral immune cells from 909 donors from Yazar et al., and a healthy lung atlas with 3 donors containing lung-resident immune and non-immune

cells from Travaglini *et al.* Finally, we integrated our fine-mapped HLA-pQTL with fine-mapped HLA-disease analyses in the UK Biobank from Sakaue *et al.* to create an atlas and of potential HLA-pQTL effects on disease, and hypotheses about the mechanisms underlying HLA genetic associations with disease.”

3. Figure 1: suggest to extend Fig 1a by adding follow-up analyses after pQTL mapping to make it more clear about the study design.

Response:

We agree with the reviewer that Fig. 1a would benefit from a description of the follow-up analyses after pQTL mapping—we have now amended Fig. 1a as depicted below to depict the myriad follow-up analyses, including subgroup pQTL mapping in non-European ancestries, examination of HLA-pGene expression on two independent scRNA-seq datasets, and integration of our HLA-pQTL fine-mapping with HLA-disease fine-mapping summary statistics from Sakaue *et al.* Nature Genetics 2021. These follow-up analyses are described in more detail in response to the reviewer’s subsequent questions.

Fig R1.1 (Fig. 1a of the revised manuscript): Schematic of the study. We used imputation of one and two-field HLA alleles and amino acid polymorphisms across 45,330 Europeans from the UKB Pharma Proteomics Project (UKB-PPP) together with plasma protein levels quantified using the OLINK platform as input to pQTL mapping for 2940 proteins. For each protein, we tested each HLA variant (allele or amino acid) independently in a multivariable linear model incorporating covariates as specified by the UKB-PPP. We performed replication studies in subgroups of non-European ancestry (African (AFR); Central/South Asian (CSA), Middle Eastern (MID), East Asian (EAS) and admixed American (AMR)). Following systematic pQTL mapping, conditional, and fine-mapping analyses, we examined the expression of HLA-pGenes on two independent single cell RNA-sequencing datasets, and integrated our HLA-pQTL fine-mapping with summary statistics of HLA-disease fine-mapping from Sakaue *et al.*

4. Results “Discovery of HLA-pQTL in the UK Biobank”:

a. Please define the term “pGene” at first use in the text, including what statistical significance cutoff was used to identify the pGenes.

b. In the statement “we found that pGenes tagged by HLA-pQTL were less correlated with each other compared to pairs of all other proteins”, while the difference in correlation is statistically significant, it does not appear from Extended Data Fig 3a,b that the difference is substantial and I am not convinced that the point that “HLA-pQTL do not simply tag correlated proteins” is not well-supported. The authors should remove this speculative point from this section and could consider presenting it as a Discussion point with some more development and/or supporting literature.

Response:

We thank the reviewer for these suggestions and will address them together here. Regarding point a), we have now amended the results section “Discovery of HLA-pQTL in the UK Biobank” to define pGene at first use and the significance cutoff as follows:

“To fine map HLA-pQTL, we first tested the association between each imputed HLA variant and scaled, covariate-residualized inverse rank-normalized protein expression with REGENIE LOCO (leave one chromosome out) values subtracted (Methods) using linear regression models. The LOCO values are known to more precisely account for population structure, sample relatedness, and LD—indeed, according to the original REGENIE and flagship pQTL studies, this approach performs these corrections while avoiding proximal contamination, which can result in reduced power of genetic association analyses^{1,2}. We termed proteins whose expression was affected by an HLA genetic variant as “pGenes” or “HLA-pGenes”. We included demographic and technical covariates (Methods); additionally, to account for latent sources of variation in the data and the presence of diseases in each cohort, we included the top 20 protein expression PCs as covariates (Extended Data Fig. 1). In order for a pGene to be considered statistically significant in our analysis, we set a stringent Bonferroni significance threshold of $P = 1.70 \times 10^{-11}$ ($5 \times 10^{-8} / 2940$ proteins tested).”

The reviewer raises an important point regarding our statement in the original manuscript that “HLA-pQTL do not simply tag correlated proteins”. Upon re-examination of the boxplot originally included in Extended Data Fig. 3b, we agree with the reviewer that the difference between the correlations between all proteins (second box) and correlations between HLA-pGenes (first box) is modest at best, though statistically significant (i.e., correlations between all proteins are greater than those between HLA-pGenes, suggesting that HLA-pGenes do not simply tag correlated proteins).

However, per the reviewer’s suggestion, in reality the statistical significance observed may actually simply reflect the large number of pairwise correlations performed; in addition, as discussed in the flagship pQTL paper by Sun *et al.* and

follow-up paper by Eldjarn *et al.*, choice of platform or assay may also affect correlations between proteins^{2,3}.

In agreement with the reviewer's suggestion to remove these data from the results section but also acknowledging the potential importance of investigating correlations between proteins, we have instead moved these plots to the end of the paper in Extended Data Figure 13, and have now strongly caveated the data by stating that there does not appear to be a strong difference between the two sets of correlations.

The revised text in the discussion now reads as follows:

"A question that should be comprehensively addressed in the future as protein panels for pQTL mapping increase in size and scope is whether HLA-pGenes are more correlated with each other than by chance, e.g. compared to correlations between all proteins on the panel. In general, we observed weak correlations between most proteins on the Olink panel analyzed here (Median Spearman rho across all proteins: 0.05 (I.Q.R. 0.02-0.13; Extended Data Fig. 13a). Furthermore, when we compared the distributions of correlations pairwise between HLA-pGenes and all pGenes, we found that statistically, correlations between the background set of all pGenes were higher than those between HLA-pGenes (Extended Data Fig. 13b). However, the difference between these two distributions was modest at best and the observed statistical significance may simply reflect the large sample size in each distribution (e.g. pairwise correlations between hundreds of pGenes). In the present study we perform gene set enrichment and cell type-specificity analyses arguing in favor of gene families or cell types affected by HLA genetic variation; however, there does not appear to be strong evidence for HLA-pQTL tagging correlated proteins. Whether the choice of protein expression profiling platform itself affects correlations between proteins^{2,3} should also be investigated thoroughly in the future."

5. Results "Discovery of HLA-pQTL in the UK Biobank"/ Methods: Clarify distinction between cis- and trans-pQTL mapped to the HLA region, including what genomic coordinates were considered for pQTL mapping, and what window size was used to define each of cis- and trans-pQTL corresponding to the HLA region.

Response:

We thank the reviewer for this suggestion and certainly agree that the distinction between cis and trans-pQTL in our study should be clarified in the main text of the manuscript and the methods.

In the general SNP QTL literature, it is standard to define *trans* QTL as QTL more than 1MB away from the TSS of a gene; *cis* QTL are any QTL within this window. However, our analyses focus on pQTL mapping for HLA alleles rather than biallelic SNPs; furthermore, the MHC region is itself very gene-dense, and characterized by high LD.

For these reasons, instead of relying on the standard cis / trans definitions and nomenclature, we separated our HLA-pQTL into two classes as shown in Fig. 1c- "MHC region", for which the window is chr6: 28000000-34000000; and "trans",

defined in our study as any pGene outside of the MHC region in chr6: 28000000-34000000. While the window of the MHC region is much larger than the standard 1MB window used to define trans associations, we believe this more conservative classification of “MHC region” vs “trans (outside MHC region)” is more appropriate for our study—i.e., associations of HLA alleles with pGenes within the MHC region, even if more than 1MB away from an HLA locus itself, may not reflect a true *trans* association given the high LD in the MHC region.

Regarding the genomic coordinates used for pQTL mapping, we considered all 2940 proteins assayed in the flagship pQTL study—in other words, no pre-selection of proteins based on their genomic location was considered in our analysis; we simply tested all proteins.

We have now clarified these definitions in three key sections of the manuscript. First, in the main text at the first mention of *trans* HLA-pQTL- the revised text now reads as follows:

“Strikingly, the majority (480 / 504; 95.2%) of the HLA-pGenes were in trans (i.e. proteins outside the MHC region on chromosome 6) (Fig. 1b), suggesting that HLA genetic variation has widespread effects on protein expression. Importantly, here we defined trans HLA-pQTL as HLA-pQTL involving pGenes outside of the MHC region (hg 19 chr6: 28000000-34000000) due to the strong linkage disequilibrium within the MHC region.”

Secondly, we have amended the legend of Fig. 1c—instead of the prior “MHC” region and “trans” labels, we have amended the legend to now say “pGene in MHC region” and “pGene outside MHC region”.

Thirdly and finally, we have amended the methods section to clarify the definition of trans HLA-pQTL as follows:

“Significance in the discovery cohort for lead HLA-pQTL was defined as $P = 1.70 \times 10^{-11}$ ($5 \times 10^{-8} / 2940$ proteins tested). Trans HLA-pQTL were defined as HLA-pQTL involving pGenes outside of the MHC region (hg19 chr6: 28000000-34000000); in our analyses, the significance threshold was the same regardless of whether the pGene was inside or outside the MHC region.”

6. Results “Fine-mapping...”: How does the approach to fine mapping account for LD? Could the authors quantify uncertainty in the identified pQTL signals using credible sets or another related approach? In the absence of formal fine mapping, it is difficult to tell whether the findings related to enrichment of pQTL within the peptide binding groove represent true enrichment vs. an artifact attributable to LD.

Response:

We thank the reviewer for this question and the opportunity to clarify the core assumptions of MHC fine-mapping, and the key differences between MHC fine-mapping studies, which are consistent with recently published best practices towards

MHC fine mapping⁴, and standard/formal fine-mapping analyses, which use methods that are not currently tailored or appropriate for the MHC.

In general, the MHC region is characterized by extreme polymorphism and LD. In addition, a core assumption of HLA fine-mapping is that coding variants in the MHC (which affect antigen presentation; the central function of the MHC widely established in the immunology literature) drive the majority of the genetic heritability of the trait, which has been shown in many prior HLA association studies. Standard fine-mapping methods do not have these assumptions and have not been appropriately benchmarked on the MHC, which is why specialized, established methods to fine-map the MHC based on iterative conditional analyses and joint modeling of HLA alleles and amino acids have been developed to rigorously handle this region^{4–6}. More generally, standard genetic fine-mapping seeks to identify a small set of variants (usually bi-allelic SNPs) as credible sets—in HLA fine-mapping, the goal is to find specific amino acid positions that contribute to a causal haplotype. Understanding the effect of these amino acid positions is critical to HLA fine-mapping, which is why specialized methods such as the omnibus and conditional haplotype tests implemented in our manuscript have been extensively used to fine-map the MHC.

First, a core assumption of HLA fine-mapping studies is that coding variation drives associations between HLA genetic variation and the trait of interest, as demonstrated in many prior studies of HLA associations with autoimmune diseases^{6–14}. For example, work from our group and others have revealed that amino acid variation in the HLA-II peptide binding groove explains > 75% of the genetic heritability of rheumatoid arthritis⁷. This is a key reason why we and others seek to impute the 2-field HLA alleles and amino acids to fine map HLA associations, rather than simply using bi-allelic SNPs. Importantly, standard fine-mapping procedures are not built on the assumption that coding variation drives the genetic heritability of a trait; as exemplified in part by the enrichment of GWAS signals in non-coding regions of the genome.

Relatedly, MHC fine-mapping is distinct from standard fine-mapping in that the analysis motivated by the biology and core function of the MHC—that is, to present peptide antigens for recognitions by T cells. For this reason, we perform specific imputation of HLA alleles and amino acids, which correspond to coding variation within the MHC. Indeed, the core antigen presentation function of the MHC is carried out by the peptide binding groove of the MHC; as such, identifying amino acids within the peptide binding groove that mediate disease risk as in prior studies, or effects on protein expression as in our study, are of paramount importance. Altogether, the strong emphasis on coding variation—borne out quantitatively by prior studies investigating associations between HLA alleles/amino acids and disease risk, and by the biology of the MHC—renders standard fine-mapping methods inappropriate for the MHC, as they do not make this core assumption.

There exist several additional key differences between standard fine-mapping and MHC fine-mapping. First, many standard fine-mapping / credible set approaches assume a single causal variant—this is likely not the case for the MHC; neither for disease risk nor protein expression levels, as multiple alleles and amino acid

positions contribute to the heritability of the trait. Some recent fine-mapping and colocalization methods can handle multiple causal variants, but these methods have not been applied nor benchmarked on HLA, which is highly multi-allelic.

For these reasons, we feel that deep, quantitative investigation and benchmarking of standard fine-mapping approaches applied to the MHC is required and warrants dedicated future studies.

We certainly agree with the reviewer that it would be helpful to clarify how the MHC fine-mapping procedure accounts for LD. In HLA association studies such as ours or prior studies investigating HLA effects on disease, the imputation of the HLA genetic variants implies a hierarchical structure- the two field HLA allele nomenclature (e.g. DRB1*04:01) means that the allele is defined by a distinct amino acid sequence, both inside and outside of the peptide binding groove. In the MHC fine-mapping procedure, 1-field alleles (e.g. DRB1*04), 2-field (e.g. DRB1*04:01) alleles, and amino acid variants (e.g. DRB1 Pos. 74 GR) are each first tested separately, one at a time—then, we perform stepwise conditional analyses for each pGene conditioning on the lead variant (HLA imputed variant with strongest p-value), which would yield additional HLA variant independent of the lead variant.

To illustrate this point, for each pGene, we examined the pairwise LD R^2 between all HLA variants significantly associated with the pGene (e.g. the lead HLA variant for the pGene and all of its conditionally independent HLA variants from the single-marker tests). We performed this analysis for the 305 pGenes that had any significant variants after conditioning on the lead variant. Examining this distribution (represented as a histogram depicting the median R^2 across all conditional variants for each pGene), we found that despite the extreme LD of the MHC region, the pairwise R^2 between all variants associated with a particular pGene was very low. These data confirm that the reported peptide binding groove enrichments—and the number of conditionally independent variants reported in the manuscript in general—are not simply artifacts attributable to LD.

b

Distribution of median R^2 computed for all lead and conditional HLA variants for each pGene (N = 305 pGenes with conditional variants)

For each pGene, median R^2 across lead and all conditional HLA-pQTL: 1-field alleles, 2-field alleles, amino acids

Fig R1.2 (Extended Data fig. 3b of the revised manuscript): For each of the 305 pGenes with at least one conditionally independent HLA variant, we calculated the pairwise R^2 between all variants associated with the pGene. We then calculated the median of these pairwise R^2 values, and plotted the distribution of these median values across all 240 pGenes. The data show that in general, the pairwise R^2 between all lead and conditionally independent HLA variants associated with each pGene is very low (median 0.019), suggesting that the total number of conditionally independent variants reported in the manuscript, and the enrichment of HLA-pQTL in the peptide binding groove, are likely not an artifact attributable to LD.

These considerations also extend to the reviewer's question 16 regarding choice of variants for conditional analysis—as described above, we do not pre-select HLA variants for conditional analysis based on window size or LD; rather, we condition the model on the lead variant and test all other HLA variants.

We have added explanations of these points and the above histogram to the revised manuscript as follows. First, we add a paragraph explaining HLA fine-mapping to the beginning of the results section:

“...The final set of HLA variants included 2284 one-field alleles (e.g. DRB1*04), two-field alleles (e.g. DRB1*04:01), and amino acids (e.g. AA_DRB1_position13_exon2_FLS) in the classical HLA-I (HLA-A, B, and C) and II (HLA-DRB1, DQB1, DQA1, DPB1, DPA1) genes.

“We used these data to fine-map HLA-pQTL. Importantly, HLA fine-mapping is distinct from standard fine-mapping procedures—in HLA studies, it is assumed that coding variation is the primary driver of the heritability of the trait, which is not assumed by standard fine-mapping. Moreover, in contrast to standard fine-mapping, HLA fine-mapping is squarely related to the biology of the MHC—namely, the presentation of peptide antigens. Thus, HLA fine-mapping seeks to discover amino acid polymorphisms—either through single-marker conditional tests or conditional haplotype analysis jointly testing all amino acid variants at a particular position—that are associated with a trait, as these are the functional components of the MHC that are responsible for antigen presentation. These core assumptions of HLA fine-mapping are consistent with many prior studies which have investigated the effects of HLA genetic variation on disease risk.”

To fine map HLA-pQTL, we first tested the association between each imputed HLA variant...”

Next, we have amended the following section of the results to describe the histogram shown above and its implications:

Of the 504 pGenes, 304 had secondary independent associations or more (**Extended Data Fig. 3a**). Notably, despite high LD within the MHC region, the lead and conditionally independent variants demonstrated low pairwise R^2 (**Extended Data Fig. 3b**), suggesting that the conditionally independent HLA-pQTL reported here are not simply an artifact attributable to LD.

Finally, we have amended the discussion (also mentioned in the reviewer’s later question about expanding the discussion) to emphasize the need for dedicated future studies benchmarking and developing methods to perform more traditional fine-mapping in the MHC:

“Importantly, given the key differences in assumptions and methodology between standard fine-mapping methods and HLA fine-mapping, further benchmarking and developments in statistical methodology are warranted to perform formal fine-mapping, colocalization, and Mendelian randomization analyses with HLA genetic variants.”

7. Results “Fine-mapping...”: Please include some description of the Results from Figure 1e “HLA-B conditional haplotype analysis” within the main text. Current text is restricted to HLA-DRB1 (corresponding to Figure 1f).

Response:

We thank the reviewer for this suggestion and apologize for omitting discussion of Figure 1e from the main text. The results in Fig. 1b are especially important given the key role of MICA and MICB in inflammation. Per the reviewer’s suggestion we

have now added a discussion of this result to the main text as follows:

*“To pinpoint HLA amino acid positions that affect protein expression, we performed omnibus and conditional haplotype tests for all 2940 proteins (Methods, **Fig. 1e,f, Extended Data Fig. 4, and Supplementary Table 3**). These analyses revealed strong associations between multiple independent amino acid positions—themselves previously associated with risk of autoimmune diseases—with disease relevant proteins. For example, the most significant signal observed for conditional haplotype analyses at the HLA-B locus was for the MICB_MICA protein. MICB_MICA serves as a critical ligand for the NKG2D receptor expressed by NK and gamma-delta T cells, with prior reports suggesting a critical role for this protein in inflammation¹⁵ and anti-tumor immunity¹⁶. The most significant signal at the DRB1 locus was for the LILRB4 protein...”*

8. Results “Fine-mapping...”: Which figure or table supports the results noted in the text for CPVL? Additionally, please provide literature reference(s) for the association with risk of type 1 diabetes and rheumatoid arthritis for CPVL.

Response:

We apologize to the reviewer for the lack of clarity on this point. In our original manuscript, we wrote:

“For example, expression of CPVL, an enzyme with loosely defined roles in antigen processing²², was affected by HLA-DRB1 position 74 ($P = 1.0 \times 10^{-936}$), position 11 ($P = 6.10 \times 10^{-37}$), and position 71 ($P = 7.25 \times 10^{-13}$), all previously associated with risk of type 1 diabetes and rheumatoid arthritis.”

Our intention was to convey that amino positions 74, 11, and 71 in DRB1 have previously been found to be associated with either risk of type 1 diabetes⁷ and rheumatoid arthritis⁶ (position 71). To our knowledge, there is no evidence directly linking CPVL to type 1 diabetes risk.

We have thus re-written this sentence to be clearer as follows:

“For example, expression of CPVL, an enzyme with loosely defined roles in antigen processing²², was affected by HLA-DRB1 position 74 ($P = 1.0 \times 10^{-936}$), position 11 ($P = 6.10 \times 10^{-37}$), and position 71 ($P = 7.25 \times 10^{-13}$). These amino acid positions in DRB1 have previously been shown to be associated with risk of type 1 diabetes and rheumatoid arthritis, suggesting that HLA-DRB1 effects on CPVL and antigen presentation may influence type 1 diabetes and rheumatoid arthritis risk.”

9. Results-Replication of lead HLA-pQTL: In describing “combined analysis of both discovery and replication cohorts”

a. Please clarify in the Methods what approach was used combined analysis – was it meta-analysis or pooled analysis?

b. What significance cutoff was used to identify pGenes from combined analysis?

Response:

We thank the reviewer for these questions about the combined analysis and are happy to clarify. Since we have access to individual-level data from the UK Biobank discovery and replication cohorts, we performed pooled analysis, and the significance cutoff to identify pGenes was the exact same as in the discovery cohort, e.g. 1.70×10^{-11} ($5 \times 10^{-8} / 2940$ proteins tested).

We have now amended the methods section of the manuscript as follows:

“Significance in the discovery cohort for lead HLA-pQTL was defined as $P = 1.70 \times 10^{-11}$ ($5 \times 10^{-8} / 2940$ proteins tested). For analyses combining the discovery and replication cohorts, we used a pooled analysis, directly combining individual-level data from the discovery and EUR replication cohorts, and adding a covariate to the pQTL linear model indicating whether the individual was from the discovery or replication cohort. The significance threshold in this combined analysis was the same as in the discovery cohort.”

10. Results “Replication of lead HLA-pQTL”: Current recommendations for consideration of race/ancestry in genetic analysis suggest prioritizing pooled approaches over exclusion of specific ancestry groups (e.g. <https://www.sciencedirect.com/science/article/pii/S2666979X22000921>). While we understand the preference for reduced heterogeneity afforded by restricting replication to European ancestry, the authors may consider (a) additionally presenting pooled analyses incorporating other race/ancestry groups, or (b) providing additional information regarding the number of individuals represented from other race/ancestry groups and some brief consideration of whether it was worth examining ancestry-stratified replication at these sample sizes.

Response:

We thank the reviewer for this excellent suggestion—indeed, we originally limited our analyses in the replication cohort to the EUR ancestry group since the UK Biobank discovery cohort designed in the flagship pQTL paper was comprised only of EUR individuals. Moreover, the EUR group is by far the most powered in both the discovery (N = 34490 EUR individuals) and replication (N = 10840 EUR individuals).

However, we acknowledge the reviewer’s important suggestion to explore other genetic ancestry groups in the analysis. Per the flagship study, 5 additional genetic ancestry groups were analyzed- African (AFR), Central/South Asian (CSA), Middle Eastern (MID), East Asian (EAS), and admixed American (AMR). As the number of admixed Americans in the cohort was extremely low (N = 97) and the flagship study did not detect any significant pQTL at Bonferroni significance in this group, we proceeded to perform subgroup HLA-pQTL analyses (as performed in the flagship study) in the other 4 genetic ancestry groups- AFR N = 919, CSA N = 914, MID N = 308, EAS N = 262. We note that these additional ancestry groups were QC’d in exactly the same way as the original EUR group; we performed HLA imputation for these diverse groups with the same multi-ancestry reference panel in the Michigan Imputation Server used for the original EUR analysis. We note that we obtained the genetic ancestry labels (AFR, CSA, MID, EAS) directly from the flagship study, which

are the same genetic ancestry labels computed by the UK Biobank (pan-UKBB; <https://biobank.ctsu.ox.ac.uk/crystal/dset.cgi?id=2442>).

We used these ancestry-specific HLA-pQTL analyses to ask two questions. First, similar to the original EUR replication analyses, we asked how correlated the effect sizes were between the discovery cohort and each of the non-EUR groups. Secondly, we asked whether we could detect any ancestry-specific HLA-pQTL.

The data shown below suggest that, for pGenes detected in the discovery cohort and significant at $P < 0.05$ in each of the new groups analyzed, that effect sizes were strongly correlated. This result is consistent with our original EUR replication analyses. Intriguingly, at the stringent significance cutoff of $P < 1.7 \times 10^{-11}$ used in the original EUR discovery cohort, we detected one pGene (DPP7) that was detected in the AFR group but not the EUR group. While these results could suggest that there exist ancestry-specific HLA-pQTL, we strongly caution against overinterpretation given the very low sample size relative to the EUR group. Critically, more samples in non-EUR ancestry groups are urgently needed to validate the observed HLA-pQTL effects in our analyses.

Fig R1.3 (Extended Data fig. 5 of the revised manuscript): Concordance of effect sizes for HLA-pQTL that were nominally significant ($P < 0.05$) in additional genetic ancestry-specific subgroups.

We have now added a description of these results to the revised manuscript in the section discussing replication:

*“We observed high replicability of lead HLA- pQTL, both by significance and concordance of effect sizes (96% at $P < 0.05$ in the EUR replication cohort; Spearman $\rho = 0.94$ between discovery and replication; **Extended Data Fig. 3g** and **Supplementary Table 4**), and combined analysis of both the discovery and replication cohorts together yielded an additional 119 pGenes (623 total, **Supplementary Table 5**). Furthermore, we explored ancestry-specific HLA-pQTL analyses in four additional genetic ancestry groups-- African (AFR; $N = 919$), Central/South Asian (CSA; $N = 914$), Middle Eastern (MID; $N = 308$), East Asian (EAS; $N = 262$) (**Extended Data Fig. 5, Supplementary Tables 7-10**). As in our EUR replication analyses, we detected strong correlations of effect sizes between the discovery and non-EUR replication cohorts. Furthermore, at the stringent significance cutoff of $P < 1.7 \times 10^{-11}$ used in the original EUR discovery cohort, we detected one pGene (DPP7) that was detected in the AFR group but not the EUR group. While these results could suggest that there exist ancestry-specific HLA-pQTL, we strongly caution against overinterpretation given the very low sample size relative to the EUR group. Critically, more samples in non-EUR ancestry groups are urgently needed to validate the observed HLA-pQTL effects in our analyses.”*

11. Results “Distinct trans regulatory effects”: Please comment more on the interpretation of Fig. 2c-g to support the statement that “we observed effects on proteins with less well-defined functions and relevance to the immune response”? Additionally, the language in the statement should be clarified, as the concept of “proteins with less well-defined functions” is vague.

Response:

We thank the reviewer for the opportunity to clarify our statements regarding the relevance of HLA-pGenes to the immune response. Motivated by the reviewer’s question, we have first added clarification on this point to the main text as discussed below. Moreover, to comprehensively address the reviewer’s question, we will present new analyses include in the revised manuscript to explore transcriptional expression of genes corresponding HLA-pGenes in both immune and non-immune scRNA-seq data.

Our original statement that the proteins in Fig. 2c-g (now moved to Fig. 3d) in the revised manuscript) was based on literature review of some of the top pGenes across all of our analysis (in particular, the pGenes depicted below all had $P < 1e-300$; and thus were some of the strongest signals from our analysis—the select pGenes shown here were among the top 50 by p-value out of 504 pGenes detected in the analysis). Upon examining our pGenes in general, some of the strongest associations—e.g. CPVL, and SFTPD, have some prior documented connections to the immune response. For example, as mentioned in the original manuscript, CPVL

is an enzyme with loosely defined roles in antigen processing, which may implicate the gene in the adaptive immune response. SFTPD is generally thought of as a pattern recognition receptor, and thus might be considered a part of the innate immune response¹⁷. However, even this result is surprising and novel in the context of HLA genetics, which—through their central role in antigen presentation and recognition by T cells (the adaptive arm of the immune system)—might be thought to affect only genes of the adaptive immune response, rather than adaptive and innate.

The remaining four selected proteins now shown in Fig. 3d have no clear connection to the adaptive or innate immune response—and certainly not to HLA genetics—as far as we can ascertain from the literature. LRPAP1 is the LDL receptor-related protein 1; aside from a recent report suggesting that LRPAP1 may facilitate immune evasion from viruses¹⁸, it is both unclear and intriguing to speculate as to how LRPAP1 expression may be affected by HLA genetics. ENPP6 is an enzyme involved in choline metabolism; NPTX1 is a member of the pentraxin gene family with reported roles in endoplasmic reticulum stress¹⁹, and SGSH encodes the enzyme N-sulfoglucosamine sulfohydrolase, with no known roles in immunity as far as we can tell. Of note, we added SGSH to the revised manuscript given the strength of its HLA-pQTL association.

The boxplots for these genes are shown for the reviewer below (now included in Fig. 3d of the revised manuscript), including the newly added SGSH gene. The reviewer may notice that some of these pGenes have a different lead variant than what was reported in the original manuscript Fig. 2c-g; this is because we detected a minor bug that led to incorrect identification of the lead variant (e.g. HLA variant with strongest p-value). The revised plots are shown below; the associations are similar or even stronger (lower p-value) than what was reported in the original manuscript.

We have now added an expanded and more precise discussion of these points to the revised manuscript, as presented below.

Fig R1.4 (Fig. 3d of the revised manuscript): Selected pGenes from among the top 50 lead HLA-pQTL. These proteins were selected due to their strong associations with HLA genetic variation and unclear or undefined roles in the immune response.

In addition, we sought to explore alternative methods and datasets we could employ to address the reviewer's question, and more formally investigate the relationship between our HLA-pGenes, immune cells, and non-immune cells. First, we leveraged

tissue-specific gene expression gene sets from Uhlen *et al.* to ask whether the HLA pGenes detected in our study are enriched in genes that are specific to particular tissues²⁰. In the Uhlen *et al.* study, the authors performed RNA-seq analysis of 32 tissues and performed additional analyses to determine which genes are specific to particular tissues. This analysis revealed enrichment of HLA-pGenes in spleen, lymph node, tonsil, appendix, bone marrow, and lung, some of which (spleen, lymph node, tonsil, bone marrow, and lung in particular) are thought to be highly immune-infiltrated tissues.

Fig R1.5 (Extended Data fig. 8 of the revised manuscript): Enrichment of HLA-pGenes in tissue-specific gene sets determined from RNA-seq of 32 tissues from Uhlen *et al.* Science 2015. Two-sided p-values calculated using Fisher's exact test.

Motivated by this result, the identity of the pGenes themselves (i.e. many chemokines and immune-related genes), and the pathway analyses we presented in

the original manuscript, we next asked to what extent our HLA-pGenes are transcriptionally expressed on peripheral blood immune cells. To explore this question, we obtained a doublet-cleaned version of the Yazar *et al.* scRNA-seq atlas²¹—comprised of 909 donors and 735,000 PBMC immune cells—from Kang *et al.* We reprocessed these data according to the methods described in Kang *et al.*, and used it to explore expression patterns of HLA-I-specific and HLA-II-specific pGenes. Using methods outlined in Kang *et al.*, we uniformly re-processed the Yazar *et al.* dataset, plotting some markers from the original publication and new markers from our re-analysis to provide confidence in the cell type names assigned to each cluster, as shown below:

Fig R1.6 (Fig. 3a,b of the revised manuscript): Re-processed PBMC immune cell atlas originally described in Yazar *et al.* according to methods described in Kang *et al.* a, UMAP of cell types and cluster assignments. b, Selected differentially expressed genes defining clusters in a.

Using this resource, we asked whether the HLA-I-specific or HLA-II-specific pGenes we identified in the manuscript have varying expression across different immune lineages (e.g. T cells vs myeloid cells). We were motivated to do this based on the biology of HLA-I vs HLA-II—HLA-I is generally expressed on all nucleated cells, whereas HLA-II is expressed primarily on antigen presenting cells (APCs) such as myeloid cells and B cells. We were further motivated to conduct this analysis given our own results, which identified pGenes specifically affected by HLA-I compared to HLA-II and vice versa—i.e. if there are different pGenes affected, are there also different cell types?

We plotted all of the HLA-I-specific and HLA-II-specific pGenes on the re-processed Yazar *et al.* atlas, and added a color bar along the rows of the heatmap to indicate whether the gene is HLA-I-specific or HLA-II-specific. First, this analysis showed strong cell type-specificity of the pGenes in general—e.g. we observed that subsets of the pGenes were more highly expressed on myeloid cells, some more on T cells,

etc. Curiously, this analysis also showed that despite the fact that there are pGenes specific to HLA-I or HLA-II, the cell type expression patterns are largely overlapping between the sets of pGenes (i.e., no clustering by purple / orange along the rows (genes) of the heatmap). This result is especially intriguing given the well-known, more specific expression and activity of the HLA-II genes on APCs²²—i.e. despite this specific expression, HLA-II genetic variants affect expression of genes on non-APCs (i.e. T cells) as well.

Fig R1.7 (Fig. 3c of the revised manuscript): Immune cell type-specificity of HLA-pGenes. Mean expression of HLA-I-specific (purple) and HLA-II-specific

(orange) pGenes across clusters identified in the Yazar *et al.* immune cell atlas. Heatmap is clustered across rows and columns; dendrograms are omitted from the plot for clarity. The data show that HLA-I and HLA-II-specific pGenes are broadly expressed across immune lineages (i.e. no obvious clustering of purple/orange along the rows).

We next sought to explore the expression patterns of HLA-pGenes on non-immune cells, and compare them to expression on immune cells. Given the enrichment of HLA-pGenes on lung-specific genes, we obtained a healthy lung cell scRNA-seq atlas from Travaglini *et al.*²³, and used it to plot HLA-I-specific and HLA-II-specific pGenes as we did with the immune cell atlas from Yazar *et al.* Importantly, we chose this dataset because it is comprised of normal lung unaffected by disease, and has excellent representation of both immune cells and non-immune cells. We used cluster definitions exactly as specified in the Travaglini *et al.* original study. While we acknowledge that there exist larger lung scRNA-seq atlases (e.g. the HLCA effort by Sikemma *et al.* Nature Medicine 2023), that dataset developed its cell type annotations on a shared embedding with both healthy and diseased lung, whereas the Travaglini dataset focuses exclusively on normal lung, unaffected by disease. This difference is important, as the HLA-pQTL we detected in the UK Biobank were primarily from healthy donors.

As observed in the Yazar *et al.* dataset, we observed no clear clustering between HLA-I-specific and HLA-II-specific pGenes, but we did detect pronounced expression of HLA-pGenes on non-canonical immune cells, e.g. in fibroblasts and lung epithelial cells, and strong cell type-specificity overall. The expression of pGenes on these on these non-immune cells was comparable to pGene expression on immune cells. Of course, the pGenes we identified were found using peripheral blood data, and not lung tissue—as such, we caution against overinterpretation of these data. However, the data do suggest that pGenes affected by HLA genetic variants may be expressed comparably in non-immune cells compared to immune cells.

Fig R1.8 (Extended Data Fig. 9 of the revised manuscript): Lung cell type-specificity of HLA-pGenes. Mean expression of HLA-I-specific (purple) and HLA-II-specific (orange) pGenes across clusters identified in the Travaglini *et*

a/ lung cell atlas. Heatmap is clustered across rows and columns; dendrograms are omitted from the plot for clarity. The data show that HLA-I and HLA-II-specific pGenes are broadly expressed across immune lineages (i.e. no obvious clustering of purple/orange along the columns). Furthermore, the data show that pGene expression in non-canonical immune cells, such as fibroblasts, goblet cells, and alveolar epithelial cells.

Altogether, these data—spanning tissue-specific gene sets and two scRNA-seq atlases comprising both immune and non-immune cells—add depth to the interpretation of HLA-pGenes, suggest that HLA-I-specific and HLA-II-specific pGenes converge on shared cell types, and that HLA-pGenes can be expressed on both immune and non-immune cells. More generally, our data motivate future single cell *trans* HLA-pQTL studies to more formally investigate the myriad effects of HLA genetic variation on gene expression in individual cell types.

We have added a detailed description of these data to the results section of the revised manuscript as follows:

*“To explore the expression patterns of pGenes identified in our analysis, we first asked whether our pGenes were enriched in tissues-specific gene sets developed from gene expression data from Uhlen et al³⁴. (**Extended Data Fig. 8**). This analysis revealed enrichment of HLA-pGenes in spleen, lymph node, tonsil, appendix, bone marrow, and lung, which are thought to be extensively immune infiltrated.*

*Motivated by this result, we next asked to what extent our HLA-pGenes are transcriptionally expressed on peripheral blood immune cells. To explore this question, we obtained a doublet-cleaned version of a peripheral blood immune cell single cell RNA-seq (scRNA-seq) atlas¹⁹—comprised of 909 donors and 735,000 PBMC immune cells—to explore the expression patterns of HLA-I-specific and HLA-II-specific pGenes (**Fig. 3a,b**). We then plotted all of the HLA-I-specific and HLA-II-specific pGenes on the re-processed Yazar et al. atlas (**Fig. 3c, Supplementary Table 16**). This analysis demonstrated strong cell type-specificity of HLA-pGenes, with subsets of the pGenes preferentially expressed on myeloid cells, T cells, and B cells. Curiously, this analysis also showed that despite the fact that there are pGenes specific to HLA-I or HLA-II, the cell type expression patterns are largely overlapping between the sets of pGenes. This result is especially intriguing given what is known about the well-known, specific expression and activity of the HLA-II genes on APCs³⁵—i.e. despite this specific expression, HLA-II genetic variants affect expression of genes on non-APCs (i.e. T cells) as well.*

We next sought to explore the cell-type expression patterns of HLA-pGenes on non-immune cells compared to immune cells. Given the enrichment of HLA-pGenes on lung-specific genes, we obtained a lung cell scRNA-seq atlas from Travaglini et al²⁰, and plotted HLA-I-specific and HLA-II-specific pGenes. Importantly, we chose this dataset because it is comprised of normal lung unaffected by disease, and has excellent representation of both immune cells and non-immune cells. We used cluster definitions exactly as specified by the Travaglini et al. study.

As observed in immune cell atlas, we observed no clear differences in cell type-specific expression between HLA-I-specific and HLA-II-specific pGenes in the lung,

but we did detect pronounced expression of HLA-pGenes on non-canonical immune cells, e.g. in fibroblasts and lung epithelial cells (**Extended Data Fig. 9, Supplementary Table 17**). Importantly, the pGenes we originally identified in the UK Biobank were found using peripheral blood data, and not lung tissue—as such, we caution against overinterpretation of these data. However, the data do suggest that pGenes affected by HLA genetic variants may be expressed comparably in non-immune cells compared to immune cells. More generally, our data motivate future single cell trans HLA-pQTL studies to more formally investigate the myriad effects of HLA genetic variation on gene expression in individual cell types.

To extend our scRNA-seq analysis which suggested both immune and non-immune functions of the pGenes, we manually examined individual pGenes identified in our analysis, particularly among the top 50 strongest associations by p-value (**Fig. 3d**). For example, SFTPD is generally thought of as a pattern recognition receptor, and thus might be considered a part of the innate immune response. The identification of this pGene is surprising and novel in the context of HLA genetics, which—through their central role in antigen presentation and recognition by T cells—might be thought to primarily affect T cell-related pathways. LRPAP1 is the LDL receptor-related protein 1; aside from a recent report suggesting that LRPAP1 may facilitate immune evasion from viruses³⁶, it is both unclear and intriguing to speculate as to how LRPAP1 expression may be affected by HLA genetics. ENPP6 is an enzyme involved in choline metabolism; NPTX1 is a member of the pentraxin gene family with reported roles in endoplasmic reticulum stress³⁷, and SGSH encodes the enzyme N-sulfoglucosamine sulfohydrolase, with no known roles in immunity. Altogether, these data highlight the potential of using HLA-pQTL associations to discover novel relationships between HLA genetic variation and proteins with unclear connections to the immune response.

12. Results “Atlas...”: Please clarify in the Methods the approach used to create the atlas of proteins affected by diseases

Response:

We thank the reviewer for the opportunity to clarify this point in the Methods section of the manuscript. To create the atlas, we searched the literature for major autoimmune diseases that have well-known, prior reported genetic associations with HLA. We reasoned that given the strong prior associations of HLA genetic variants with these diseases, it would be of interest to readers to know which proteins are affected by those same HLA variants in our analysis. In addition, since the 1-field and 2-field alleles are often described as being associated with disease (e.g. without fine-mapping information), we performed an additional conditional analysis for each pGene using only 1 and 2-field alleles, and populated this table with the results from that analysis.

These details are now described in the results section of the revised manuscript:

“To create the atlas, we searched the literature for major autoimmune diseases that have prior reported genetic associations with HLA. Furthermore, since the 1-field and 2-field alleles are often described as being associated with disease (e.g. without amino acid-level fine-mapping information), we performed an additional conditional

analysis for each pGene using only 1 and 2-field alleles, and asked which alleles are associated with the immune-mediated diseases of interest.”

And also in the revised methods section:

“Atlas of proteins affected by disease-associated HLA alleles

We manually searched the literature for major autoimmune diseases associated with HLA genetic variants—in particular, 1-field and 2-field alleles. To link our HLA-pQTL to these diseases, we first performed conditional analyses for each pGene using only 1- and 2-field alleles (i.e. without fine-mapping) to identify significant ($P < 5 \times 10^{-8}$) HLA alleles associated with each pGene, and used these associations to populate Table 1.”

However, the reviewer’s question prompted us to address the relationship between HLA-pQTL and HLA-associated diseases in a more systematic, comprehensive fashion, rather than simply pre-selecting major autoimmune diseases commonly thought to be associated with HLA genetic variation.

We reasoned that since we have fine-mapped HLA-pQTL, we might overlap our results with any fine-mapped HLA-trait associations in the UK Biobank that also used the correct MHC fine-mapping methods. Encouragingly, the analysis in Sakaue *et al.* Nature Genetics 2021 did exactly this—the authors rigorously imputed HLA alleles and amino acids using the same methods as we do in our current study, and then ran association tests for the HLA imputed variants against diseases and biomarkers (e.g. red blood cell counts) in the UK Biobank. Thus, we obtained these fine-mapped summary statistics and overlapped them with our HLA-pQTL fine-mapped data and displayed them in the following network visualization, also noting directions of effect. As this analysis overlaps fine-mapped signals from HLA-trait studies and HLA-pQTL, it effectively amounts to a colocalization between the two studies:

Fig. 4

Fig R1.9 (Fig. 4 of the revised manuscript): Integration of fine-mapped HLA-pQTL with UK Biobank fine-mapped HLA-trait signals from Sakaue *et al.* Nature Genetics 2021. Abbreviations for diseases and biomarkers are included in supplementary table 20 of the revised manuscript.

This map in the revised manuscript allows us to visualize novel links between fine-mapped HLA genetic variants, proteins, diseases, and traits. For example, this analysis reveals that B_180_Q affects both CLEC4A, a C-type lectin receptor important for the innate immune response, and HDL cholesterol. Moreover, the DRB1_01_03 allele—which is thought to be a risk allele for ulcerative colitis based on prior GWAS—was associated with both UC and several inflammatory genes such GZMA, GZMH, and IL10.

The reviewer may notice that some of the diseases listed in the manually curated Table 1 are not included in this map. This is likely because those HLA-disease associations were detected in bespoke, large GWAS cohorts curated with many more diseases cases than are present in the UK Biobank; indeed, the UK Biobank has a well-documented healthy volunteer bias that may limit the power to detect HLA-disease associations. For these reasons, we believe it is best to present both pieces of data in the revised manuscript—the manually curated table with well-known HLA-disease associations from the literature together with the 1-field and 2-

field alleles, and the more systematic “colocalization” between fine-mapped HLA-pQTL and fine-mapped HLA-disease/trait signals from Sakaue *et al.*

We have added a description of these points to the results section of the revised text, clarifying the distinction between the manually curated Table 1 and the overlap of fine-mapped signals (“colocalization”) presented in Fig. 4:

“We sought to more systematically explore the relationships between HLA-pQTL and HLA associations with disease. Standard colocalization methods are limited by the extreme polymorphism and LD of the MHC; in addition, a major assumption of HLA-trait association studies is that coding variation drives the majority of the genetic heritability. Moreover, most colocalization methods assume a single causal variant; while some methods do allow for more than one causal variant, such methods have not been tested or rigorously benchmarked on the MHC.”

*Despite these limitations, we obtained summary statistics for fine-mapped HLA-disease summary statistics from the UK Biobank from Sakaue *et al.*, and overlapped them with our fine-mapped HLA-pQTL (Fig. 4, Supplementary Table 20). Specifically, Sakaue *et al.* performed single-marker tests with their HLA imputed variants; thus, we overlapped all of our single-marker lead and conditional HLA-pQTL with the single-marker summary statistics from Sakaue *et al.* We reasoned that since this analysis represents an overlap between two sets of fine-mapped summary statistics, it effectively amounts to a colocalization between HLA-pQTL and HLA-disease signals. Notably, the analysis by Sakaue *et al.* analyzed HLA associations with both diseases and quantitative traits (e.g. HDL cholesterol) in the UK Biobank.*

*The integration of HLA-pQTL and HLA-disease summary statistics allows us to visualize novel links between fine-mapped HLA genetic variants, proteins, diseases, and traits. For example, this analysis reveals that HLA-B position 180_Q affects both CLEC4A, a C-type lectin receptor important for the innate immune response, and HDL cholesterol. The DRB1*01:03 allele—which is thought to be a risk allele for ulcerative colitis (UC) based on prior GWAS⁴¹—was associated with both UC and several inflammatory genes such as GZMA, GZMH, and IL10. Mechanistically, as the GZM genes are thought to be expressed on gut T and NK cells, our data may suggest that the HLA risk allele DRB1*01:03 modulates T and NK function to affect risk of UC. Moreover, the IL10 gene—also affected by DRB1*01:03—has long been considered a potential drug target for immune-mediated diseases⁴². Thus, our data suggest that targeting IL10 may be particularly viable in individuals with the DRB1*01:03 UC risk allele.”*

13. Discussion: The first paragraph of Discussion states that “our fine-mapping analyses strongly implicate amino acid positions within the peptide binding groove.” However, I only see the peptide binding groove mentioned in the Results section labeled “Discovery of HLA-pQTL in the UK Biobank” and not in the section on fine-mapping. Please clarify.

Response:

We apologize to the reviewer for the lack of clarity on this point and are happy to provide additional data and text to clarify. The “discovery of HLA-pQTL in the UK Biobank” section of the results section refers to the single-marker lead HLA-pQTL and single-marker conditional tests. These tests identify either 1-field alleles (e.g. DRB1*04), 2-field alleles (e.g. DRB1*04:01), or specific amino acid polymorphisms (e.g. DRB1 Position 77 N (exon 2)) that are either the lead or conditionally significant association for a given protein. Based on these results we plotted the enrichment in the peptide binding groove for all single marker lead and conditional HLA-pQTL shown in Fig. 1d. Since the single marker conditional analyses also narrow the signals to specific amino acid polymorphisms, these analyses can also be considered HLA fine-mapping.

The section on fine-mapping HLA-pQTL in the original manuscript specifically referred to conditional haplotype tests, which as described in the methods, refers to joint modeling of all amino acid polymorphisms at a particular amino acid position, in order to enable statements such as “DRB1 position 11 is associated with protein LILRB4”. We acknowledge the reviewer’s confusion that we did not explicitly show that the amino acid positions identified from these fine-mapping analyses are enriched in the peptide binding groove.

Thus, to address the reviewer’s question, we have now aggregated all significant results from the conditional haplotype tests and asked how many are inside and outside of the peptide binding groove. This analysis revealed that 78.1% of HLA-I, 81.8%, and 80.0% of the significant amino acid positions were in the peptide binding groove compared to outside of it, consistent with the results from the single marker lead / conditional single-marker HLA-pQTL:

Proportion of fine-mapped amino acid positions within and outside the peptide binding groove

Fig R1.10 (Extended Data Fig. 4f of the revised manuscript): Proportion of fine-mapped amino acid positions within and outside the peptide binding groove (exons 2 and 3 for HLA-I; exon 2 for HLA-II).

These data have been added as Extended Data Fig. 4f in the revised manuscript. To clarify that single marker conditional tests and conditional haplotype tests are both examples of HLA fine-mapping, we have now removed the section header “Fine-mapping HLA amino acid positions that affect protein expression” and instead amended the first section header to now read “Discovery and fine-mapping of HLA-pQTL in the UK Biobank”. We have amended the results section on fine-mapping to include mention of these data:

“To pinpoint HLA amino acid positions that affect protein expression, we performed omnibus and conditional haplotype tests for all 2940 proteins (Methods, Fig. 1e,f, Extended Data Fig. 4, and Supplementary Table 3). These analyses revealed enrichment of the fine-mapped amino acid positions in peptide binding groove, consistent with the enrichment observed for single-marker lead and conditional HLA-pQTL (Extended Data Fig. 4f). The fine-mapping analyses also revealed strong associations between multiple independent amino acid positions—themselves previously associated with risk of autoimmune diseases^{6,8}—with disease relevant proteins....”

In addition, we have amended the discussion to be clearer regarding the enrichment in the peptide binding groove. Instead of the prior “our fine-mapping analyses strongly implicate amino acid positions within the peptide binding groove”, we now state, “our HLA-pQTL analyses—both single marker tests and fine-mapping analyses—strongly implicate amino acid positions within the peptide binding groove.”

14. Discussion: Overall, this section is very brief. It would be good to have more interpretation of key results and limitations, possibly including: (a) value of fine mapping of HLA-pQTL, (b) further consideration of disease associations for the HLA-pQTL, (c) limitations related to LD within the HLA region, and (d) limitations related to race/ancestry composition of the study.

Response:

We thank the reviewer for this suggestion and agree that our discussion section should be expanded. We have now done so in the revised manuscript in an attempt to synthesize the key HLA-pQTL findings (i.e. discovery of trans associations), mechanistic hypotheses enabled by the data, various genomic follow-up analyses described in our response to the reviewers, and limitations of the study, specifically with respect to ancestry and statistical methodology (i.e., the need for more formal statistical methods to perform colocalization with HLA variants).

We have included the full, expanded discussion section of the revised manuscript below:

“Altogether, our data suggest that HLA risk alleles exert disease-relevant trans effects on plasma protein expression. We hypothesize that there may be two primary mechanisms by which HLA polymorphism affects protein expression. First, as our fine-mapping analyses strongly implicate amino acid positions within the peptide binding groove, we suspect that variation in antigen presentation affects T and NK cell recognition of peptide-MHC, which in turn may modulate the expression levels of proteins related to T and NK effector function. Indeed, the expression of a wide array and chemokines and cytokines may be related to T and NK cell recognition of peptides presented by antigen presenting cells²⁵. The second potential mechanism by which such changes in protein expression may arise is through cell intrinsic effects in macrophages and other antigen presenting cells (APCs) that demonstrate high expression of HLA. Indeed, this could explain the discovery of proteins involved in antigen processing and presentation, macrophage activating and inhibitory receptors. Future functional studies clarifying the exact mechanisms by which HLA alleles affect protein expression are warranted, especially since several of the proteins affected by HLA genetic variation do not have an immediate connection to T cell or APC biology.

Indeed, to support these observations, we examined the transcriptional expression of HLA-pGenes across two independent scRNA-seq datasets, and evaluated enrichment of HLA-pGenes in tissue-specific gene sets. These data confirmed that while HLA-pGenes are highly expressed on immune cells, they may also be expressed on non-immune cells in tissues. Furthermore, the data suggest that while HLA-I and HLA-II genetic variation do affect distinct genes (as demonstrated by our conditional analyses adjusting for all 2-field alleles of the opposite locus), the HLA-I-

specific and HLA-II-specific pGenes largely converge on shared immune cell types. These data may be surprising given the restricted expression of HLA-II to APCs, as compared to HLA-I, which is expressed on all nucleated cells. These analyses strongly motivate future studies formally investigating the extent to which HLA genetic variation exerts trans effects on gene and/or protein expression in particular cell types (e.g, single cell HLA-pQTL).

As we discuss in our manuscript, standard fine-mapping and colocalization methods optimized for bi-allelic SNPs in other parts of the genome are not appropriate for the MHC. Despite these limitations, we integrated our fine-mapped HLA-pQTL with fine-mapped HLA-trait data from Sakaue et al. to create a map of potential effects of HLA-pQTL on human disease. Such integration enables exploratory hypothesis generation regarding the mechanisms underlying HLA associations with disease and potential drug targets; indeed, the proteins shown in Fig. 4 may serve as candidates for future functional studies. Importantly, given the key differences in assumptions and methodology between standard fine-mapping methods and HLA fine-mapping, further benchmarking and developments in statistical methodology are warranted to perform fine-mapping, colocalization, and Mendelian randomization analyses with HLA genetic variants.

Our study reports a wide range of variance in protein expression explained by HLA-pQTL. Given the trans SNP pQTL reported by the flagship Sun et al. study, an intriguing question is whether HLA effects on protein expression act in similar pathways to trans pQTL outside the HLA locus. We performed an exploratory analysis investigating the pathways enriched in the genes proximal to these non-HLA trans pQTL (Extended Data Fig. 11)- curiously, no pathways reached statistical significance after adjusting for multiple tests, and only one—natural killer cell lectin-like receptor binding—was related to the immune response. This analysis may suggest that both HLA genetic variation and non-immune-related genetic variation act independently to influence protein expression. However, we urge caution in over-interpretation of the results, as we cannot rule out the possibility that genes containing non-HLA trans-pQTL are indirectly related to the immune response or the HLA genes themselves through currently unknown mechanisms.

A question that should be comprehensively addressed in the future as protein panels for pQTL mapping increase in size and scope is whether HLA-pGenes are more correlated with each other than by chance, e.g. compared to correlations between all proteins on the panel. In general, we observed weak correlations between most proteins on the Olink panel analyzed here (Median Spearman rho across all proteins: 0.05 (I.Q.R. 0.02-0.13; Extended Data Fig. 13a). Furthermore, when we compared the distributions of correlations pairwise between HLA-pGenes and all pGenes, we found that statistically, correlations between the background set of all pGenes were higher than those between HLA-pGenes (Extended Data Fig. 13b). However, the difference between these two distributions was modest at best, and the observed statistical significance may simply reflect the large sample size in each distribution (e.g. pairwise correlations between hundreds of pGenes). In the present study we perform gene set enrichment and cell type-specificity analyses arguing in favor of gene families or cell types affected by HLA genetic variation; however, there does not appear to be strong evidence for HLA-pQTL tagging correlated proteins.

Whether the choice of protein expression profiling platform itself affects correlations between proteins should also be investigated thoroughly in the future.

An important caveat of the current study is that only soluble proteins are measured, which may not reflect the fraction of proteins in physiologically relevant compartments. Prior studies have indicated that the soluble forms of at least a few of the proteins included in the Olink panel (e.g. PD1, LAG3) are cleaved directly from the cell surface through membrane shedding²⁶. However, where possible, future studies should investigate the effects of HLA genetic variation on membrane-bound proteins.

Moreover, since application of genetic colocalization and Mendelian randomization may not be nuanced enough for the MHC¹² given the high degree of linkage disequilibrium, mechanistic functional studies may be necessary to validate the extent to which HLA-pQTL are truly causal for disease risk. Moreover, while our study did conduct preliminary replication analysis in individuals of non-European ancestry, the sample sizes of the non-European ancestry cohorts were much smaller than that of the European cohort. Thorough investigation of individuals of diverse genetic ancestry is sorely needed, both for HLA-pQTL analysis and more broadly. In general, we emphasize the value of expanding HLA-pQTL analyses to specific cell types, tissues, and diseases pending data availability, as current efforts are limited to blood and a limited panel of proteins. More generally, we hope that our current study will pave the way for investigation of HLA regulatory effects on quantitative traits beyond protein expression.”

15. Methods “Plasma protein quality control” and “Covariates and linear models for HLA-pQTL mapping”:

a. Please clarify the relevant covariates when saying “For each protein, we performed..., regressed all covariates...”

Response:

We thank the reviewer for this suggestion. The sentence the reviewer refers to is in the results section of the manuscript—per the reviewer’s suggestion, we have amended it to specify the exact covariates used in the study. The amended section with this change in the revised manuscript now reads as:

*“For each protein, we performed inverse rank-normalized transformation, regressed all covariates, and scaled the protein expression values to mean 0 and standard deviation 1 prior to pQTL mapping. We used the exact covariates used for pQTL mapping as in the flagship study—these covariates were age, age², sex, age*sex, age²*sex, batch, UKB centre, UKB genetic array, time between blood sampling and measurement, and the top 20 principal components (PCs) of genetic ancestry.”*

b. Please clarify what are REGENIE LOCO values and define the abbreviation LOCO at first use in the text. Be consistent with the usage of term, you used “Regenie LOCO scheme” later in the Methods.

Response:

We thank the reviewer for this suggestion and we are happy to clarify. LOCO stands for “leave one chromosome out”. The flagship pQTL paper generated predictions of protein expression using REGENIE—for each protein, the LOCO values represent the overall contribution of SNPs on all chromosomes except for chromosome 6 to the expression of the protein of interest; thus, they represent polygenic effects that may confound the main genetic effect being tested. These values are known to more precisely account for population structure, sample relatedness, and LD—indeed, according to the REGENIE paper, this approach performs these corrections while avoiding proximal contamination, which can result in reduced power of genetic association analyses. Critically, the REGENIE LOCO values were also used in the flagship pQTL paper (subtracted from the protein expression residuals regressing all covariates); thus, we chose to include them in our study to remain consistent with the earlier study.

Per the reviewer’s suggestion, we have defined the LOCO abbreviation as (leave one chromosome out) at first mention in the text; in the methods, we have clarified what the REGENIE LOCO values are as written below, under ‘Plasma protein measurements and quality control’. Furthermore, we have removed mention of “Regenie LOCO scheme” in order to remain consistent with usage of the term “REGENIE LOCO values” in the revised manuscript.

“For each protein, the LOCO values represent the overall contribution of SNPs on all chromosomes except for chromosome 6 to the expression of the protein of interest; thus, they represent polygenic effects that may confound the main genetic effect being tested. These values are known to more precisely account for population structure, sample relatedness, and LD—indeed, according to the REGENIE paper, this approach performs these corrections while avoiding proximal contamination, which can result in reduced power of genetic association analyses. Critically, the REGENIE LOCO values were also used in the flagship pQTL paper (subtracted from the protein expression residuals regressing all covariates); thus, we chose to include them in our study to remain consistent with the earlier study. Thus, consistent with the flagship study, for each protein, we performed inverse rank-normalized transformation and regressed all covariates. We then subtracted the REGENIE LOCO values for each protein. The resulting expression values were scaled to mean 0 and standard deviation 1, and then used as input to pQTL mapping.”

16. Methods “Conditional haplotype analysis”:

- a. What window size did you apply to select variants for conditional analysis?
- b. Did you consider LD for the selection of variants for conditional analysis?

Response:

We thank the reviewer for this question and refer them to our response to our response to the reviewer’s question 6, in which we explain the critical differences between standard fine-mapping and HLA fine-mapping. As part of that response, we state that we do not pre-select HLA variants for single-marker conditional analysis or conditional haplotype analyses (which represent joint models including all amino acid polymorphism at a particular amino acid position, as described in response to the

reviewer's question 13) based on window size or LD; rather, we condition the model on the lead variant and test all other HLA variants. We repeat this for each pGene identified in our analyses.

Minor:

1. Figure 1b: the conditional HLA-pQTL should also be presented in the legend

Response:

Per the reviewer's suggestions, we have added colors to depict conditional HLA-pQTL in 1b, and to the legend. We note to the reviewer an error in the original manuscript reporting the correct number of conditional HLA-pQTL, this number should be 1339 HLA-pQTL as now correctly depicted in Fig. 1b (the ordering of loci with most HLA-pQTL, e.g. HLA-DRB1, followed by HLA-B, etc was correct as presented originally). We apologize for this error; the revised plot is shown below with the reviewer's suggestions incorporated:

Fig R1.11 (Fig. 1b of the revised manuscript): Number and location of proteins (pGenes) in the discovery cohort affected by lead and conditional HLA variants corresponding to each HLA-I and HLA-II locus.

2. Figure 1d: please define the abbreviation “PBG” in the legend.

Response:

To address the reviewer’s suggestion, we have defined PBG in the plot title in Fig. 1c—the title now reads “**Proportion of HLA-pQTL within and outside the peptide binding groove (PBG)**”; the legend reads “within PBG” and “outside PBG” for brevity.

3. Results “Fine-mapping...”: In the following sentence “These analyses revealed strong...”, please clarify what is meant by “strong” association – is it based on p-value, effect size, or otherwise. If p-value, we suggest the wording “These analyses revealed highly statistically significant associations...”

Response:

We thank the reviewer for this comment and have amended this sentence per the reviewer’s suggestion. This sentence now reads as “***These analyses also revealed highly statistically significant associations between multiple independent amino acid positions—themselves previously associated with risk of autoimmune diseases—with disease relevant proteins.***”

4. Results “Fine-mapping...”: Please define the abbreviation “APC” at first use in the text.

Response:

Per the reviewer’s suggestion, APC is now defined at first use in the text: “...suggesting that HLA-pQTL may reflect cell-cell interactions between **antigen-presenting cells** (APCs) and myeloid cells expressing LILR proteins.”

5. Results “Fine-mapping...”: In the sentence “The unexpectedly strong associations between HLA...”, please clarify why the associations were considered unexpectedly strong?

Response:

*We thank the reviewer for this suggestion—the use of the word “unexpectedly” is unclear in this context. Our original intention was to convey that these are highly statistically significant associations—per the reviewer’s earlier suggestion, we have reworded this sentence in the revised text as follows: “The **highly statistically significant** associations between HLA polymorphism and expression of proteins...”*

6. Supplementary Tables: Please add footnotes to define abbreviations and column labels.

a. Need to improve the resolution of Figs. 2a and 2b, they are not readable. Additionally, these figures provide too much information, and it is hard to tell the

main message from these two panels.
b. Please add figure legends for panels f and g.

Response:

We thank the reviewer for these suggestions, and agree with the reviewer that the resolution of Figs. 2a and 2b needs to be improved. First, per the reviewer's suggestion, we have added a dedicated legend defining all abbreviations and columns to each supplementary table in the revised manuscript.

As the prior boxplots in Fig. 2 have now been moved to the new Fig. 3 in the revised manuscript, we enlarged the network plots in Fig. 2, and limited them to the strongest pathway associations (FDR $P < 0.0001$ instead of the prior FDR $P < 0.1$). The prior FDR $P < 0.1$ plots, which show a wider representation of the pathways enriched in HLA-I-specific or HLA-II-specific pGenes, are now shown as new extended data figures in the revised manuscript; one extended data figure each for HLA-I and HLA-II. We present the revised figure 2 below, which has significantly improved readability compared to the prior figure:

Fig R1.12 (Fig. 2 of the revised manuscript): Trans HLA-pQTL networks and gene families affected by HLA-I and HLA-II genetic variation. a, Network depicting proteins and gene ontology terms affected by only HLA-I lead variants after adjusting for all HLA-II two-field alleles. Edges connect proteins to gene ontology terms. b, Network depicting proteins affected by only HLA-II lead variants from after adjusting for all HLA-I two-field alleles. Plots show pathways enriched at FDR $P < 0.0001$; full depiction of all significant pathways (FDR $P < 0.1$) are shown in the Extended Data.

7. Table 1: Please define the disease abbreviations in a footnote

Response:

We have now added a footnote defining all disease abbreviations to Table 1 per the reviewer's suggestion.

8. Methods "Plasma protein quality control..." and "Covariates and linear models for HLA-pQTL mapping": It is confusing of what protein value was used as input for PQTL. In the Results, you noted that "covariate-residualized inverse rank-normalized protein expression", but in the Methods you stated that "we subtracted REGENIE LOCO values from the residualized protein expression values to account for..." which is not consistent with the description of Results.

Response:

We thank the reviewer for this question and apologize for the confusion. Consistent with the flagship study and with our statement in the methods, the exact protein value used as input for pQTL mapping is the covariate-residualized inverse rank-normalized protein expression, with REGENIE loco values for that protein subtracted. The resulting protein expression values are then scaled to mean 0 and standard deviation 1. The protein expression values are then used as input to pQTL mapping.

We have clarified this in the methods section of the revised manuscript:

"Consistent with the flagship study, for each protein, we performed inverse rank-normalized transformation and regressed all covariates. We then subtracted the REGENIE LOCO values for each protein. The resulting expression values were scaled to mean 0 and standard deviation 1, and then used as input to pQTL mapping."

The results section has been updated to now read:

"To fine map HLA-pQTL, we first tested the association between each imputed HLA variant and scaled, covariate-residualized inverse rank-normalized protein expression with REGENIE LOCO (leave one chromosome out) values subtracted, (Methods) using linear regression models. These values are known to more precisely account for population structure, sample relatedness, and LD—indeed, according to the REGENIE paper, this approach performs these corrections while avoiding proximal contamination, which can result in reduced power of genetic association analyses.^{1,2}"

9. Methods “Quality control of SNP...”:
- Add literature reference for PLINK
 - What is the software/tool used for IBD and IBS?
 - Add reference for Minimac4 or Michigan Imputation Server.

Response:

Per the reviewer’s suggestions, we have added literature references for PLINK, clarified that PLINK was used for IBD and IBS, and added a reference for the Michigan Imputation server to the methods section of the revised manuscript.

10. Methods “Covariates and linear models for HLA-pQTL mapping”: In the first sentence of this section, please clarify what was the outcome variable used in pQTL mapping.

Response:

We thank the reviewer for this suggestion. This sentence now reads,

“For each of the 2940 proteins, we fit a multivariable linear regression model testing the protein expression against the imputed dosage (continuous value between 0 and 2) for each of the 2284 HLA variants independently together with all covariates included in the flagship UKB-PPP study. Consistent with the flagship study, for each protein, we performed inverse rank-normalized transformation and regressed all covariates. We then subtracted the REGENIE LOCO values for each protein. The resulting expression values were scaled to mean 0 and standard deviation 1, and then used as input to pQTL mapping.”

11. Methods: Would help to re-order this section so that the sub-sections are presented in the same order as the Results.

Response:

We thank the reviewer for this suggestion and agree- the sub-sections are now presented in the same order as the Results.

12. Methods “Network analyses”:
- Need to clarify the association that was tested in the sentence “we tested the lead HLA variant adjusting for all two-field...”.
 - Add reference for cluster Profiler package.

Response:

Per the reviewer’s suggestion, we have re-written this section to be clearer, with an added reference for the clusterProfiler package. It now reads as:

“For the initial assessment of proteins uniquely affected by HLA-I or HLA-II genetic variation, we assembled two sets of mutually exclusive pGenes—those with a lead variant in HLA-I (HLA-A, B, or C), and those with a lead variant in HLA-II (HLA-DRB1, DQB1, DQA1, DPB1, or DPA1). To identify HLA-I-specific pGenes, we sought to rigorously rule out any confounding of HLA-II genetic variation for pGenes with an HLA-I lead variant (and likewise for pGenes with an HLA-II lead variant, i.e.

rule out confounding by HLA-I). Thus, for each pGene, we tested the lead HLA variant adjusting for all two-field alleles of the other class (i.e. for a pGene with HLA-I lead pQTL, we included all two-field HLA-II alleles as covariates in the linear regression model). pGenes that remained significant ($P < 5 \times 10^{-8}$) after this analysis were used as input to Gene Ontology enrichment analyses, using the full Olink panel ($N = 2940$ proteins) as the background. Significant (FDR $P < 0.1$) Gene Ontology terms were used as input to the `cnetplot()` function in the `clusterProfiler` package²⁷.”

13. Methods “Coding variation...”: Add reference for the genebase browser.

Response:

Per the reviewer’s suggestion, this has now been added.

Reviewer 2:

Reviewer #2 (Remarks to the Author):

Krishna et al. aim to determine the influence of HLA polymorphisms on plasma proteomic profiles. The major histocompatibility complex (MHC) region is characterized by its high polymorphism and strong linkage disequilibrium (LD). While various autoimmune and other diseases are associated with this region, pinpointing the precise alleles and variants responsible has proven challenging. Imputation within this region has improved the granularity of potential links to HLA alleles, variants and amino acid changes.

The authors leverage the recently released UK Biobank proteomics (3k platform from OLINK) dataset encompassing approximately 50,000 individuals with proteomics and genomic data. Their primary goal was to gain insights into how specific polymorphism within HLA genes, including previously identified disease risk loci, influence plasma protein levels. They employ a systematic workflow involving extensive conditional analyses to identify potentially independent variants.

The authors tackle a highly complex and intricate region of the genome and make interesting observations. I have questions and concerns that arise in the context of this densely packed and complex region:

We thank the reviewer for their questions and appreciate their comments regarding the systematic nature of the analyses presented in our manuscript. Below we present a point-by-point response to the reviewer's questions, which have greatly enhanced the robustness of the results.

Importantly, we also clarify the critical differences between our study and the flagship study. A key point is that the flagship study did not perform HLA imputation, HLA fine-mapping, conditional HLA analyses, or genomic follow-up analyses of HLA-pGenes (which were not discovered with imputed HLA genetic variants, and thus are not based on the most functionally important representation of HLA polymorphism). Moreover, motivated by the reviewer's questions regarding expression data, therapeutic targets, and integration of HLA-pQTL with HLA-associated diseases, we have performed extensive follow-up analyses to investigate the mechanisms and disease relevance of HLA-pQTL.

We anticipate that our manuscript will serve as an important resource both for future efforts to map HLA effects on quantitative traits, and to motivate biological hypothesis generation to discover new molecular effects of HLA genetic variation.

We thank the reviewer again for their questions and present a point-by-point response to their questions below:

1. It would be valuable to provide a more comprehensive overview of the analyses conducted in the recent flagship gwas paper in Nature. Specifically, what were the associations from this region previously reported?

Response:

We thank the reviewer for this important question, which represents an opportunity to clarify the key differences between our current HLA-pQTL study and the flagship pQTL analysis (Sun *et al.* Nature 2023).

As the reviewer correctly notes, the flagship paper was a GWAS on plasma protein levels using biallelic SNPs. The authors did include the MHC region in their analysis, reporting 861 proteins affected by biallelic SNPs anywhere in the MHC region (which includes 100s of genes including and beyond the classical HLA loci).

Our analysis is distinct from—and greatly extends—the flagship study by performing a focused, rigorous analysis of the MHC region. Our paper implements best practices for imputation of HLA alleles and amino acids, which are the functionally critical representations of HLA genetic variation (i.e. responsible for antigen presentation by the MHC). Critically, we detected 46 novel associations from the MHC region (using imputed HLA variants in our analysis) not detected in the flagship paper, suggesting that rigorous imputation of HLA alleles and amino acids can detect novel associations not detected with standard univariate GWAS / pQTL analyses:

HLA_allele	protein	beta	pval
HLA_DPB1_02_01	NUMB	0.0987684	4.20E-72
HLA_DRB1_03_01	HAVCR2	-0.0933709	6.02E-66
HLA_DRB1_03_01	DSC2	-0.0745259	3.98E-42
AA_DRB1_98_32549641_exon3_Ex	EBI3	0.055949	7.29E-25
AA_DRB1_71_32551949_exon2_ARS	VSIG4	0.0552555	3.82E-24
AA_DRB1_74_32551940_exon2_LR	PIK3IP1	-0.0506428	1.56E-20
AA_DRB1_77_32551931_exon2_Nx	CCDC134	0.0489802	2.35E-19
AA_DRB1_13_32552123_exon2_LS	C17orf49	-0.0481437	8.62E-19
AA_DQA1_25_32609153_exon2	PIGR	-0.0478103	3.29E-18
AA_DRB1_37_32552051_exon2_Y	MXRA8	0.0472276	8.15E-18
AA_DRB1_74_32551940_exon2_AE	RPA2	-0.04537	1.83E-16
HLA_B_08	CLEC14A	-0.0441769	5.42E-16
HLA_DQB1_03	MYOC	0.0441121	5.98E-16
AA_DQA1_25_32609153_exon2	ARHGEF12	0.0432732	2.15E-15
AA_DRB1_13_32552123_exon2_GHL	DSG3	0.0418784	1.40E-14
AA_A_76_29910759_exon2_AE	MYDGF	-0.0415979	1.90E-14
AA_A_142_29911198_exon3_T	VSIR	0.0416428	2.22E-14
AA_DRB1_11_32552129_exon2_DPS	CDH3	-0.0412364	3.13E-14
AA_DRB1_13_32552123_exon2_RY	BID	0.0411735	4.45E-14
AA_B_163_31324003_exon3_AT	WFDC2	-0.0420234	4.49E-14
AA_DQB1_84_32632600_exon2	BPIFA2	0.0416997	1.05E-13
AA_C_147_31238961_exon3_Lx	HEPH	0.0403019	1.19E-13
AA_B_97_31324201_exon3_W	LAMA4	-0.0402617	1.36E-13
HLA_DQA1_05	APCS	0.0403498	1.36E-13
AA_DRB1_149_32549488_exon3_Q	VSIG2	0.0401489	1.68E-13
AA_DRB1_74_32551940_exon2_GRx	NT5C1A	-0.0400267	1.87E-13
AA_DRB1_11_32552129_exon2_DGP	INSR	0.0402936	2.65E-13
HLA_C_08	GDF2	-0.0396066	3.78E-13
AA_DPB1_86_33048697_exon2	FOLR1	0.0394732	4.52E-13
AA_C_116_31239054_exon3_Y	GFRA1	0.0391766	5.72E-13
AA_DRB1_30_32552072_exon2_CGL	AXL	0.0391976	5.99E-13
AA_B_97_31324201_exon3_CNR	CLEC1A	0.0395502	6.84E-13
AA_DRB1_37_32552051_exon2_SYx	C1S	-0.0388748	8.01E-13
HLA_DRB1_03_01	GIMAP7	0.0384532	2.58E-12
AA_DRB1_13_32552123_exon2_LRY	TLR1	0.0385331	2.72E-12
AA_DRB1_74_32551940_exon2_GLR	CCL14	-0.0381863	2.81E-12
AA_B_325_31322303_exon7_C	CLEC6A	-0.0385313	2.89E-12
AA_DRB1_74_32551940_exon2_LR	DDI2	0.0383052	3.13E-12
AA_DQB1_minus18_32634341_exon1_V	ITGAM	-0.037531	5.90E-12
HLA_DRB1_03	DKK4	0.037519	7.53E-12
HLA_C_08	BMP10	-0.0375018	8.76E-12
AA_DQB1_57_32632681_exon2_A	BCL2L15	0.0375535	9.53E-12
AA_DQB1_37_32632741_exon2_Y	IL1RN	-0.0371556	1.02E-11
AA_A_77_29910762_exon2_DP	CD84	0.0370182	1.16E-11
AA_DRB1_minus24_32557503_exon1_L	RGMB	-0.0367911	1.26E-11
AA_C_152_31238946_exon3_EV	PODXL	-0.0369801	1.33E-11

Table R2.1: Proteins associated with imputed HLA genetic variants that were assayed in the original flagship study, but not detected as associated with HLA SNP genetic variation.

Furthermore, consistent with many prior studies implementing specialized fine-mapping methods to pinpoint specific MHC amino acid positions linked to disease risk, we now implement these methods to fine map MHC amino acid positions to plasma protein levels.

Moreover, our original analysis performed conditional analyses to detect protein effects and gene families specific to HLA-I or HLA-II. Prompted by this reviewer's suggestions, we have extended these original analyses to investigate HLA-pGene expression patterns across multiple single cell RNA-sequencing datasets, tissue specific gene expression signatures enriched in HLA-pGenes, enrichment of drug targets in HLA-pGenes, and integration of our fine-mapped HLA-pQTL with fine-mapped HLA-disease summary statistics from the UK Biobank performed in Sakaue *et al.* Nature Genetics 2021.

Altogether, the analyses presented in the revised manuscript represent a comprehensive investigation of the effect of functional HLA genetic variation on plasma protein expression, which will hopefully serve as a guide and resource for future studies investigating the effects of HLA polymorphism on quantitative traits including and beyond protein expression.

2. Given the complexity of the MHC region, it is crucial to address whether the analysis accounts for potential effects from HLA Class III genes.

Response:

The reviewer raises a critical point regarding the potential effects of genetic variation in the HLA class III (hereafter class III) region on plasma protein expression, and how these effects may affect our HLA-pQTL associations.

To address the reviewer's question, we first checked whether any of the 504 HLA-pGenes identified in our analysis also had significant SNP pQTL associations from the class III region, from the flagship pQTL paper. We identified 29 such genes that had at least significant class III pQTL.

For these 29 genes that have significant HLA-pQTL in our study and significant class III pQTL from the flagship study, we then investigated whether both the HLA-pQTL and class III-pQTL independently contribute to plasma protein expression.

Thus, for each of the 29 pGenes in the discovery cohort, we fit a combined pQTL model incorporating our lead HLA-pQTL, the significant class III SNP pQTL from the flagship study, and all clinical and demographic covariates. We display the results below, which show—for each of the 29 pGenes—the original p-values for our lead HLA-pQTL and class III pQTL, and the p-values for our lead HLA-pQTL and class III

pQTL in a combined model incorporating both the lead HLA-pQTL and class III pQTL.

We note to the reviewer that the TNF gene is shown four times on the below barplot—this is because TNF as measured separately in the 4 different Olink panels (cardiometabolic, inflammation, neurology, oncology; loosely corresponding to the relevance of each protein on each panel to the broad disease area in the panel name) which combine to form the full protein dataset, as described in the flagship paper.

Fig R2.1 (Extended Data fig. 3d of the revised manuscript): Sensitivity analysis adjusting for class III pQTL. For each of the 29 HLA-pGenes that had a significant class III SNP pQTL from the flagship analysis, we fit a combined model incorporating our lead HLA-pQTL together with the class III SNP pQTL and all clinical and demographic covariates. The bar plot depicts the p-values for our HLA-pQTL and the class III pQTL from their respective original analyses (our original manuscript, or the original flagship paper), and the p-values for our lead HLA-pQTL and the class III SNP pQTL from the combined model with both pQTLs. P-values are capped at 1e-200. Dotted line indicates genome-wide significance (5e-08).

We can divide the associations from the combined model into 4 categories—“both”, indicating that both our lead HLA-pQTL (dark red bar) and class III pQTL (dark blue bar) remain significant at $P < 5e-08$ in the combined model; “class III only”, in which only the class III pQTL remains significant in the combined model; “Classical HLA only”, in which only our lead HLA-pQTL remains significant in the combined model, and “neither”, in which neither our lead HLA-pQTL nor the class III pQTL remained significant in the combined model. Of each of the 29 pGenes, the breakdown into these 4 categories is as follows:

Combined model category	Number of pGenes (out of 29 total)
Both (both dark red and dark blue bars above dotted line)	14
Class III only (dark blue bar above dotted line, but not dark red bar)	8
Classical HLA only (dark red bar above dotted line, but not dark blue bar)	2
Neither (neither dark red nor dark blue bar above dotted line)	5

Thus, out of the 29 HLA-pGenes, for 16 of them (“Both” and “HLA allele” categories”) our lead HLA-pQTL remains significant after conditioning on the significant class III SNP pQTL; whereas 8 are driven primarily by class III. The 5 pGenes for which neither our HLA-pQTL nor the originally identified class III SNP pQTL remain significant perhaps reflect LD between the HLA-pQTL and the class III SNP pQTL.

Overall, these data show that both classical HLA-pQTL (alleles / amino acids in the classical HLA genes) and genetic variation in the class III can independently drive plasma protein expression. Furthermore, the data indicate that in general, our HLA-pQTL associations are not strongly affected by class III genetic variation, as only 8 out of the 504 pGenes are confirmed to be affected by class III variants, which are themselves part of the MHC region.

We have added these important data to the revised manuscript, and describe it in the revised text as follows:

“Moreover, we asked whether plasma protein expression levels were affected by genetic variants in the HLA class III region, which itself contains the complement genes and others associated with the immune response. Of our 504 pGenes, 29 of them had a significant class III SNP pQTL from the flagship pQTL study by Sun et al. Thus, we performed a sensitivity analysis, in which for each of these 29 pGenes we fit a combined model assessing the effect of both our lead HLA-pQTL and the class III SNP-pQTL (Extended Data 3d). This analysis revealed that expression of only 8 of the 29 pGenes were driven by class III compared to our lead HLA-pQTL, whereas 16 of the 29 were driven either primarily by our lead HLA-pQTL or both the lead HLA-pQTL and the class III SNP pQTL independently. For the remaining 5 pGenes, neither our original lead HLA-pQTL nor the class III SNP pQTL remained significant in the combined model, suggesting that the two signals may be in tight LD. Altogether, these data indicate that both imputed genetic variants from the classical HLA loci and class III SNPs can independently drive protein expression.”

3. The authors use of Bonferroni correction for conditional analyses to identify independent polymorphism associations. However given the limited region being tested, should there not be a less stringent p value. Could the threshold used, influence the number of truly “distinct” signals?

Response:

We thank the reviewer for this question and are happy to clarify the choice of significance thresholds in the analysis. While we acknowledge the reviewer’s point that perhaps a less stringent p-value could be used for the analyses, there are several reasons why we used the strict Bonferroni cutoff.

The first reason that we adopted these conservative thresholds is the extremely strong LD in the MHC region—this would particularly affect conditional analyses, which seek to find additional independent HLA variants. To investigate this point, we analyzed, for each pGene, the pairwise LD between all HLA variants (lead and conditional) associated with the pGene. Reassuringly this analysis showed that the pairwise LD was low for all the HLA-pQTL for a given pGene which may be due in part to the strict significance threshold adopted in our analysis.

b

Distribution of median R^2 computed for all lead and conditional HLA variants for each pGene (N = 305 pGenes with conditional variants)

For each pGene, median R^2 across lead and all conditional HLA-pQTL: 1-field alleles, 2-field alleles, amino acids

Fig R2.2 (Extended Data fig. 3b of the revised manuscript): For each of the 240 pGenes with at least one conditionally independent HLA variant, we calculated the pairwise R^2 between all variants associated with the pGene. We then calculated the median of these pairwise R^2 values, and plotted the distribution of these median values across all 240 pGenes. The data show that in general, the pairwise R^2 between all lead and conditionally independent HLA variants associated with each pGene is very low (median 0.019), suggesting that the total number of conditionally independent variants reported in the manuscript, and the enrichment of HLA-pQTL in the peptide binding groove, are likely not an artifact attributable to LD.

It is true that a limited region is being tested, and it could be argued that a less stringent threshold should be used—which could increase the number of conditional associations found. However, a less stringent threshold might also result in more spurious associations due to LD. Indeed, optimizing such a threshold—perhaps with extensive permutation analyses and validation in independent datasets—should be investigated comprehensively in a future methods study.

Secondly, in both our original and revised manuscript, we make several comparisons to the flagship SNP pQTL paper, which used Bonferroni thresholds throughout to identify significant pQTL. Adopting a different threshold in our study would complicate comparisons of our results to those of the flagship study. Indeed, as the reviewer has pointed out regarding class III and associations reported in the flagship paper, such comparisons are critical for the interpretation and novelty of our current HLA-pQTL study.

Finally, we adopted a Bonferroni threshold to remain as robust as possible to future developments in HLA imputation. While here we were able to test 2284 HLA alleles and amino acids, we elected not to use this number (2284) to pre-define a significance threshold, as improvements in reference panels and imputation methodology could increase the number of imputed HLA variants tested. As with our current study compared to the flagship study, significance cutoffs defined based on the number of HLA variants tested may complicate comparisons between future studies using different imputation strategies and our present study.

For these reasons, we feel it is best to keep the stringent Bonferroni cutoff as implemented in both our study and the flagship pQTL study.

4. While it is not entirely surprising to find immune-related proteins associated with the region, the manuscript could delve deeper into how this information can be leveraged to inform autoimmune disease pathology. Are there insights into the mechanistic links between genetic variations, protein expression, and autoimmune diseases? Can the study provide potential therapeutic targets? Are cis pqtls of genes identified linked in a more mechanistic manner to diseases associated with HLA alleles?

5. The manuscript could explore the potential for colocalization analysis, acknowledging any limitations in the context of this challenging region. This approach may offer additional insights into shared causal variants between disease risk associations and protein levels.

Response:

We thank the reviewer for these important questions, which represent an opportunity to more directly link our HLA-pQTL to HLA-associated diseases and explore the mechanism involved. Given the related nature of the reviewer's questions 4 and 5, we will address them together here.

First, the reviewer is correct to note the challenges of applying standard colocalization methods to the MHC, as also discussed in Kang *et al.* The limitations of standard SNP colocalization—which is effectively an overlap between fine-mapped SNP signals and fine-mapped SNP signals from a QTL GWAS—are similar to the limitations of applying standard fine-mapping methods to the MHC, as we also describe in response to reviewer 1. In general, the extreme polymorphism and LD of the MHC, coupled with the major assumption in HLA-trait association studies that coding variation drives the majority of the genetic heritability of a trait (borne out in many prior studies of HLA-disease associations), make standard fine-mapping and colocalization methods inappropriate for the MHC, which do not have the coding variant assumption. Moreover, most colocalization methods assume a single causal variant; while some methods do allow for more than one causal variant, such methods have not been tested or rigorously benchmarked on the MHC.

Despite these limitations, we sought to alternative ways to address the reviewer's question regarding colocalization. We reasoned that since we have fine-mapped HLA-pQTL using fine-mapping methods tailored to the MHC, we might overlap our results with any fine-mapped HLA-trait associations in the UK Biobank that also used the correct MHC fine-mapping methods. Encouragingly, the analysis in Sakaue *et al.* Nature Genetics 2021 did exactly this—the authors rigorously imputed HLA alleles and amino acids using the same methods as we do in our current study, and then ran association tests for the HLA imputed variants against diseases and biomarkers (e.g. red blood cell counts) in the UK Biobank.

Thus, we obtained these fine-mapped summary statistics and overlapped them with our HLA-pQTL fine-mapped data and displayed them in the following network visualization, also noting directions of effect. As this analysis overlaps fine-mapped signals from HLA-trait studies and HLA-pQTL, it effectively amounts to a colocalization between the two studies:

Fig. 4

Fig R2.3 (Fig. 4 of the revised manuscript): Integration of fine-mapped HLA-pQTL with UK Biobank fine-mapped HLA-trait signals Sakaue *et al.* Nature Genetics 2021.

This map in the revised manuscript allows us to visualize novel links between fine-mapped HLA genetic variants, proteins, diseases, and traits. For example, this analysis reveals that B_180_Q affects both CLEC4A, a C-type lectin receptor important for the innate immune response, and HDL cholesterol. Moreover, the DRB1_01_03 allele—which is thought to be a risk allele for ulcerative colitis (UC) based on prior GWAS—was associated with both UC and several inflammatory genes such as GZMA, GZMH, and IL10. Mechanistically, as the GZM genes are thought to be expressed on gut T and NK cells, our data may suggest that the HLA risk allele DRB1_01_03 modulates T and NK function to affect risk of UC. Moreover, the IL10 gene—also affected by DRB1_01_03—has long been considered a potential drug target for immune-mediated diseases²⁴. Thus, our data suggest that targeting IL10 may be particularly viable in individuals with the DRB1_01_03 UC risk allele.

To further explore potential therapeutic implications of our HLA-pQTL analyses, we obtained data from the druggable genome²⁸, which combines multiple lines of evidence to separate genes into tiers based on their potential druggability. In

particular, tier 1 genes are the targets of approved drugs or clinical-phase drug candidates obtained from the ChEMBL database. Using the Olink panel genes as background as we did previously for pathway enrichments, we asked whether our HLA pGenes are enriched in druggable genes. Indeed this was the case, regardless of whether we considered all tiers (e.g. including lower tiers defined by various molecular evidence for the target) or tier 1 approved targets:

Fig R2.4 (Extended Data fig. 8 of the revised manuscript): Enrichment of HLA-pGenes in druggable genes from the druggable genome study²⁸.

Among the overlap between our HLA-pGenes and Tier 1 targets were IL10, CD80/CD86 (targets of abatacept, used for the treatment of Rheumatoid Arthritis), and the well known cancer tyrosine kinase MERTK—thus, our analysis may motivate consideration of the use of drugs targeting these genes in individuals with predisposing HLA alleles. Further mechanistic hypotheses enabled by the integration of our HLA-pQTL with HLA-disease data are the effect of HLA B position 114 DK on the CD1C gene and rheumatoid arthritis, as shown above. These data may implicate dendritic cells as the primary antigen presenting cell in the pathogenesis of RA, as CD1C is a canonical lineage marker for conventional dendritic cells.

Thus, the map created by integration of our HLA-pQTL with HLA-disease data enables extensive hypothesis generation and reveals novel potential links between HLA effects on proteins and HLA effects on traits (i.e. the effect on HDL cholesterol described above).

Finally, we sought to address the reviewer's interesting question regarding whether cis-pQTLs of the pGenes identified in our study are linked in a more mechanistic way to HLA-associated diseases. We interpreted this question as being conceptually

similar to our analysis in the original manuscript, in which we asked whether HLA-pGenes also harbor trait-associated coding variation as obtained from the genebase dataset (which associates exome mutations with disease risk in the UK Biobank).

To explore this question systematically, we first obtained the cis-pQTLs for our HLA-pGenes from the flagship pQTL study. We next asked whether these cis SNP pQTLs are themselves associated with HLA-associated diseases by querying the GWAS catalog. We detected 5 HLA-pGenes (CD5, CD6, IL7R, SLAMF8, TNFSF11) with a cis-pQTL associated ($P < 5e-08$) with diseases previously associated with HLA genetic variation:

cis-pQTL from flagship study	HLA-pGene from our study	GWAS catalog study ID	Disease
rs4939490	CD5	GCST000949	Multiple Sclerosis
rs4939490	CD5	GCST009597	Multiple Sclerosis
rs11230563	CD6	GCST005537	Chronic inflammatory diseases
rs11230563	CD6	GCST004131	Inflammatory bowel disease
rs11230563	CD6	GCST003043	Inflammatory bowel disease
rs11230563	CD6	GCST001725	Inflammatory bowel disease
rs11230563	CD6	GCST003044	Crohn's disease
rs11230563	CD6	GCST003045	ulcerative colitis
rs4129267	IL6R	GCST005537	Chronic inflammatory diseases
rs4129267	IL6R	GCST005529	Ankylosing spondylitis
rs6897932	IL7R	GCST001198	Multiple sclerosis
rs6897932	IL7R	GCST90270940	Systemic lupus erythematosus (MTAG)
rs34687326	SLAMF8	GCST004132	Crohn's disease
rs2062305	TNFSF11	GCST000879	Crohn's disease

Fig R2.5 (Extended Data Fig. 10a of the revised manuscript): cis-pQTLs of pGenes identified in our study and their association with HLA-associated diseases.

CD5, CD6, and IL7R were abundantly expressed on T and myeloid cells in the Yazar et al. dataset; IL7R in particular is a critical marker for both CD4+ and CD8+ T cells. Prior studies have implicated CD5 in T cell activation and Th17 cell differentiation^{29,30}; similarly, CD6 is thought to have important roles in T cell development in the thymus and the generation of regulatory T cells³¹. However, SLAMF8 and TNFSF11 were not expressed on the immune cell atlas, providing further evidence that HLA-pQTL affects both immune and non-immune mechanisms.

Fig R2.6 (Extended Data Fig. 10b of the revised manuscript): Expression on the Yazar *et al.* immune cell atlas of HLA-pGenes with cis SNP pQTLs from the flagship study significantly associated with HLA-associated immune diseases.

We have added a detailed description of these data to the results section of the revised manuscript as follows:

“We sought to more systematically explore the relationships between HLA-pQTL and HLA associations with disease. Standard colocalization methods are limited by the extreme polymorphism and LD of the MHC; in addition, a major assumption of HLA-trait association studies is that coding variation drives the majority of the genetic heritability. Moreover, most colocalization methods assume a single causal variant; while some methods do allow for more than one causal variant, such methods have not been tested or rigorously benchmarked on the MHC.”

*Despite these limitations, we obtained summary statistics for fine-mapped HLA-disease summary statistics from the UK Biobank from Sakaue *et al.*, and overlapped them with our fine-mapped HLA-pQTL (Fig. 4, Supplementary Table 20). Specifically, Sakaue *et al.* performed single-marker tests with their HLA imputed variants; thus, we overlapped all of our single-marker lead and conditional HLA-pQTL with the single-marker summary statistics from Sakaue *et al.* We reasoned that since this analysis represents an overlap between two sets of fine-mapped summary statistics, it effectively amounts to a colocalization between HLA-pQTL and HLA-disease signals. Notably, the analysis by Sakaue *et al.* analyzed HLA*

associations with both diseases and quantitative traits (e.g. HDL cholesterol) in the UK Biobank.

The integration of HLA-pQTL and HLA-disease summary statistics allows us to visualize novel links between fine-mapped HLA genetic variants, proteins, diseases, and traits. For example, this analysis reveals that HLA-B position 180_Q affects both CLEC4A, a C-type lectin receptor important for the innate immune response, and HDL cholesterol. The DRB1*01:03 allele—which is thought to be a risk allele for ulcerative colitis (UC) based on prior GWAS⁴¹—was associated with both UC and several inflammatory genes such as GZMA, GZMH, and IL10. Mechanistically, as the GZM genes are thought to be expressed on gut T and NK cells, our data may suggest that the HLA risk allele DRB1*01:03 modulates T and NK function to affect risk of UC. Moreover, the IL10 gene—also affected by DRB1*01:03—has long been considered a potential drug target for immune-mediated diseases⁴². Thus, our data suggest that targeting IL10 may be particularly viable in individuals with the DRB1*01:03 UC risk allele.

To further explore potential therapeutic implications of our HLA-pQTL analyses, we obtained data from the druggable genome⁴³, which combines multiple lines of evidence to separate genes into tiers based on their potential druggability. In particular, tier 1 genes are the targets of approved drugs or clinical-phase drug candidates obtained from the ChEMBL database. Using the Olink panel genes as background as we did previously for pathway enrichments, we asked whether our HLA pGenes are enriched in druggable genes⁴³. Indeed this was the case, regardless of whether we considered all tiers (e.g. including lower tiers defined by various molecular evidence for the target) or tier 1 approved targets (**Extended Data Fig. 8b**).

Among the overlap between our HLA-pGenes and Tier 1 targets were IL10, CD80/CD86 (targets of abatacept, used for the treatment of Rheumatoid Arthritis), and the well-known cancer tyrosine kinase MERTK (**Supplementary Table 21**). Thus, our analysis may motivate consideration of the use of drugs targeting these genes in individuals with predisposing HLA alleles. Further mechanistic hypotheses enabled by the integration of our HLA-pQTL with HLA-disease data are the effect of HLA B position 114 DK on the CD1C gene and rheumatoid arthritis, as shown above. These data may implicate dendritic cells as the primary antigen presenting cell in the pathogenesis of RA, as CD1C is a canonical lineage marker for conventional dendritic cells.

Finally, we asked whether cis-pQTLs of the HLA-pGenes identified in our analysis are themselves linked to diseases commonly associated with HLA genetic variation. To explore this question systematically, we first obtained the cis-pQTLs for our HLA-pGenes from the flagship pQTL study. We next asked whether these cis SNP pQTLs are themselves associated with HLA-associated diseases by querying the GWAS catalog. We detected 5 HLA-pGenes (CD5, CD6, IL7R, SLAMF8, TNFSF11) with a cis-pQTL associated with diseases previously associated with HLA genetic variation (**Extended Data Fig. 10a**)—CD5, CD6, and IL7R were abundantly expressed on T and myeloid cells in the Yazar et al¹⁹. dataset (**Extended Data Fig. 10b**); IL7R in particular is a critical marker for both CD4+ and CD8+ T cells. Prior

studies have implicated CD5 in T cell activation and Th17 cell differentiation^{44,45}; similarly, CD6 is thought to have important roles in T cell development in the thymus and the generation of regulatory T cells⁴⁶. However, SLAMF8 and TNFSF11 were not expressed on the immune cell atlas (**Extended Data Fig. 10b**), providing further evidence that HLA-pQTL affects both immune and non-immune mechanisms. Altogether, these data highlight the importance and potential mechanisms underlying HLA-pQTL associations with disease.”

6. Is there expression data that support the observed associations between genetic variants and plasma protein levels?

Response:

The reviewer raises an intriguing question regarding potentially supportive gene expression data. We expect that the reviewer is asking whether there are summary statistics of trans HLA-eQTL mapped in a manner similar to the approach we have taken here. Unfortunately, no such data exist; indeed, the existing excellent eQTL resources such as GTEx or the eQTL catalog are much smaller than the UK Biobank and do not perform HLA imputation and fine-mapping. While a recent important study mapped the effects of cis regulatory genetic variants on HLA gene expression¹², the goal of this study was cis effects on the HLA genes themselves in single cell RNA-seq cohorts (which are not population-scale in size, as the UK Biobank is), rather than assessing the effects of imputed HLA alleles and amino acids on trans protein expression, which is the focus of our study.

However, motivated by the reviewer’s question, we considered alternative ways that we could use gene expression data to support and extend our manuscript. First, we leveraged tissue-specific gene expression gene sets from Uhlen *et al.* to ask whether the HLA pGenes detected in our study are enriched in genes that are specific to particular tissues²⁰. In the Uhlen *et al.* study, the authors performed RNA-seq analysis of 32 tissues and performed additional analyses to determine which genes are specific to particular tissues. This analysis revealed enrichment of HLA-pGenes in spleen, lymph node, tonsil, appendix, bone marrow, and lung, some of which (spleen, lymph node, tonsil, bone marrow, and lung in particular) are thought to be highly immune-infiltrated tissues.

Enrichment of HLA-pGenes in tissue-specific gene sets from the Human Protein Atlas / GTEx

Fig R2.7 (Extended Data fig. 8a of the revised manuscript): Enrichment of HLA-pGenes in tissue-specific gene sets determined from RNA-seq of 32 tissues from Uhlen *et al.* Science 2015. Two-sided p-values calculated using Fisher's exact test.

Motivated by this result, the identity of the pGenes themselves (i.e. many chemokines and immune-related genes), and the pathway analyses we presented in the original manuscript, we next asked to what extent our HLA-pGenes are transcriptionally expressed on peripheral blood immune cells. To explore this question, we obtained a doublet-cleaned version of the Yazar *et al.* scRNA-seq atlas—comprised of 909 donors and 735,000 PBMC immune cells—from Kang *et al.* We reprocessed these data according to the methods described in Kang *et al.*, and used it to explore expression patterns of HLA-I-specific and HLA-II-specific pGenes. Using methods outlined in Kang *et al.*, we uniformly re-processed the Yazar *et al.* dataset, plotting some markers from the original publication and new markers from

our re-analysis to provide confidence in the cell type names assigned to each cluster, as shown below:

Fig R2.8 (Fig. 3a,b of the revised manuscript): Re-processed PBMC immune cell atlas originally described in Yazar *et al.* according to methods described in Kang *et al.* a, UMAP of cell types and cluster assignments. b, Selected differentially expressed genes defining clusters in a.

Using this resource, we asked whether the HLA-I-specific or HLA-II-specific pGenes we identified in the manuscript have varying expression across different immune lineages (e.g. T cells vs myeloid cells). We were motivated to do this based on the biology of HLA-I vs HLA-II—HLA-I is generally expressed on all nucleated cells, whereas HLA-II is expressed primarily on antigen presenting cells (APCs) such as myeloid cells and B cells. We were further motivated to conduct this analysis given our own results, which identified pGenes specifically affected by HLA-I compared to HLA-II and vice versa—i.e. if there are different pGenes affected, are there also different cell types?

We plotted all of the HLA-I-specific and HLA-II-specific pGenes on the re-processed Yazar *et al.* atlas, and added a color bar along the rows of the heatmap to indicate whether the gene is HLA-I-specific or HLA-II-specific. Curiously, this analysis showed that despite the fact that there are pGenes specific to HLA-I or HLA-II, the cell type expression patterns are largely overlapping between the sets of pGenes (i.e., no clustering by purple / orange along the rows (genes) of the heatmap). This result is especially intriguing given the well-known, more specific expression and activity of HLA-II on APCs²²—i.e. despite this specific expression, HLA-II genetic variants affect expression of genes on non-APCs (i.e. T cells) as well.

Fig R2.9 (Fig. 3c of the revised manuscript): Mean expression of HLA-I-specific (purple) and HLA-II-specific (orange) pGenes across clusters identified in the Yazar *et al.* immune cell atlas. Heatmap is clustered across rows and columns; dendrograms are omitted from the plot for clarity. The data show that HLA-I and HLA-II-specific pGenes are broadly expressed across immune lineages (i.e. no obvious clustering of purple/orange along the rows).

We next sought to explore the expression patterns of HLA-pGenes on non-immune cells, and compare them to expression on immune cells. Given the enrichment of HLA-pGenes on lung-specific genes, we obtained a lung cell scRNA-seq atlas from

Travaglini *et al.*, and used it to plot HLA-I-specific and HLA-II-specific pGenes as we did with the immune cell atlas from Yazar *et al.* Importantly, we chose this dataset because it is comprised of normal lung unaffected by disease, and has excellent representation of both immune cells and non-immune cells. We used cluster definitions exactly as specified by the Travaglini *et al.* study. While we acknowledge that there exist larger lung scRNA-seq atlases (e.g. the HLCA effort by Sikemma *et al.* Nature Medicine 2023), that dataset developed its cell type annotations on a shared embedding with both healthy and diseased lung, whereas the Travaglini dataset focuses exclusively on normal lung, unaffected by disease. This difference is important, as the HLA-pQTL we detected in the UK Biobank were primarily from healthy donors.

As observed in the Yazar *et al.* dataset, we observed no clear clustering between HLA-I-specific and HLA-II-specific pGenes, but we did detect pronounced expression of HLA-pGenes on non-canonical immune cells, e.g. in fibroblasts and lung epithelial cells. The expression of pGenes on these on these non-immune cells was comparable to pGene expression on immune cells. The pGenes we identified were found using peripheral blood data, and not lung tissue—as such, we caution against overinterpretation of these data. However, the data do suggest that pGenes affected by HLA genetic variants may be expressed comparably in non-immune cells compared to immune cells.

Fig R2.10 (Extended Data Fig. 9 of the revised manuscript): Mean expression of HLA-I-specific (purple) and HLA-II-specific (orange) pGenes across clusters identified in the Travaglini *et al.* lung cell atlas. Heatmap is clustered across rows and columns; dendrograms are omitted from the plot for clarity. The data show that HLA-I and HLA-II-specific pGenes are broadly expressed across immune lineages (i.e. no obvious clustering of purple/orange along the columns). Furthermore, the data show that pGene expression in non-canonical immune cells, such as fibroblasts, goblet cells, and alveolar epithelial cells.

Altogether, these data—spanning tissue-specific gene sets and two scRNA-seq atlases comprising both immune and non-immune cells—add depth to the

interpretation of HLA-pGenes, suggest that HLA-I-specific and HLA-II-specific pGenes converge on shared cell types, and that HLA-pGenes can be expressed on both immune and non-immune cells. More generally, our data motivate future single cell *trans* HLA-pQTL studies to more formally investigate the myriad effects of HLA genetic variation on gene expression in individual cell types.

We have added a detailed description of these data to the results section of the revised manuscript as follows:

“To explore the expression patterns of pGenes identified in our analysis, we first asked whether our pGenes were enriched in tissues-specific gene sets developed from gene expression data from Uhlen et al. (Extended Data Fig. 8a). This analysis revealed enrichment of HLA-pGenes in spleen, lymph node, tonsil, appendix, bone marrow, and lung, which are thought to be extensively immune infiltrated.

Motivated by this result, we next asked to what extent our HLA-pGenes are transcriptionally expressed on peripheral blood immune cells. To explore this question, we obtained a doublet-cleaned version of the Yazar et al. scRNA-seq atlas—comprised of 909 donors and 735,000 PBMC immune cells—from Kang et al. We reprocessed these data according to the methods described in Kang et al., and used it to explore expression patterns of HLA-I-specific and HLA-II-specific pGenes (Fig. 3a,b). We plotted all of the HLA-I-specific and HLA-II-specific pGenes on the re-processed Yazar et al. atlas (Fig. 3c). Curiously, this analysis showed that despite the fact that there are pGenes specific to HLA-I or HLA-II, the cell type expression patterns are largely overlapping between the sets of pGenes. This result is especially intriguing given what is known about the more specific expression of the HLA-II genes on APCs—i.e. despite this specific expression, HLA-II genetic variants affect expression of genes on non-APCs (i.e. T cells) as well.

We next sought to explore the expression patterns of HLA-pGenes on non-immune cells, and compare them to expression on immune cells. Given the enrichment of HLA-pGenes on lung-specific genes, we obtained a healthy lung cell scRNA-seq atlas from Travaglini et al., and used it to plot HLA-I-specific and HLA-II-specific pGenes as we did with the immune cell atlas from Yazar et al. Importantly, we chose this dataset because it is comprised of normal lung unaffected by disease, and has excellent representation of both immune cells and non-immune cells. We used cluster definitions exactly as specified by the Travaglini et al. study.

*As observed in the Yazar et al. dataset, we observed no clear clustering between HLA-I-specific and HLA-II-specific pGenes, but we did detect pronounced expression of HLA-pGenes on non-canonical immune cells, e.g. in fibroblasts and lung epithelial cells (Extended Data Fig. 9). The expression of pGenes on these non-immune cells was comparable to pGene expression on immune cells. The pGenes we identified were found using peripheral blood data, and not lung tissue—as such, we caution against overinterpretation of these data. However, the data do suggest that pGenes affected by HLA genetic variants may be expressed comparably in non-immune cells compared to immune cells. More generally, our data motivate future single cell *trans* HLA-pQTL studies to more formally investigate the myriad effects of HLA genetic variation on gene expression in individual cell types.*

Examination of individual pGenes identified in our analysis—particularly among the strongest associations—corroborated our interpretation of the scRNA-seq data. SFTPD is generally thought of as a pattern recognition receptor, and thus might be considered a part of the innate immune response. The identification of this pGene is surprising and novel in the context of HLA genetics, which—through their central role in antigen presentation and recognition by T cells—might be thought to primarily affect T cell-related pathways. LRPAP1 is the LDL receptor-related protein 1; aside from a recent report suggesting that LRPAP1 may facilitate immune evasion from viruses, it is both unclear and intriguing to speculate as to how LRPAP1 expression may be affected by HLA genetics. ENPP6 is an enzyme involved in choline metabolism; NPTX1 is a member of the pentraxin gene family with reported roles in endoplasmic reticulum stress, and SGSH encodes the enzyme N-sulfoglucosamine sulfohydrolase, with no known roles in immunity. Altogether, these data highlight the use of our HLA-pQTL associations to discover novel relationships between HLA genetic variation and proteins with unclear connections to the immune response.

Finally, we considered a third use of gene expression data—specifically, eQTL data—to explore the relative contribution of pQTL and eQTL to plasma protein expression. Specifically, we asked whether cis-SNP eQTLs (obtained from the GTEx whole blood cohort) of the genes encoding our HLA-pGenes also drive expression of the corresponding protein, and whether such a signal might account for the observed HLA-pQTL effects reported in our manuscript.

Thus, for each HLA-pGene identified in our analysis, we looked up the corresponding gene in the GTEx whole blood eQTL summary statistics. We then obtained all significant eQTLs for the gene, and checked to see whether those variants were present in the UKB data. 266 of the 504 HLA-pGenes had at least one significant eQTL in GTEx and had their SNP eQTLs represented in the UK Biobank cohort.

Then, similar to our class III conditioning analyses presented in response to the reviewer's earlier question, for each of the 266 pGenes we fit a combined model together with our lead HLA-pQTL and the GTEx eQTLs for that pGene. Notably, this analysis revealed that of the 266 pGenes, 265 remained significant in this analysis at $P < 5 \times 10^{-8}$. The only pGene that lost significance in this analysis was BTN3A2, which is within the MHC region.

Moreover, for 157 of the pGenes, both the HLA-pQTL and cis-eQTLs remained significant in the combined models. Thus, these data suggest that there are no long-range confounding effects between HLA-pQTL and cis-eQTLs of our pGenes, and furthermore, that both HLA-pQTL and cis-eQTL can independently contribute to plasma protein expression.

The summary statistics of this analysis are included as a supplementary table in the revised manuscript, and described in the revised text as follows:

"In addition, we sought to explore whether both our HLA-pQTL and cis-eQTLs of the HLA-pGenes identified in our study independently contribute to protein expression.

For 266 of our 504 HLA-pGenes, we detected significant cis-eQTLs from the GTEx whole blood cohort that were also present in the UK Biobank. For each of these 266 pGenes we fit a combined model of plasma protein expression including both the lead HLA-pQTL and the cis-eQTLs for the corresponding gene from GTEx. Notably, this analysis revealed that of the 266 pGenes, 265 HLA-pQTL remained significant in this analysis at $P < 5 \times 10^{-8}$ (**Supplementary Table 18**). The only pGene that lost significance for HLA-pQTL in this analysis was *BTN3A2*, which is within the MHC region. Moreover, for 157 of the pGenes, both the HLA-pQTL and cis-eQTLs remained significant in the combined models (**Supplementary Table 18**). Thus, these data suggest that there are no long-range confounding effects between HLA-pQTL and cis-eQTLs of our pGenes, and furthermore, that both HLA-pQTL and cis-eQTL can independently contribute to plasma protein expression.”

Altogether, we believe that the gene expression analyses presented in our revised manuscript- spanning multiple scRNA-seq datasets and pQTL/eQTL conditional analyses—have greatly expanded the mechanistic explorations and robustness of the manuscript.

Reviewer 3:

Reviewer #3 (Remarks to the Author):

The manuscript titled "Exploring the Impact of HLA Genetic Variation on Plasma Protein Expression" investigates the influence of HLA genetic variation on the expression of 2940 plasma proteins by employing imputed HLA variants in a cohort of 45,330 Europeans from the UK Biobank. The study, which generates an atlas of HLA-pQTL, is intriguing; however, the presented level of novelty does not meet the publication standards of Nature Communications. I have several major comments for the authors to consider:

We thank the reviewer for their constructive reading of our manuscript and appreciate the reviewer's comment regarding the novelty of our work. Motivated by suggestions from this reviewer and others, in our revised manuscript we have now added new datasets and analyses to bolster the novelty of the work, including two single cell RNA-seq datasets exploring the expression patterns of HLA-pGenes in both immune and non-immune cell types, integration of fine-mapped HLA-pQTL with fine-mapped HLA-disease summary statistics from the UK Biobank, pathway analyses of non-HLA-pQTL, additional exploration of the mechanistic underpinnings of HLA-pQTL associations with disease, and enrichment of HLA-pQTL in known drug targets.

Moreover, a critical analysis suggested by the reviewer is the sensitivity analysis removing individuals with immune-mediated diseases. Encouragingly, we find that our HLA-pQTL effects are robust to the removal of these individuals. All of these data are now included in the revised manuscript, comprising two additional main figures and multiple additional extended data figures.

More generally, our manuscript represents a thorough investigation of the effect of imputed HLA genetic variants on plasma protein expression at unprecedented scale in the UK Biobank, discovering hundreds of trans associations, in addition to multiple genomic follow-up analyses. Taken together, we anticipate that our manuscript will serve as an important resource both for future efforts to map HLA effects on quantitative traits, and to motivate biological hypothesis generation to discover new molecular effects of HLA genetic variation.

We thank the reviewer again for their questions and present a point-by-point response to their questions below:

1. The manuscript reports that HLA-pQTL accounts for over 20% of the protein expression variance. It would be valuable to understand how much variance is attributed to non-HLA pQTL, and whether these non-HLA variants operate in the same pathways as their corresponding HLA genes.

Response:

We thank the reviewer for this question, which represents an intriguing opportunity to explore the pathways enriched in non-HLA pQTL.

First, to address the reviewer's question regarding variance attributed to non-HLA pQTL, we obtained summary statistics and heritability estimates from the flagship pQTL study from Sun *et al.* For each pGene detected our analysis, we plotted the variance explained by genome-wide SNP pQTLs (cis + trans) against variance explained by our HLA-pQTL. This analysis revealed that while non-HLA pQTL can explain up more than 80% of the protein expression variance, the estimates from non-HLA compared to our HLA-pQTL were weakly positively correlated.

We have included this figure in Extended Data Fig. 3f and describe it in the revised manuscript as follows:

Fig R3.1 (Extended Data fig. 3f of the revised manuscript): Proportion of protein expression variance explained by lead and conditional HLA-pQTL

compared to genome-wide trans SNP pQTL from the flagship UKB-PPP analysis

“Comparing the variance explained by HLA-pQTL to genome-wide pQTL from the flagship UKB-pQTL study, we found that for several pGenes (e.g. LRP1, LRPAP1, SGSH, KIR), much of the phenotypic variance explained by trans genetic variation is accounted for by HLA genetic variation”

The reviewers’ second question regarding enrichment of non HLA-pQTL is intriguing—motivated by the reviewer’s question, we considered two hypotheses. First, for the 504 pGenes identified in the study, it is possible that genes near non HLA trans-pQTL are enriched in pathways related to the HLA genes or the general adaptive immune response. Such a result would argue that genetic variation in the immune response, in general, may affect protein expression. Alternatively, we considered that the pathways enriched in non HLA-pQTL are *not* related to immunity or the HLA. This would imply that genetic effects are complex, and perhaps separate into distinct non-immune pathways (e.g. non HLA-pQTL) and HLA effects (our HLA-pQTL).

To explore these hypotheses, we obtained all trans, non HLA-pQTL from the flagship pQTL study, corresponding to pGenes identified in our study (495 / 504 proteins overlapping). The flagship study also provided the genes proximal to these pQTL SNPs. We then conducted a pathway analysis (gene ontology; GO enrichment) on these 495 genes using the Olink panel genes (N = 2940) as a background.

While this analysis did not yield any pathways that were significant after FDR adjustment for multiple tests, we depict below that pathways that were significant at unadjusted $P < 0.05$. Curiously, none of these pathways were directly related to the immune response, except for “natural killer cell lectin-like receptor binding”; which, upon closer inspection, was defined by the genes MICA/MICB, which are in the extended MHC region (though not classical HLA loci themselves).

Fig R3.2 (Extended Data fig. 11 of the revised manuscript): Gene ontology enrichment analysis depicting pathways enriched in trans, non HLA-pQTL at unadjusted $P < 0.05$.

This analysis may suggest that both HLA genetic variation and non-immune-related genetic variation act independently to influence the expression of the pGenes

detected in our study. However, we cannot rule out the possibility that perhaps the genes corresponding to these non HLA-pQTL are related to the immune response in a complex, indirect manner. For this reason, we present this analysis as exploratory in the discussion section of the revised manuscript, and include the pathway enrichments / genes defining the pathways as an additional supplementary table.

“Our study reports a wide range of variance in protein expression explained by HLA-pQTL. Given the trans SNP pQTL reported by the flagship Sun et al. study, an intriguing question is whether HLA effects on protein expression act in similar pathways to trans pQTL outside the HLA locus. We performed an exploratory analysis investigating the pathways enriched in the genes proximal to these non-HLA trans pQTL (Extended Data Fig. 11)- curiously, no pathways reached statistical significance after adjusting for multiple tests, and only one—natural killer cell lectin-like receptor binding—was related to the immune response. This analysis may suggest that both HLA genetic variation and non-immune-related genetic variation act independently to influence protein expression. However, we urge caution in over-interpretation of the results, as we cannot rule out the possibility that genes containing non-HLA trans-pQTL are indirectly related to the immune response or the HLA genes themselves through currently unknown mechanisms.”

2. Considering the substantial role of HLA in immune-mediated diseases, it is important to know the number of samples in the cohort with autoimmune or autoinflammatory diseases. If a significant portion of samples exhibit such conditions, the authors should address whether excluding these samples would impact the results.

Response:

The reviewer raises a critical point regarding the effects of HLA genetic variation on immune-mediated diseases, and how these may affect the HLA-pQTL results reported here. In our original manuscript, we included principal components of protein expression to account for potential disease effects (Extended Data Fig. 1b). However, the reviewer’s question motivated us to perform a sensitivity analysis addressing potential disease effects more directly.

To address the reviewer’s question, we first carefully considered how to assemble the list of diseases for which we would exclude patients. We identified patients with diagnoses or deaths (ICD10) due to the major autoimmune diseases previously linked to HLA genetic variation as depicted in Table 1 of the manuscript. To address the reviewer’s question rigorously, we also excluded patients with any of the diseases associated with HLA genetic variation from the UK Biobank HLA analysis in Sakaue *et al.* Nature Genetics 2021.

In the discovery cohort, we found 5115 individuals with a confirmed diagnoses of any HLA-associated diseases. We thus excluded these 5115 individuals from the discovery cohort and re-tested the lead HLA-pQTL for each of the 504 pGenes.

Encouragingly, of the 504 pGenes identified originally, 503 remained significant at $P < 5 \times 10^{-8}$ when removing individuals with HLA-associated diseases. The one gene that lost formal significance in this analysis (CST7) trended towards significance ($P = 6.19 \times 10^{-7}$, and was just above the significance threshold in the original analysis (CST $P = 1.00 \times 10^{-7}$ in full discovery cohort; threshold for significance $P = 1.70 \times 10^{-11}$).

In addition, the effect sizes between lead HLA-pQTL from the full discovery cohort compared to the discovery cohort without HLA-associated diseases were strongly correlated:

Fig R3.3 (Extended Data fig. 3e of the revised manuscript): Sensitivity analysis removing individuals with HLA-associated immune-mediated diseases and HLA-associated diseases from Sakaue *et al.* Nature Genetics 2021. Concordance of effect sizes for lead HLA-pQTL between the full discovery cohort (original analysis) and discovery cohort without individuals with HLA-associated diseases.

We have added the summary statistics for lead HLA-pQTL in the reduced discovery cohort as supplementary table 4 in the revised manuscript, and have added a description of this sensitivity analysis to the results section of the revised manuscript as follows:

*“Given the strong associations of HLA genetic variation with immune-mediated diseases, we explored whether removing individuals with HLA-associated diseases—either major autoimmune diseases identified from prior bespoke GWAS, or HLA-associated diseases identified in an analysis of the UK Biobank by Sakaue *et al.*—impacted the discovery of the 504 lead HLA-pQTL in the discovery cohort. We identified 5021 individuals with diagnosis or death due to immune diseases (identified via ICD10 codes) in the UKB discovery cohort. We removed these individuals from the discovery cohort and repeated the analysis. We found that of the 504 pGenes discovered initially, 503 remained significant at $P < 5 \times 10^{-8}$; the remaining pGene (CST7) trended towards significance in this sensitivity analysis*

(6.19 x 10⁻⁷) (Supplementary Table 4). Effect sizes were also strongly correlated between the full discovery cohort and discovery cohort without HLA-associated diseases (Extended Data Fig. 3e). These data suggest that the HLA-pQTL reported in our study are not confounded by the presence of HLA-associated diseases in the UK Biobank.”

3. While the authors adjust for HLA-DRB1 and observe that the majority of HLA-I lead pQTL remains significant, the rationale for selecting HLA-DRB1 is not explicitly justified. Although its variants are associated with the largest number of proteins, it is unclear how this choice is supported.

Response:

We thank the reviewer for this question and are happy to provide additional context for the analysis conditioning on all HLA-DRB1 alleles. Indeed, we were struck by the fact that HLA-DRB1 harbored the most associations. In addition, genetic variation in HLA-DRB1 has been linked to many autoimmune diseases and more recently, TCR repertoire diversity. These reasons motivated us to perform the HLA-DRB1 conditional analysis, though we acknowledge the reviewer’s point that there is not formal support for the analysis.

For this reason, we have now re-worded the text as written below to clarify that this analysis is exploratory. If the reviewer prefer, we are also happy to remove this analysis from the final manuscript.

Whereas before we wrote:

“Notably, HLA-DRB1 variants were associated with the largest number of proteins. Prior studies have shown that alleles and fine-mapped amino acid positions in HLA-DRB1 drive risk of autoimmune diseases^{6,8} and affect the TCR repertoire¹⁹. We found that after adjusting for all 2-field HLA-DRB1 alleles, the majority of HLA-I lead pQTL remained significant...”

We now write in the revised manuscript:

“Notably, HLA-DRB1 variants were associated with the largest number of proteins. Prior studies have shown that alleles and fine-mapped amino acid positions in HLA-DRB1 drive risk of autoimmune diseases^{6,8} and affect the TCR repertoire¹⁹. Motivated by the extensively documented effects of HLA-DRB1 genetic variation on disease risk and the TCR repertoire, we performed an exploratory analysis in which we conditioned each lead HLA-pQTL on all 2-field HLA-DRB1 alleles. Notably, we found that after adjusting for all 2-field HLA-DRB1 alleles, the majority of HLA-I lead pQTL remained significant...”

References

1. Mbatchou, J. *et al.* Computationally efficient whole-genome regression for quantitative and binary traits. *Nat. Genet.* **53**, 1097–1103 (2021).
2. Sun, B. B. *et al.* Plasma proteomic associations with genetics and health in the UK Biobank. *Nature* **622**, 329–338 (2023).
3. Eldjarn, G. H. *et al.* Large-scale plasma proteomics comparisons through genetics and disease associations. *Nature* **622**, 348–358 (2023).
4. Sakaue, S. *et al.* A statistical genetics guide to identifying HLA alleles driving complex disease. *bioRxiv* 2022.08.24.504550 (2022).
doi:10.1101/2022.08.24.504550
5. de Bakker, P. I. W. & Raychaudhuri, S. Interrogating the major histocompatibility complex with high-throughput genomics. *Hum. Mol. Genet.* **21**, R29-36 (2012).
6. Raychaudhuri, S. *et al.* Five amino acids in three HLA proteins explain most of the association between MHC and seropositive rheumatoid arthritis. *Nat. Genet.* **44**, 291–296 (2012).
7. Hu, X. *et al.* Additive and interaction effects at three amino acid positions in HLA-DQ and HLA-DR molecules drive type 1 diabetes risk. *Nat. Genet.* **47**, 898–905 (2015).
8. Fellay, J. *et al.* A whole-genome association study of major determinants for host control of HIV-1. *Science* **317**, 944–947 (2007).
9. Sakaue, S. *et al.* A cross-population atlas of genetic associations for 220 human phenotypes. *Nat. Genet.* **53**, 1415–1424 (2021).
10. Tian, C. *et al.* Genome-wide association and HLA region fine-mapping studies identify susceptibility loci for multiple common infections. *Nat. Commun.* **8**, 599 (2017).
11. Luo, Y. *et al.* A high-resolution HLA reference panel capturing global population diversity enables multi-ancestry fine-mapping in HIV host response. *Nat. Genet.* **53**, 1504–1516 (2021).
12. Kang, J. B. *et al.* Mapping the dynamic genetic regulatory architecture of HLA genes at single-cell resolution. *Nat. Genet.* 2023.03.14.23287257 (2023).
doi:10.1101/2023.03.14.23287257
13. Patsopoulos, N. A. *et al.* Fine-Mapping the Genetic Association of the Major Histocompatibility Complex in Multiple Sclerosis: HLA and Non-HLA Effects. *PLOS Genet.* **9**, e1003926 (2013).
14. Sakaue, S. *et al.* Tutorial: a statistical genetics guide to identifying HLA alleles driving complex disease. *Nat. Protoc.* **18**, 2625–2641 (2023).
15. Raulet, D. H., Gasser, S., Gowen, B. G., Deng, W. & Jung, H. Regulation of ligands for the NKG2D activating receptor. *Annu. Rev. Immunol.* **31**, 413–441 (2013).
16. Ferrari de Andrade, L. *et al.* Antibody-mediated inhibition of MICA and MICB shedding promotes NK cell-driven tumor immunity. *Science* **359**, 1537–1542 (2018).

17. Lawson, P. R. & Reid, K. B. M. The roles of surfactant proteins A and D in innate immunity. *Immunol. Rev.* **173**, 66–78 (2000).
18. Li, H. *et al.* Secreted LRPAP1 binds and triggers IFNAR1 degradation to facilitate virus evasion from cellular innate immunity. *Signal Transduct. Target. Ther.* **8**, 374 (2023).
19. Coutelier, M. *et al.* NPTX1 mutations trigger endoplasmic reticulum stress and cause autosomal dominant cerebellar ataxia. *Brain* **145**, 1519–1534 (2022).
20. Uhlén, M. *et al.* Proteomics. Tissue-based map of the human proteome. *Science* **347**, 1260419 (2015).
21. Yazar, S. *et al.* Single-cell eQTL mapping identifies cell type–specific genetic control of autoimmune disease. *Science (80-.)*. **376**, eabf3041 (2024).
22. Roche, P. A. & Furuta, K. The ins and outs of MHC class II-mediated antigen processing and presentation. *Nat. Rev. Immunol.* **15**, 203–216 (2015).
23. Travaglini, K. J. *et al.* A molecular cell atlas of the human lung from single-cell RNA sequencing. *Nature* **587**, 619–625 (2020).
24. Wang, X., Wong, K., Ouyang, W. & Rutz, S. Targeting IL-10 Family Cytokines for the Treatment of Human Diseases. *Cold Spring Harb. Perspect. Biol.* **11**, (2019).
25. Castellino, F. *et al.* Chemokines enhance immunity by guiding naive CD8+ T cells to sites of CD4+ T cell-dendritic cell interaction. *Nature* **440**, 890–895 (2006).
26. Khan, M., Zhao, Z., Arooj, S., Fu, Y. & Liao, G. Soluble PD-1: Predictive, Prognostic, and Therapeutic Value for Cancer Immunotherapy . *Frontiers in Immunology* **11**, (2020).
27. Yu, G., Wang, L.-G., Han, Y. & He, Q.-Y. clusterProfiler: an R Package for Comparing Biological Themes Among Gene Clusters. *Omi. A J. Integr. Biol.* **16**, 284–287 (2012).
28. Finan, C. *et al.* The druggable genome and support for target identification and validation in drug development. *Sci. Transl. Med.* **9**, (2017).
29. Ceuppens, J. L. & Baroja, M. L. Monoclonal antibodies to the CD5 antigen can provide the necessary second signal for activation of isolated resting T cells by solid-phase-bound OKT3. *J. Immunol.* **137**, 1816–1821 (1986).
30. Wang, C. *et al.* CD5L/AIM Regulates Lipid Biosynthesis and Restrains Th17 Cell Pathogenicity. *Cell* **163**, 1413–1427 (2015).
31. Orta-Mascaró, M. *et al.* CD6 modulates thymocyte selection and peripheral T cell homeostasis. *J. Exp. Med.* **213**, 1387–1397 (2016).

REVIEWER COMMENTS

Reviewer #1 (Remarks to the Author):

The authors are to be commended on their excellent work in response to the prior review. There are a few comments remaining:

1. Major: the manuscript cited Sakaue et al. several times but some of them did not specify which Sakaue's paper was referred since there are three papers of Sakaue et al. in the Reference, e.g., the end of Introduction, Fig 1, Fig 4, and 3rd paragraph of Discussion.
2. To our Major Comment #8 from the prior review: still need reference(s) to support your revised statement "These amino acid positions in DRB1 have previously been shown to be associated with risk of type 1 diabetes and rheumatoid arthritis".
3. To our Major Comment #13: do not see revisions in the Discussion as stated in the response that "our HLA-pQTL analyses-both single marker tests and fine-mapping analyses-strongly implicate amino acid positions within the peptide binding groove", since your revised results suggested that the implication of amino acid within the PBG was concluded from both single marker test and fine-mapping results. Please make sure the revised text is incorporated in the manuscript.

Reviewer #2 (Remarks to the Author):

The authors have done well to systematically address my major concerns regarding the complexity of the MHC region. The addition of transcriptomics data and clear contrasts to prior proteomics gwas from flagship paper has significantly improved the quality and novelty of the analyses.

Reviewer #3 (Remarks to the Author):

I thank the authors for addressing my questions carefully. I have some more questions regarding the revised manuscript.

1. It is difficult to understand that "This analysis revealed that while non-HLA pQTL can explain up more than 80% of the protein expression variance". It seems that the protein expression of pGenes were largely determined by genetics (~20% by HLA and ~80% by non-HLA pQTL). Is there possible to run GCTA to estimate the heritability of some of these proteins? Such as LRP1, MICB_MICA with two-component models?
2. 5021 individuals with HLA-associated diseases were removed from the analysis. A list of ICD-10 codes for these diseases as supplementary file will be great. Are these diseases enriched more in the UKB-PPP cohort than full UKBB cohort?
3. Figure 1e, "the most significant signal observed for conditional haplotype analyses at the HLA-B locus was for the MICB_MICA protein". MICB and MICA are coded by two genes. Were their protein expression measured in UKB-PPP as single protein?

Reviewer #4 (Remarks to the Author):

The authors did great job to respond my prior review. There are a few comments remaining:

1. Major: the manuscript cited Sakaue et al. several times but some of them did not specify which Sakaue's paper was referred since there are three papers of Sakaue et al. in the Reference, e.g., the end of Introduction, Fig 1, Fig 4, and 3rd paragraph of Discussion.
2. To our Major Comment #8 from the prior review: still need reference(s) to support your revised statement "These amino acid positions in DRB1 have previously been shown to be associated with risk of type 1 diabetes and rheumatoid arthritis".
3. To our Major Comment #13: do not see revisions in the Discussion as stated in the response that "our HLA-pQTL analyses-both single marker tests and fine-mapping analyses-strongly implicate amino acid positions within the peptide binding groove", since your revised results suggested that the implication of amino acid within the PBG was concluded from both single marker test and fine-mapping results. Please make sure the revised text is incorporated in the manuscript.

Response to Reviewers' Comments

Research article for Nature Communications ID: NCOMMS-23-38578A

“The influence of HLA genetic variation on plasma protein expression”
Krishna *et al.*

We are very grateful to the editor and reviewers for their positive assessment of our revised manuscript. Below we address the remaining points raised by all reviewers, providing minor additions to the text and supplementary tables as requested, in addition to clarification regarding the methods for the heritability and disease sensitivity analyses.

In summary, our revised HLA-pQTL manuscript represents a comprehensive investigation of the fine-mapped effects of HLA genetic variation on plasma protein expression, with extensive genomic follow-up analyses to link HLA-pGenes to gene families, cell types, and disease risk. Our manuscript greatly extends our understanding of HLA biology and should serve as a critical resource for future investigation aimed at understanding the mechanisms underlying HLA genetic associations with disease.

As before, we provide a point-by-point response to the reviewers in blue, and new additions to the revised manuscript are highlighted in yellow throughout. We hope that the additions included herein comprehensively address the reviewers' remaining questions.

Point-by-point response to reviewers' comments:

Reviewer 1:

Reviewer #1 (Remarks to the Author):

The authors are to be commended on their excellent work in response to the prior review. There are a few comments remaining:

We thank the reviewer for their rigorous suggestions on the first version of the manuscript, the incorporation of which has greatly strengthened the revised version. We also thank the reviewer for their positive assessment of our revised manuscript and address their remaining points below.

1. Major: the manuscript cited Sakaue et al. several times but some of them did not specify which Sakaue's paper was referred since there are three papers of Sakaue et al. in the Reference, e.g., the end of Introduction, Fig 1, Fig 4, and 3rd paragraph of Discussion.

Response:

We apologize to the reviewer for the confusion—we have now ensured that all references to Sakaue *et al.* in the text refer specifically to the Sakaue *et al.* UK Biobank study published in Nature Genetics 2021. Furthermore, one of the other Sakaue *et al.* references (the HLA imputation tutorial) was a preprint; since the published version (Sakaue *et al.* Nature Protocols 2023) is also cited, we have now removed the citation to the preprint so only two Sakaue *et al.* papers (Nature Genetics 2021 and Nature Protocols 2023) are in the revised references section.

To avoid any ambiguity and to clarify that all references to Sakaue *et al.* in the text and figure legends correspond to the Sakaue *et al.* Nature Genetics 2021 study, we have amended the first mention of Sakaue *et al.* in the main text as follows:

“Finally, we integrated our fine-mapped HLA-pQTL with fine-mapped HLA-disease analyses in the UK Biobank from Sakaue et al. Nature Genetics 2021¹² (hereafter ‘Sakaue et al. 2021’) to create an atlas and of potential HLA-pQTL effects on disease, and hypotheses about the mechanisms underlying HLA genetic associations with disease.”

2. To our Major Comment #8 from the prior review: still need reference(s) to support your revised statement “These amino acid positions in DRB1 have previously been shown to be associated with risk of type 1 diabetes and rheumatoid arthritis”.

Response:

We apologize to the reviewer for this omission and have now added the required references to the sentence the reviewer points out in the text.

3. To our Major Comment #13: do not see revisions in the Discussion as stated in the response that “our HLA-pQTL analyses-both single marker tests and fine-mapping analyses-strongly implicate amino acid positions within the peptide binding groove”, since your revised results suggested that the implication of amino acid within the PBG was concluded from both single marker test and fine-mapping results. Please make sure the revised text is incorporated in the manuscript.

Response:

We apologize to the reviewer for this omission- per the reviewer’s suggestion, the revised discussion now has this sentence from the first round of response to this reviewer, and is reproduced below:

“We hypothesize that there may be two primary mechanisms by which HLA polymorphism affects protein expression. First, as our HLA-pQTL analyses—both single marker tests and fine-mapping analyses—strongly implicate amino acid positions within the peptide binding groove, we suspect that variation in antigen presentation affects T and NK cell recognition of peptide-MHC, which in turn may modulate the expression levels of proteins related to T and NK effector function.”

Reviewer 2:

Reviewer #2 (Remarks to the Author):

The authors have done well to systematically address my major concerns regarding the complexity of the MHC region. The addition of transcriptomics data and clear contrasts to prior proteomics gwas from flagship paper has significantly improved the quality and novelty of the analyses.

We very much appreciate the reviewer's thoughtful and constructive questions during the review process, which have led to a greatly enhanced revised manuscript.

Reviewer 3:

Reviewer #3 (Remarks to the Author):

I thank the authors for addressing my questions carefully. I have some more questions regarding the revised manuscript.

We greatly appreciate the reviewer's constructive reading of the first version of our manuscript and positive assessment of the revised version. Below we provide point-by-point responses to the reviewer's outstanding questions.

1. It is difficult to understand that "This analysis revealed that while non-HLA pQTL can explain up more than 80% of the protein expression variance". It seems that the protein expression of pGenes were largely determined by genetics (~20% by HLA and ~80% by non-HLA pQTL). Is there possible to run GCTA to estimate the heritability of some of these proteins? Such as LRP1, MICB_MICA with two-component models?

Response:

We apologize to the reviewer for the confusion regarding the protein heritability estimates. It is important to clarify that we obtained the SNP heritability estimates directly from the flagship UKB-PPP paper¹ (Supplementary Table 19 of Sun *et al.* Nature 2023). In the flagship paper, the authors computed the SNP-based heritability using all significant pQTLs in addition to remaining SNPs across the genome excluding the pQTL region (i.e. polygenic component, computed using LD-score regression).

Thus, the total SNP heritability for each protein is the sum of the heritability attributable to the pQTLs + heritability attributable to polygenic effects. This total heritability can then be partitioned into cis heritability and trans heritability; the latter (trans heritability) is what we originally displayed on the y axis of our prior revised manuscript.

Indeed, reviewing the total SNP heritability estimates in Supplementary Table 19 of that paper, for some proteins that are also HLA-pGenes in our analysis, the total SNP heritability is indeed high- for example, for MICB_MICA, the total SNP heritability is 84.73%; for LRP1, the total SNP heritability is 35.52%.

For the reviewer's convenience, we have included below the relevant sections on heritability calculation from the flagship UKB-PPP paper, which show that heritability was calculated using established methods such as LDSC:

From the flagship paper main text:

"We estimated single-nucleotide polymorphism (SNP)-based heritability as the sum of contributions from significant independent pQTLs identified by SuSiE (pQTL component) and the remaining SNPs across the genome (excluding the pQTL region), which assumes a polygenic model (polygenic component) using an approach that was described previously (Methods and Supplementary Table 19)."

From the flagship paper methods:

“We estimated the SNP-based heritability as a sum of variance explained from the independent pQTLs through the SuSiE analyses for each protein at each loci (pQTL component) and the polygenic component using the genome-wide SNPs excluding the pQTL regions of each protein. The polygenic component, which mostly likely satisfies the polygenic model of small genetic contributions across the genome, was estimated using LD-score regression.”

We certainly acknowledge the reviewer’s suggestion to use GCTA to investigate heritability for some proteins. However, both LDSC (which was used for heritability estimation in the flagship paper) and GCTA are both widely used for heritability estimation² and can perform similarly³. Indeed, in the Speed *et al.* paper “Reevaluation of SNP heritability in complex human traits”, the authors compare GCTA and LDSC across a variety of traits and state “*Estimates from LDSC were not significantly different to those from GCTA, which is to be expected considering that GCTA and LDSC assume the same relationship between heritability and LD.*”³

Moreover, a recent analysis comparing multiple methods for estimating total SNP heritability—including GCTA and LDSC, stated that “*Overall, our results show that while existing methods can yield biases, for the purpose of estimating total SNP-heritability, most methods are relatively robust.*”⁴

Importantly, we wish to present fair and consistent comparisons between data from our paper and data from the flagship paper. For these reasons, we feel it is best to use the total SNP heritability estimates as originally computed by the flagship paper. In our view, recomputing the total SNP heritability with GCTA—which would require extensive effort through computation of a GRM using large-scale, individual-level data from the UK Biobank—would be redundant given the already-computed heritability estimates from the flagship study.

However, the reviewer’s question has prompted us to present the heritability data in a more intuitive way. Instead of plotting the HLA-pQTL heritability against the genome-wide trans heritability as we did previously, we have now presented a simpler plot- we show the HLA-pQTL heritability against the total SNP heritability described above.

This allows the reader to see the fraction of the total heritability explained by HLA-pQTL. Furthermore, as the y-axis shows the total SNP heritability, we can clearly see that there is a wide range of heritability values, with some having very low heritability, and others high, as perhaps expected in a sample size this large:

f**Proportion variance explained by HLA-pQTL compared to all SNPs (Sun *et al.* Nature 2023)**
Fig. R3.1 (Extended Data Fig. 3f): Proportion of protein expression variance explained by lead and conditional HLA-pQTL compared to all SNPs from the flagship UKB-PPP analysis. Parentheses indicate whether the protein is located within or outside the MHC region.

As before, the HLA-pQTL heritability and total SNP heritability are weakly positively correlated; in addition, for the genes that the reviewer points out (MICB_MICA and LRP1), HLA-pQTL explain the majority of the total SNP heritability:

Fig. R3.2: Total SNP heritability for MICB_MICA and LRP1, separated by HLA-pQTL heritability and non-HLA-pQTL heritability. Percentages in each bar show the proportion variance explained by either HLA-pQTL (blue) or non-HLA-pQTL (red).

We have thus replaced Extended Data Fig. 3f with the revised, more intuitive correlation plot shown above in Fig. R3.1 with total SNP heritability on the y axis. The methods section of the revised manuscript has also been revised to clarify that the total SNP heritability estimates were obtained directly from the flagship study, together with a brief description of the flagship study methods/LDSC used to compute total SNP heritability.

We have also amended the text that describes Extended Data Fig. 3f to be clearer, as follows:

“Comparing the variance explained by HLA-pQTL to variance explained by all SNPs from the flagship UKB-pQTL study, we found that for several pGenes (e.g. MICB_MICA; LRP1), the majority of the total SNP variance explained is accounted for by HLA genetic variation (Extended Data Fig. 3f)”

2. 5021 individuals with HLA-associated diseases were removed from the analysis. A list of ICD-10 codes for these diseases as supplementary file will be great. Are these diseases enriched more in the UKB-PPP cohort than full UKBB cohort?

Response:

We thank the reviewer for this suggestion and are happy to provide a supplementary table with the ICD10 codes for the HLA-associated diseases. This is now provided in supplementary table 4, together with the summary statistics of the HLA-pQTL from the sensitivity analysis removing individuals with those diseases. The table is reproduced below for the reviewer's convenience:

Disease	ICD10
Skin cancer	C43/C44
Prostate cancer	C61
Malignant lymphoma	C81/C82/C83/C84/C85/C86/C88/C90/C91/C92/C93/C94/C95/C96
Iron deficiency anemia	D50
Sarcoidosis	D86
Hypothyroidism	E03
Hyperthyroidism / Graves' disease	E05
Type 1 diabetes	E10
Type 2 diabetes	E11
Iritis / Uveitis	H20
Nasal polyp	J33
Asthma	J45
Gastric polyp	K317
Ulcerative colitis	K51
Autoimmune hepatitis	K754
Psoriasis vulgaris	L40
Rheumatoid arthritis	M06
Systemic lupus erythematosus	M32
Sjogren Syndrome	M350
IgA_nephritis	N028
Chronic Glomerulonephritis	N03
Nephrotic Syndrome	N04
Celiac disease	K90
Multiple sclerosis	G35
Ankylosing spondylitis	M45

Table R3.2 (Supplementary Table 4, tab 1 of the revised manuscript): List of HLA-associated diseases and ICD10 codes in the UK Biobank identified in Sakaue *et al.* Nature Genetics 2021⁵, plus Celiac disease, Multiple sclerosis, and Ankylosing spondylitis, three major immune diseases which were not associated with HLA in

Sakaue *et al.* Nature Genetics 2021 but have previously been shown to be associated with HLA DQ2⁶ HLA DR15⁷, and HLA B27⁸ respectively.

The reviewer's question prompted us to carefully review this analysis again; upon doing so, we caught an error in the number of individuals with HLA-associated diseases- the true number of patients removed is 6833, rather than 5021. We apologize for this mistake. We thus re-ran the disease sensitivity analysis originally proposed by the reviewer by removing this correct number individuals.

Reassuringly, this had hardly any effect on the overall conclusions—of the 504 pGenes, 500/504 (99.2%) remained significant at $P < 5.0 \times 10^{-8}$, and the effect sizes remained strongly correlated (Extended Data Fig. 3e; $P < 2.2e-16$; Spearman rho = 0.99). For the 4 remaining genes that did not meet the genome-wide significance threshold, the p-values from the sensitivity analysis strongly trended towards significance; thus, we can conclude that even after removing individuals with HLA-associated diseases, our HLA-pQTL model is extremely robust. Indeed, the strong robustness of our model even after removing individuals with HLA-associated diseases may be due to the incorporation of protein expression principal components, which as we show in Extended Data Fig. 1b, capture variation due to the presence of diseases in the UK Biobank.

Protein	allele	beta_original	beta_removing_diseases	pval_original	pval_removing_diseases
CCL14	AA_DRB1_74_32551940_exon2_GLR	-0.03818629	-0.030017551	2.81E-12	8.78E-07
CST7	AA_B_156_31324024_exon3_DGx	0.03709942	0.029995004	1.00E-11	8.35E-07
GGH	AA_B_97_31324201_exon3_CS	-0.03839375	-0.032555893	3.00E-12	1.17E-07
TSHB	AA_DRB1_98_32549641_exon3_Ex	0.03707617	0.030225629	1.05E-11	6.93E-07

Table R3.3: 4 genes that did not meet the genome-wide significance threshold of 5.0×10^{-8} after removing individuals with HLA-associated diseases. Table shows original beta and p-value, and beta and p-value after removing individuals with HLA-associated diseases.

We next sought to address the reviewer's question regarding enrichment of these HLA-associated diseases relative to the full UKB. Examining the proportions of individuals in either UKB-PPP or full UKB with HLA-associated diseases revealed no enrichment in UKB-PPP, consistent with the original description of the UKB-PPP in the flagship paper as not being enriched or depleted for any particular diseases:

Fig. R3.4: Proportion of individuals with HLA-associated diseases in the UKB-PP discovery cohort and the full UKB cohort. Fractions above each bar show the total number of individuals with HLA-associated diseases (6833 and 99032 in UKB-PPP and Full UKB respectively) out of the total number of individuals (34490 and 502412 in UKB-PPP and Full UKB respectively). P-value calculated using two-sided proportion test.

We have thus amended the text to include the supplementary table suggested by the reviewer and a description of the results demonstrating strong robustness of our HLA-pQTLs in the disease sensitivity analysis:

*“Given the strong associations of HLA genetic variation with immune-mediated diseases, we explored whether removing individuals with HLA-associated diseases—either major autoimmune diseases identified from prior bespoke GWAS, or HLA-associated diseases identified in an analysis of the UK Biobank by Sakaue et al 2021.—impacted the discovery of the 504 lead HLA-pQTL in the discovery cohort. We identified 6833 individuals with diagnosis or death due to immune diseases (identified via ICD-10 codes) in the UKB discovery cohort (**Supplementary Table 4**). We removed these individuals from the discovery cohort and repeated the analysis. We found that of the 504 pGenes discovered initially, 500/504 (99.2%) remained significant at $P < 5 \times 10^{-8}$ (**Supplementary Table 4**). Effect sizes were also strongly correlated between the full discovery cohort and discovery cohort without HLA-associated diseases (**Extended Data Fig. 3e**). These data suggest that the HLA-pQTL reported in our study are not confounded by the presence of HLA-associated diseases in the UK Biobank, and underscore the robustness of our HLA-pQTL model.”*

3. Figure 1e, “the most significant signal observed for conditional haplotype analyses at the HLA-B locus was for the MICB_MICA protein”. MICB and MICA are coded by two genes. Were their protein expression measured in UKB-PPP as single protein?

Response:

We apologize to the reviewer for the confusion regarding MICB_MICA—indeed, as listed in supplementary table 3 of the flagship UKB-PPP pQTL paper (Sun *et al.* Nature 2023), MICB_MICA was measured as a single analyte, likely due to their similar function, i.e. both are ligands for the NKG2D receptor⁹, and highly similar structures¹⁰. The Olink MICA_MICB assay therefore detects both MICA and MICB and cannot differentiate between the two, as shown here- <https://olink.com/products-services/explore/protein/?proteinID=OID20593>

We have amended the sentence the reviewer describes to clarify this point as follows:

“For example, the most significant signal observed for conditional haplotype analyses at the HLA-B locus was for MICB_MICA (MICB and MICA were measured together as a single analyte) (Fig. 1e).”

Reviewer 4:

Reviewer #4 (Remarks to the Author):

The authors did great job to respond my prior review. There are a few comments remaining:

We thank the reviewer for their excellent questions throughout the review process and thank the reviewer for these additional questions. As these questions are repeated in reviewer 1's second round of review, we kindly refer this reviewer to our responses to reviewer 1.

1. Major: the manuscript cited Sakaue et al. several times but some of them did not specify which Sakaue's paper was referred since there are three papers of Sakaue et al. in the Reference, e.g., the end of Introduction, Fig 1, Fig 4, and 3rd paragraph of Discussion.
2. To our Major Comment #8 from the prior review: still need reference(s) to support your revised statement "These amino acid positions in DRB1 have previously been shown to be associated with risk of type 1 diabetes and rheumatoid arthritis".
3. To our Major Comment #13: do not see revisions in the Discussion as stated in the response that "our HLA-pQTL analyses-both single marker tests and fine-mapping analyses-strongly implicate amino acid positions within the peptide binding groove", since your revised results suggested that the implication of amino acid within the PBG was concluded from both single marker test and fine-mapping results. Please make sure the revised text is incorporated in the manuscript.

References

1. Sun, B. B. *et al.* Plasma proteomic associations with genetics and health in the UK Biobank. *Nature* **622**, 329–338 (2023).
2. Evans, L. M. *et al.* Comparison of methods that use whole genome data to estimate the heritability and genetic architecture of complex traits. *Nat. Genet.* **50**, 737–745 (2018).
3. Speed, D. *et al.* Reevaluation of SNP heritability in complex human traits. *Nat. Genet.* **49**, 986–992 (2017).
4. Hou, K. *et al.* Accurate estimation of SNP-heritability from biobank-scale data irrespective of genetic architecture. *Nat. Genet.* **51**, 1244–1251 (2019).
5. Sakaue, S. *et al.* A cross-population atlas of genetic associations for 220 human phenotypes. *Nat. Genet.* **53**, 1415–1424 (2021).
6. Sollid, L. M. & Lie, B. A. Celiac Disease Genetics: Current Concepts and Practical Applications. *Clin. Gastroenterol. Hepatol.* **3**, 843–851 (2005).
7. Hollenbach, J. A. & Oksenberg, J. R. The immunogenetics of multiple sclerosis: A comprehensive review. *J. Autoimmun.* **64**, 13–25 (2015).
8. Hwang, M. C., Ridley, L. & Reveille, J. D. Ankylosing spondylitis risk factors: a systematic literature review. *Clin. Rheumatol.* **40**, 3079–3093 (2021).
9. Lanier, L. L. NKG2D Receptor and Its Ligands in Host Defense. *Cancer*

- Immunol. Res.* **3**, 575–582 (2015).
10. Raulet, D. H., Gasser, S., Gowen, B. G., Deng, W. & Jung, H. Regulation of ligands for the NKG2D activating receptor. *Annu. Rev. Immunol.* **31**, 413–441 (2013).

REVIEWERS' COMMENTS

Reviewer #3 (Remarks to the Author):

The authors have done well to address my comments.